

# The Antarctic sea ice cover from ICESat-2 and CryoSat-2: freeboard, snow depth and ice thickness

Sahra Kacimi[1], Ron Kwok[1]

[1]Jet Propulsion Laboratory, California Institute of Technology, Pasadena, California, USA

*Correspondence to*: Sahra Kacimi (sahra.kacimi@jpl.nasa.gov)

**Abstract** We offer a view of the Antarctic sea ice cover from lidar (ICESat-2) and radar (CryoSat-2) altimetry, with retrievals of freeboards, snow depth, and ice volume that span an 8-month winter between April, 2019 and November 16, 2019. Snow depths are from freeboard differences. The multiyear ice in the West Weddell sector stands out with a mean sector thickness > 2 m. Thinnest ice is found near polynyas (Ross Sea and Ronne) where new ice areas are exported seaward and entrained in the
surrounding ice cover. For all months, the results suggest that ~60-70% of the total freeboard is comprised of snow. The remarkable response of the ice cover to mechanical convergence in the coastal Amundsen Sea, associated with onshore winds, was captured in the correlated increase in local freeboards and thickness. While the spatial patterns in the freeboard, snow depth, and thickness composites are as expected, the observed seasonality in these variables is surprisingly weak likely attributable to competing processes (snowfall, snow redistribution, snow-ice formation, ice deformation, basal growth/melt) that contribute to
uncorrelated changes in the total and radar freeboards. Broadly, evidence points to biases in CryoSat-2 freeboards of at least a few centimeters from high salinity snow (>10 psu) in the basal layer resulting in lower/higher snow depth/ice thickness retrievals although the extent of these areas cannot be established in the current data set. Adjusting CryoSat-2 freeboards by 3/6 cm gives a circumpolar ice volume of 14,700/12,400 $km^3$ in October, for an average thickness of ~1.09/0.93 m. Validation of Antarctic sea ice parameters remains a challenge, there are no seasonally and regionally diverse data sets that could be used to assess these
large-scale satellite retrievals.



## 1 Introduction

The gradual increase in Antarctic sea ice extents in satellite records over the last several decades reversed in 2014, with subsequent rates of decrease in 2014–2019 exceeding the decay rates in the Arctic. For these past years, the Antarctic sea ice extents were reduced to their lowest levels in the 40-year satellite record (Parkinson, 2019). Our current understanding of the behavior of the Antarctic ice cover is largely informed by these ice coverage measurements from passive microwave sensors. Ice extent, however, paints only a limited picture of sea ice response to climate change and variability. But, even with the large observed changes, available measurements of Antarctic sea ice thickness are still too few to be able to judge whether the total ice production and volume of the ice cover are decreasing, steady, or increasing (Vaughan et al., 2013)

Prior to the 2014 decline in Antarctic ice extent, coupled ice-ocean models have suggested that significant changes in ice volume and thickness have accompanied ice extent changes (Massonnet et al., 2013; Holland et al., 2014), and increases in ice thickness may have been driven by the intensification of the wind field (Zhang, 2014) noted by Holland and Kwok (2012). As well, fully-coupled climate models generally fail to capture the observed trends and variability in ice coverage during the last few decades (e.g., Mahlstein et al., 2013; Polvani & Smith, 2013; Zunz et al., 2013; Hobbs et al., 2015; Turner et al., 2015). However, large-scale estimates of ice thickness and ice production necessary to improve attribution of change, model evaluation and improvements, and for projection of future behavior have been challenging to obtain. Satellite altimeters cannot yet determine ice thickness reliably in the Antarctic, largely due to uncertainties in snow depth and freeboard (Giles et al., 2008), required for computing snow loading in the conversion of freeboard to thickness.

Wide discrepancy between ice thickness estimates from recent approaches to determine sea ice thickness persists (Yi et al., 2011; Kurtz & Markus, 2012; Xie et al., 2013). Current algorithms to derive ice thickness from data collected by ICESat-1 (Ice, Cloud, and land Elevation Satellite) have relied on either an independent measure of snow depth (Yi et al., 2011), assumed that the snow depth is equal to the ice freeboard (Kurtz & Markus, 2012), or used empirical relationships between total freeboard and ice thickness determined from field data (Xie et al., 2013). All these approaches have limitations because of simplifying assumptions about the ice cover. The first approach tends to underestimate of snow depth in areas of deformed ice. The second seems more appropriate for the thinner ice in the outer pack with low ice thickness. The third method may be most suitable for thicker ice, where knowledge of densities is subsumed into the regression coefficients. Such empirical relationships vary seasonally and regionally (Ozsoy-Cicek et al., 2013), and so the confidence in the derivations is reduced. Nevertheless, these approaches have provided the best large-scale estimates of the spatial variability of the ice and snow cover based on available knowledge of the Antarctic ice cover.

With the launch of NASA's ICESat-2 (IS-2) in late 2018 and the extension of ESA's CryoSat-2 (CS-2) mission, we are now able to combine lidar and radar altimetry of the Arctic and Antarctic ice covers from IS-2 and CS-2 for understanding ice behavior. A recent paper by Kwok et al. (2020) demonstrated the retrieval of basin-scale estimates of both Arctic snow depth and sea ice thickness from differences in IS-2 and CS-2 freeboards. Here, we follow the same approaches to examine the large-scale seasonal cycle of the Antarctic freeboards, retrieved snow depth and ice thickness from a joint analysis of IS-2 and CS-2 data (between April and November of 2019). We note at the outset that the results from this study remains exploratory because of current understanding of the snow cover of Antarctic sea ice. There are many aspects of data quality, some of which will only be revealed by assessment with snow data acquired and processed by dedicated airborne campaigns (e.g., NASA's Operation IceBridge), field programs and when a longer IS-2/CS-2 time series becomes available.

The paper is organized as follows. The next section describes the IS-2 and CS-2 freeboard data sets used in our analysis. In Section 3, we first discuss the key processes that contribute to the time evolution of Antarctic freeboards, and then describe the observed evolution of the two freeboards during the eight winter months. Section 4 outlines the principle behind the derivation of





snow depth from freeboard differences, the sampling of the satellite freeboards for calculation of snow depth, and the derived monthly estimates. Section 5 compares the thickness and volume of the Antarctic ice cover computed using the derived snow depth, and assuming that snow depth is equal to the total freeboard. Potential biases in the data are discussed. Section 6 concludes the paper by the highlighting these first observations and discuss challenges in having the appropriate data sets for

assessment of the retrievals from the two altimeters.

## 2 Data description

We use the following data sets: IS-2 and CS-2 freeboards, and surface heights from the ATM lidar and snow depth estimates from snow-radar; their attributes are described below.

### 2.1 ICESat-2 (IS-2) freeboards

Along-track freeboards are from the ICESat-2 ATL10 products (Release 002) from the National Snow and Ice Data Center (Kwok et al., 2019a). The ATL10 product provides sea ice freeboard estimates in 10-km segments that contain a sea surface reference. Local sea surface references ( $h_{ref}$ ) (i.e., the estimated local sea level) are from available sea ice leads within a 10-km segment. Freeboard heights ( $h_f$ ), in 10-km segments, are the differences between surface heights ( $h_s$ ) and the local sea surface reference (i.e., $h_f = h_s - h_{ref}$ ). For the current release (002) of data products, freeboard profiles are calculated for individual

beams and there is no dependence on the sea surface references used by the other beams. In ATL10, freeboards are calculated only where the ice concentration is >50% and where the height samples are at least 25-km away from the coast (to avoid uncertainties in coastal tide corrections). Details of the sea ice algorithms can be found in Kwok et al. (2019b) and an early assessment of surface heights are in Kwok et al. (2019c). Only the freeboards from the strong beams are used in the following analyses. We also note that IS-2 coverage is only available for the first two weeks of November 2019 in Release 002.

Uncertainty in IS-2 freeboard retrievals is ~2-4 cm based on assessment in Kwok et al. (2019c).

### 2.2 CS-2 radar freeboards

Along-track CS-2 freeboards are derived using the procedure in Kwok and Cunningham (2015), which contains a detailed description of the retrievals and an assessment of these freeboard estimates. The CS-2 freeboards used here have been weighted by satellite-derived ice concentration. As there are no direct assessments of these freeboard estimates, comparisons with

25 available ice thickness measurements provide an indirect measure of data quality – freeboard is approximately one-ninth of ice thickness. The assessed differences between CS-2 and various thickness measurements are (Kwok & Cunningham, 2015): 0.06±0.29 m (ice draft from moorings), 0.07±0.44 m (submarine ice draft), 0.12±0.82 m (airborne electromagnetic profiles), and -0.16±0.87 m (OIB) .

## 3 IS-2 and CS-2 freeboards

In this section, we first discuss expected changes in IS-2 and CS-2 freeboards based on our understanding of the key processes before examining the spatial patterns and distributions of the monthly freeboards. Here, we divide the circumpolar Southern Ocean into five main sectors, namely: Weddell Sea, Amundsen Sea/Bellingshausen Sea, Ross Sea, Pacific Ocean, and Indian Ocean (Figure 1); these are typically used in ice extent analyses (Comiso & Nishio, 2008). Further, we sub-divide the Weddell sector into an east sector and west sector, and added a coastal Amundsen-Bellingshausen region to sample to impact of

the remarkable ice convergence seen in 2019.





### 3.1 Interpretation of time-varying IS-2 and CS-2 freeboards

Since this is the first large-scale examination of the combined IS-2 and CS-2 freeboards of the Antarctic ice cover, it is worthwhile reviewing the key processes that contribute to regional-scale freeboard changes to aid in the interpretation of the observations. As a reminder, the changes in total freeboard ($\Delta h_f$) are the sum of the changes in thickness of the snow layer

($\Delta h_{fs}$)and changes in ice freeboard ($\Delta h_i$), i.e., $\Delta h_f(t) = \Delta h_{fs}(t) + \Delta h_i(t)$(Figure 2). In the winter Arctic, there are three key processes that contribute to the changes in total freeboard: basal growth, ice deformation, and snow accumulation/redistribution. And, since the Arctic Ocean export only ~10% of its area annually (mainly through the Fram Strait) (Kwok et al., 2013) and there is relatively little melt in winter away from the ice margins, it is simpler to observe a coherent seasonal cycle of freeboard growth (i.e.. the correlated increases in both the IS-2 and CS-2 freeboards seen in Kwok et al. (2020)) over a fixed region of the

Arctic Basin. In the Antarctic, however, the heavier snowfall (Massom et al., 1997), ice production in large coastal polynyas (Drucker et al., 2011), formation of snow-ice (Jeffries et al., 2001; Maksym & Markus, 2008), larger ice divergence (i.e., production of areas of open water), and the continuous large-scale export of sea ice towards the ice margins (where the ice melts) (Kwok et al., 2017) add complexity to the interpretation of the seasonal evolution of freeboards.

Below, we briefly summarize five key processes that contribute to the modification of the total freeboard ($h_f$) of a drifting

ice parcel during the Antarctic winter. Separating the contributions from the snow ($h_{fs}$) and ice layers ($h_{fi}$), we write,

$$\Delta h_{fs}(t) = \delta h_{snow} + \delta h_{\phi} - \delta h_{sti} + \delta h_{def}^s$$
$$\Delta h_i(t) = -\alpha(\delta h_{snow} + \delta h_{\phi}) + \beta \delta h_{sti} + \delta h_{def}^i + \delta h_{gm} \tag{1}$$

$\alpha$ and $\beta$ are scale factors, and signs indication the addition or removal of height from these layers. The $\delta h$'s are described below:

1)   Snowfall ($\delta h_{snow}$) adds to the snow layer but the loading depresses the ice freeboard by $-\alpha \delta h_{snow}$. $\alpha$ is a fractional

fvalue, and in this case is dependent on the densities of ice, snow, and seawater.

2)   Spatial redistribution of snow including loss into leads ($\delta h_{\phi}$): Snow is redistributed spatially due to wind forcing and is sometimes lost into open leads ($\delta h_{\phi}$); the ice freeboard adjusts hydrostatically by $-\alpha \delta h_{\phi}$.

3)   Snow-ice formation:  When sea water infiltrates the snow layer during flooding, the refrozen ice layer becomes part of the ice freeboard and this results in a loss of $\delta h_{sti}$ from the snow layer (i.e., the snow pack settles when flooded) and a

gain of $\beta \delta h_{sti}$ in the ice freeboard. $\beta$ represents the fraction of the snow thickness that is converted to ice freeboard after the transformation process.

4)   Ice deformation (convergence and divergence of the ice cover): Mechanical convergence/divergence of the ice cover tends to increase/decrease the area-averaged thickness of the snow layer ($\delta h_{def}^s$) and ice freeboard ($\delta h_{def}^i$). The relationship between $\delta h_{def}^s$ and $\delta h_{def}^i$ may be more complicated and hence written separately.

5)   Basal ice growth/melt ($\delta h_{gm}$) of sea ice adds/removes from the ice freeboard and increases/decrease the total freeboard.

This brief summary is no doubt a simplification and there are higher order processes such as changes due to snow metamorphism but their area-averaged contributions to freeboard changes are likely to be small. Another factor (note above) to bear in mind in the interpretation of regional changes of freeboard (below) is the large export and sea ice melt at the margins.



### 3.2 Monthly composites IS-2 and CS-2 freeboards

Figure 3 shows the monthly composites of IS-2 and CS-2 freeboards, for April through November 2019. The associated freeboard distributions are shown in Figure 4. The numerical values and sample statistics of the monthly distributions are in Table 2. We examine freeboard distributions of the seven sectors in the following order: Amundsen-Bellingshausen (A-B), coastal Amundsen-Bellingshausen (CoA-B), East and West Weddell (E-Wedd, W-Wedd), Ross, Pacific Ocean, and Indian Ocean.

#### 3.2.1    Amundsen and Bellingshausen Seas sectors (A-B and CoA-B)

The freeboard distributions of the Amundsen and Bellingshausen Seas between the Antarctic Peninsula and 140ºW, are constructed with samples from two sectors (Figures 4a and 4b): one is between coastal Antarctica and 70ºS (referred to as the CoA-B sector) and the other has an open boundary to include the seaward extent of the advancing winter ice edge (A-B sector).

For the eight winter months, the highest variability (amongst the seven sectors) is seen in the CoA-B sector, where the area-averaged IS-2 and CS-2 freeboards range from 29.2±16.6 (min) to 48.2±26.2 (max) cm, and 11.0±6.08 to 17.8±11.8 cm, respectively. The squared correlation ($\rho^2$) between the two freeboards of 0.86 (Figure 4c) – second highest of all seven sectors – indicates that the co-variability may be attributable to a response to the same forcing. Indeed, examination of the monthly maps of ice drift (Figure 5) suggests that the correlated increases in the two freeboards is likely due to the persistent wind-driven convergence of sea ice against the Antarctic coast (west of 90ºW). The resulting ridging in the coastal Amundsen Sea ice cover caused a redistribution of the thinner ice into thicker categories simultaneously increasing both the lidar and radar freeboards. The anomalous on-shore ice drift in 2019 (Figure 5b) can be contrasted to the mean ice drift pattern for the period 2012-2019 (Figure 5a). The large-scale atmospheric pattern in 2019 shows the location and depth of the Amundsen Sea Low (ASL) centered in the northeast Ross Sea (Figure 5b). The circulation setup in 2019 is such that on-shore wind is nearly perpendicular to the coast and the depth of the ASL can be seen in the density of the isobars. The longer tails of the freeboard distributions after May are also signatures of ice convergence, where snow accumulation would unlikely affect the tails of both distributions, i.e., ice freeboard tends to be anti-correlated to snow accumulation. Hence, the freeboard variability here seems to be dominated by wind-driven ice deformation, which masked other processes.

For the A-B sector (which includes the CoA-B sector), the seasonal signal is more muted. The IS-2 and CS-2 freeboards range from 25.3±17.8 to 35.7±25.6 cm, and 9.01±5.84 to 11.4±9.37 cm, and is generally lower because of the thinner seasonal ice cover away from the coastal zone (CoA-B). The squared correlation ($\rho^2$) between the two freeboards of 0.52 (Figure 4c) is also likely connected to due to the large signal in the CoA-B sector in the south. In November, the increase in the IS-2 freeboard not seen in the CS-2 freeboard is potentially due the limited 2-week IS-2 coverage.

#### 3.2.2    East and West Weddell Sea Sectors

The East (E-Wedd) and West Weddell (W-Wedd) sectors are located between 15ºE and 40ºW, and 40ºW and 62ºW, respectively, both with boundaries that are open to the north.  Generally, the W-Wedd sector is one of few areas in the Antarctic where multiyear sea ice is found (Lange & Eicken, 1991). Sea ice formed in the east (E-Wedd sector) is advected clockwise around the southern Weddell Sea, and the older sea ice after its transit is subsequently exported at its northwestern boundary (Figure 5a). Along its drift trajectory, the ice cover becomes thicker and deformed (Lange & Eicken, 1991); as well, younger/thinner ice areas added by mechanical divergence and formed in the Ronne and Brunt ice shelves (Drucker et al., 2011) are entrained in the outflow. The average annual areal export from the southern Weddell Sea (along a flux gate along the 1000 m





isobaths that parallels the ice fronts of the Ronne and Filchner ice shelves) is ~0.32×10⁶ km² (Kwok et al., 2017),  and is comparable to the area of ~0.28×10⁶ km² enclosed by the flux gate of ~1100 km.

In the composite fields (Figure 3), the thicker ice with its higher IS-2 and CS-2 freeboards in the W-Wedd sector is a feature that stands out in the circumpolar Antarctic ice cover. In the 2019 composites, an area of lower IS-2 and CS-2 freeboards (likely of ice formed in Ronne Polynya) is present in the southwestern corner of the Weddell Sea. While there was a decreasing trend in IS-2 freeboard in the eight months of 2019 (Figures 4c), the CS-2 freeboard only varied over a narrow range of ~2 cm (i.e., between 9.72±4.34 and 11.7±4.50 cm). The squared correlation ( $\rho^2$ ) between the two freeboards is 0.05 (Figure 4c). Unlike the clear convergence signal (correlated freeboard time series) in the A-B sectors, this behavior suggests a balance and dominance of different/competing processes discussed earlier. On an area-averaged sense, the processes that would increase the IS-2 freeboards during this winter (e.g., precipitation, convergence, and growth) must have been overwhelmed by processes that would tend to lower the IS-2 freeboards (e.g., snow-ice formation, loss of snow into leads, divergence, and ice export). Similarly, contributions to increases in CS-2 freeboards (due to convergence, growth, snow-ice formation) are likely balanced by precipitation and divergence, even though the CS-2 freeboards tend to be less sensitive to these changes. The tails freeboard distributions in the W-Wedd (Figures 4a and 4b) ice cover also suggest active ice deformation. These processes cannot be resolved at the regional scale that the data is being examined in this paper.

As for the E-Wedd, the higher total and CS-2 freeboards is likely due the area of thicker ice present in the sector early in April and May that become a much smaller fraction of the area of growing ice cover as the sea ice edge advances seaward. As ice coverage grows (Figure 3), the thinner season ice dominates the total area lowering the mean freeboards in the subsequent months.  Both the total and CS-2 freeboards remained within a narrow range after May, again suggesting a balance of different processes that reduced their range of variability. The lowest area-averaged freeboards are found in this sector.

### 3.2.3    Ross Sea Sector

Significant ice production occurs in this sector (between 140ºW and 160ºE). New ice production in the Ross Sea is located primarily in the Ross Shelf Polynya, and the Terra Nova Bay (TNB) and McMurdo Sound Polynyas. Annual ice production here (south of the 1000 m isobaths) is higher than that in the Weddell Sea (Drucker et al., 2011). The average ice area export in a 34-year record is 0.75×10⁶ km² (at a flux gate along the 1000 m isobaths that parallels the ice front of the Ross Sea Ice Shelf). The ~1400 km flux gate encloses an area of ~490×10³ km² to the south. On average, the southern Ross Sea exports more than its area of sea ice that is largely produced in the polynyas.

In all months of 2019, the signature of thinner sea ice with lower freeboards exported from the polynyas can be seen as a distinct tongue (that extends seaward then westward beyond the Ross embayment) in both the IS-2 and CS-2 freeboards composites (Figure 3). The spatial features are consistent with the cyclonic (clockwise) drift pattern associated with the ASL, which was centered over the northeast Ross Sea in all months between June and September (Figure 5b). The drift pattern shows a coastal inflow of thicker sea ice into the Ross Sea from the Amundsen Sea in the east, which is distinctly thicker than the outflow of thinner ice from the southern Ross Sea. North of Cape Adare in the northwest corner of the Ross Sea, the northward drift splits into two branches with one that moves westward into the Somov Sea and the other northeastward before it gets entrained in the Antarctic Circumpolar Current (ACC).

The IS-2 and CS-2 freeboards range from 13.8±6.45 to 21.0±10.2  cm, and 5.55±3.06 to 8.34±3.90 cm, respectively (Figure 4c, Table 1), Both freeboards show a gradual increase with a peak in the IS-2 freeboard during August, likely due to overlapping coverage of the ice convergence events by the A-B (discussed above) and Ross sectors, and to inflow of the thicker deformed ice from the A-B sector. The squared correlation ( $\rho^2$ ) between the two freeboards of 0.89 (Figure 4c) is highest of all the sectors



likely due to the continual production of thin ice in the polynyas, the growth of the thin ice as it is advected northward, and the northward drift and growth of the sea ice from the A-B sector.

### 3.2.4    Pacific and Indian Ocean Sectors

The Pacific and Indian Ocean sectors are located between 160ºW and 90ºE, and 90ºE and 15ºE, respectively. Except for the broader extent of the ice cover in Indian Ocean Sector (around 15ºE and 40ºE) where the winter edge extends into the South Atlantic and Indian Oceans, the ice cover occupies a very narrow band, and extends only ~400 km seaward at maximum extent. In 2019, associated with the location of the Davis Strait Low (DSL) pressure pattern (Kwok et al., 2017) there is an average westward ice drift in both sectors in all months consistent with that seen in the mean 2012-2019 drift patterns (Figure 5a). The Pacific sector ice cover is composed of seasonal ice formed locally and fed by coastal polynyas and by outflows from the Ross Sea. Similarly, the Indian Ocean sector is seasonal ice from coastal polynyas and from the Pacific Sector.

The behavior of the freeboards in both sectors is similar (except for magnitude) (Figure 4). The higher IS-2 and CS-2 freeboards (though less pronounced in the CS-2 freeboards) in April/May are from a small population of sea ice adjacent to the coast (see Figure 3). Broadly, we find it difficult to explain the source of higher freeboard sea ice in both sectors early in the growth season. The behavior of higher freeboards of both the IS-2 and CS-2 freeboards are consistent - the squared correlation ( $\rho^2$ ) between them are 0.41 and 0.68, in the Pacific and Indian sectors, respectively. From a retrieval perspective, we also note that the heights of the local sea surface estimates near the ice edge are affected by sea state, likely due to scattering from the troughs of waves propagating into the ice cover. The consequence is surface heights that may be tens of centimeters below the local mean sea level resulting in higher freeboards. We have filtered most of these anomalous freeboards in the IS-2 and CS-2 processing some are still present.

In general, the behavior of the sea ice cover in the Pacific and Indian Ocean sectors resembles that of the E-Wedd sector, with the lowest end-of-season IS-2 and CS-2 freeboards. The thinner seasonal ice dominates the behavior of the mean freeboards in all months (Figure 3). The lowest CS-2 freeboards are found in the Indian Ocean sector in September (6.20±1.93 cm). The CS-2 freeboards remained within a narrow range after May, the lowering of the IS-2 freeboards over the winter months suggest a balance of different processes discussed above. Again, it is difficult to resolve these processes at the regional scale that the data is being examined in this paper.

## 4    Snow depth estimates

In this section, we first briefly summarize the calculation of snow depth from freeboard differences, and the sensitivity of the retrieved snow depths to uncertainties in bulk density. Second, we discuss the procedure used to construct monthly composites with freeboards from the two altimeter platforms, and the expected uncertainties from the lack of coincidence between the two measurements. Third, the 2019 spatial patterns of snow depths are examined. Last, we discuss the large-scale relationship between snow depth and total freeboard in the monthly composites.

### 4.1    Snow depth from freeboard differences

We follow the procedure detailed in Kwok et al. (2020) (henceforth *K20*) using a layered geometry depicted in Figure 2. A layer of snow-ice, an important component of the Southern Ocean ice cover, is included and assumed to have the same bulk density as sea ice. In our simplification, the snow-ice layer is considered to be part of ice thickness ( $h_i$ ) and indistinguishable from sea ice insofar as mechanical loading or hydrostatic equilibrium is concerned; this is necessitated by our lack of knowledge





on how to effectively model the snow-ice formation process. The snow depth ( $h_{fs}$ ) can thus be expressed as the difference between the total freeboard ( $h_f$ ) as measured by IS-2, and sea ice freeboard ( $h_{fi}$ ):

$$h_{fs} = h_f^{IS2} - h_{fi}. \tag{2}$$

If the scattering from the snow-ice interface dominates the returns at $K_u$-band wavelengths (CS-2 altimeter), then snow depth ( $h_{fs}^{\Delta f}$ ) is given by,

$$h_{fs}^{\Delta f} = \frac{(h_f^{IS2} - h_{fi}^{CS2})}{\eta_s}. \tag{3}$$

This equation relates snow depth to the IS-2 and CS-2 freeboard differences (i.e., the observables) with one free parameter, $\eta_s$, which is dependent on the bulk snow density. $\eta_s$ is the refractive index at $K_u$-band, $\eta_s = c/c_s(\rho_s)$, (Ulaby et al., 1986), and $c$ is the speed of light in free space. Equation (3) accounts for the reduced propagation speed of the radar wave ($c_s$) in a snow layer with bulk density $\rho_s$. At temperatures below freezing, and at the selected electromagnetic wavelengths of IS-2 and CS-2, the lidar and radar returns can be assumed to be from the air-snow and the snow-ice interfaces thus providing observations of total and ice freeboards. The validity and shortcomings of this assumption and its implications are discussed in Section 6. A bulk density of 0.32 g/cm³ is used in all our calculations.

### 4.1.1 Sensitivity of snow depth and ice thickness to snow density

Similarly, following *K20*, we write the sensitivity of $h_{fs}^{\Delta f}$ to bulk density (for the parameterization of $\eta_s$ given above) as:

$$\frac{\partial h_{fs}^{\Delta f}}{\partial \rho_s} = -0.77(1+0.51\rho_s)^{-2.5}(h_f^{IS2} - h_{fi}^{CS2}). \tag{4}$$

which gives the fractional change in snow depth associated with a change in density as,

$$\frac{\Delta h_{fs}^{\Delta f}}{(h_f^{IS2} - h_{fi}^{CS2})} = -0.53\Delta\rho_s \quad for \quad \rho_s = 0.32 g/cm^3. \tag{5}$$

Relative to a nominal density of 0.32 g/cm³ and an uncertainty in density of ±0.07 g/cm³, the uncertainty in the snow depth is ~4% of the difference in freeboard. In effect, this represents ~1 cm uncertainty in snow depth for freeboard differences of 30 cm, suggesting that snow depth is relatively insensitive to uncertainties in the bulk density. The sign indicates that snow depth will be underestimated if the density is overestimated.

As well, the sensitivity of thickness calculations to uncertainties in bulk density in *K20* (for a fixed total freeboard) is written as,

$$\left.\frac{\partial h_i}{\partial \rho_s}\right|_{h_f} = (h_f^{IS2} - h_{fi}^{CS2})\frac{1 - 0.77\eta_s^{-5/3}(\rho_s - \rho_w)}{\eta_s(\rho_w - \rho_i)}. \tag{6}$$

And the fractional change in ice thickness associated with a change in density is,

$$\left.\frac{\Delta h_i}{(h_f^{IS2} - h_{fi}^{CS2})}\right|_{h_f} \sim 10.5\Delta\rho_s. \quad for \quad \rho_s = 0.32 g/cm^3. \tag{7}$$

Again, relative to a nominal density of 0.32 g/cm³ and an uncertainty in density of ±0.07 g/cm³, the uncertainty in thickness is ~70% of the freeboard difference. This uncertainty translates into ~0.2 m uncertainty in thickness for a 30-cm freeboard difference. An overestimation in density leads to an underestimation in snow depth (above) and an overestimation of ice





thickness since a larger fraction of freeboard is now assigned to the higher density ice instead of snow. The above values provide bounds on density-induced errors in snow depth and sea ice thickness estimates if a $\Delta \rho_s$ of $\pm 0.07$ g/cm$^3$ is indeed realistic for representing the Antarctic sea ice. The above provides guidance on the expected sensitivity to the one free parameter in our simple model to convert freeboard differences to snow depth.

### 4.1.2 Sampling of freeboards for snow depth calculations

The sampling of the IS-2 and CS-2 freeboards for snow depth calculations follows the procedure in *K20*. The Antarctic is more challenging (compared to the Arctic) because of the lower density of ground tracks covering the lower latitude ice cover. Differences are calculated using gridded (25-km spacing) daily fields of IS-2 and CS-2 freeboards. Gridded IS-2 freeboards are averages of the three strong IS-2 beams and thus provide a better sampling of the spatial mean (compared to CS-2 freeboards). Freeboard differences at each IS-2 grid cell are calculated using samples with time separations $|\Delta T|<10$ days, and CS-2 freeboards (weighted by ice concentration) in neighboring grids cells that are within a 75-km box. We find that this sampling strategy provides the best spatial coverage without sacrificing precision.

We examined the sensitivity to space-time sampling (as in *K20*), by assessing differences in calculated snow depths with time separations of $|\Delta T| <1$ day, $<10$ days and $<15$ days, and using CS-2 freeboards at the collocated grid cell only and then including the eight neighboring grids cells (i.e., within a 75-km box); this gives us six space-time combinations. For the six combinations, the standard deviations of the differences in calculated snow depths were all less than 1 cm, which suggests that the spatial variability of the CS-2 freeboards is lower than IS-2 freeboards. Indeed, the variability of the area-averaged IS-2 freeboard between April and November (18.9 to 34.0 cm) is more than double the range of the CS-2 freeboards (8.5 to 10.3 cm).

The advantage of longer time separations and looking over longer distances for CS-2 freeboards is the added coverage for constructing full composites. In fact, increasing the time-separation tolerance to 10 days provides the best gain in coverage of more than 40% (Table 1).

### 4.1.3 Ice deformation

The episodic and localized nature of ice deformation and the impact of this process on differencing freeboards separated in time are discussed in *K20*, which we briefly summarize. The time order of freeboard sampling has an asymmetric effect. If the CS-2 freeboards were sampled earlier than the IS-2 freeboards, a convergence (divergence) event that occurred between samples would cause the snow depth to be overestimated (underestimated). Conversely, if the CS-2 freeboards were sampled after the IS-2 freeboards and a convergence (divergence) event occurred between sampling, the snow depths would be underestimated (overestimated). Here, the CS-2 samples are centered on the time of the IS-2 samples; hence, random events around that center time would increase the snow depth variance but have a small impact on the average monthly snow depth. The above analysis shows, however, for all of the six combinations of space-time sampling (discussed in the previous section), the standard deviations in retrieved snow depths were less than a centimeter. These results suggest that the effect of sea-ice dynamics in biasing the results may be small.

### 4.2 Snow depth estimates in 2019

The monthly snow depth composites and their distributions are shown in Figure 3 and Figure 6a, respectively. Table 2 shows the numerical values. Due to the low variability of the CS-2 freeboards, the spatial pattern of the snow depth estimates and the IS-2 freeboards are highly correlated in all the sectors ( $\rho > 0.96$ - see Figure 7). Here, we summarize the spatial features of note. A more in-depth discussion of the relationship between snow depth and freeboard can be found in the next section and an





assessment of the quality of the snow depth estimates (whether they are biased) are given the following section and Section 5, where these estimates were used to calculate ice thickness.

The thickest snow is seen in the W-Wedd sector (sector mean = 24.0±12.6 cm in May) and the CoA-B sectors (29.4±16.5 cm in November). With the multiyear sea ice cover in the W-Wedd sector, thicker snow is expected. The thinnest snow is found

in the Ross (7.65±4.54 cm in April) and E-Wedd (8.10±5.20 cm in November) sectors. The thinner snow depth in the Ross sector is likely due to the extensive coverage by thin/young ice exported from the active Ross Sea polynyas, and in the E-Wedd sector due to the large seasonal ice cover. The spatial patterns show consistent thinning of the snow cover towards the ice margins almost everywhere and in all months; we see no spatial anomalies in snow depth near the ice edge expected of higher precipitation. Except for coastal zones with active polynyas (e.g., southern Ross and Weddell Seas), snow depth is generally

higher in coastal zones.

Seasonal increases in the monthly mean snow depth are seen only in the A-B and CoA-B sectors. In the CoA-B sector, the increase is ~10 cm (approximately half that of the IS-2 freeboard increase) over the eight months. This is likely due to precipitation delivered by the on-shore wind pattern linked to the location and depth of the Amundsen Sea Low (ASL) discussed earlier. In all other sectors, we find either a decrease or a relatively unchanging snow cover between April and November,

similar to the observed behavior of IS-2 and CS-2 freeboards. This is quite remarkable and suggests the processes that remove snow from the surface (e.g., snow-ice transformation, loss into leads, divergence, etc.) must be significant and overwhelm all precipitation signals in all months. Consequently, an in-depth understanding of these processes will be important for understanding of the behavior of the Antarctic snow cover.

### 4.3 Relationship between freeboard and snow depth

*K20* examined this relationship for the Arctic ice cover. The large-scale relationship between snow depth and total freeboard is of geophysical interest as it provides a broad connection between the two parameters, and could be potentially used to provide rough estimates of snow depths where there are gaps in CS-2 observation. Figure 7 shows the monthly scatterplots of $h_{fs}^{\Delta f}$ and IS-2 total freeboard of the Antarctic for the eight months between April and November. At the length scale here (25 km), the regression analysis (regression slope, intercept, and standard error in each plot) of the monthly fields shows that the two

values are highly correlated (with the freeboard explaining >90% of the variance in snow depth); this is not entirely surprising as snow depth is derived from IS-2 freeboard. The regression slopes vary between 0.62 and 0.73 between April and November, and tell us the average fraction of the total freeboard that is composed of snow. For this Antarctic winter at least, the results suggest that more than 60-70% of the total freeboard is snow. This can be contrasted with the 2019 Arctic winter where snow occupies a lower fraction or ~50-55% of the total freeboard.

The negative intercepts of between -2.4 and -3.9 cm are worth noting, as one should expect zero snow depth at near zero total freeboard. The consistent values of the monthly intercepts suggest that one of the estimates may be biased. Here, we write:

$$\hat{h}_{fs} = \alpha h_f + \beta = f(h_f) \tag{8}$$

where $\hat{h}_{fs}$ is the snow depth estimate, and $\alpha$ and $\beta$ are the regression slope and intercept. If zero snow depth is expected at zero total freeboard, then an unbiased estimate of snow depth ( $h_{fs}$ ) can be written as,

$$h_{fs} = \hat{h}_{fs} + \delta = f(h_f) + \delta \quad and \quad \delta = -\beta \quad if \quad h_{fs} = f(0) = 0 \tag{9}$$

where $\delta$ is the bias. To obtain the true unbiased estimate of snow depth ( $h_{fs}$ ), an adjustment of $\hat{h}_{fs}$ by $\delta$ (or $-\beta$) is needed.

The negative intercepts observed in the scatterplots imply that $\hat{h}_{fs}$ is overestimated by +2.4 and +3.9 cm.



One likely source of these biases is the displacement of retracking point (RP) of the radar altimeter (CS-2) away from the snow-ice interface resulting in higher CS-freeboards. At $K_u$-band frequencies (CS-2), when the salinity of the snow near the snow-ice interface on seasonal ice is elevated due to brine-wicking, flooding, or when the temperature of the snow layer is above -5°C, the radar returns are displaced from the true ice surface. For Antarctic sea ice, in particular, the salinity of snow layer was characterized by Massom et al. (1997) to include two components: a "background" salinity of <1 psu in the upper part of the snow column, likely contributed by blowing snow due to wicked salt or aerosol or sea spray transported during strong winds over adjacent leads and polynyas, and a high-salinity (> 10 psu) basal component (0- 3 cm), sometimes damp due to brine wicking when the snow is thin or associated with flooding of the snow-interface. It is the basal layer salinity that has a large impact on CS-2 freeboards. Massom et al. (1997) also noted that basal salinities exceeding 10 psu commonly occur under relatively thin snow covers brine is available at their surface for vertical uptake into an accumulating snow layer.

The displacement of the RP's above the snow-ice interface from radar penetration experiments in the field has been reported in a number of publications (Willatt et al., 2010; Willatt et al., 2011). In fact, based on scattering simulations using salinity profiles from snow pits (collected in the Canadian Arctic Archipelago), Nandan et al. (2017) and Nandan et al. (2020) prescribed a nominal adjustment ($\delta$) of ~7 cm for first-year ice throughout most of the year. Kwok and Kacimi (2018), in an analysis of data from CS-2 and OIB, also reported consistently higher CS-2 radar freeboards along an airborne transect of the Weddell Sea.

K20 showed that an adjustment of the snow depth ($\delta$), due to the displacement of the scattering surface, would decrease the ice thickness estimates by

$$\Delta h_i = \left( \frac{\rho_s - \rho_w}{\rho_w - \rho_i} \right) \frac{\delta}{\eta_s} \sim -5.26\delta \quad for \, \rho_s = 0.32 kg \cdot m^{-3}. \tag{10}$$

A 7 cm adjustment results in a reduction in the estimated ice thickness of -0.37 m. The physical basis of a displacement of the RP due to brine wicking is sound, but a better understanding of the time-evolution of these processes and the magnitude of this adjustment need to be carried out if these corrections were to be applied to individual freeboard estimates. This will be addressed in more detail in the discussion of thickness calculations in the next section.

## 5 Ice thickness and volume

In this section, we first describe the calculation of ice thickness and volume by using snow depths from freeboard differences, and by assuming that the snow depth is equal to the total (or IS-2) freeboard. Second, we discuss briefly the spatial statistics of the composites and address the potential biases due to effects of the snow layer on CS-2 freeboard retrievals. Last, the volume of the Antarctic ice cover is discussed.

### 5.1 Ice thickness and sector volume

We calculate two ice thicknesses: 1) $h_i$ – with snow depth from altimeter freeboards and 2) $h_i^0$ – by setting snow depth equal to total freeboard, viz.

$$h_i(h_f, h_{fs}) = \left( \frac{\rho_w}{\rho_w - \rho_i} \right) h_f + \left( \frac{\rho_s - \rho_w}{\rho_w - \rho_i} \right) h_{fs} \tag{11}$$

$$h_i^0(h_f) = \left( \frac{\rho_w}{\rho_w - \rho_i} \right) h_f \quad for \quad h_{fs} = h_f. \tag{12}$$



In the first equation, we assume that the radar derived surface is from the snow-ice interface. The ice thickness, $h_i^0$, in the second equation sets a lower bound on the thickness estimates for a given total freeboard of $h_f$ with assumed densities of water, snow, and ice ($\rho_w = 1024 \, kg \cdot m^{-3}$, $\rho_s = 320 \, kg \cdot m^{-3}$, $\rho_i = 917 \, kg \cdot m^{-3}$). When flooding and snow-ice formation occur and the ice freeboard is zero, a measurement of snow depth can be used to estimate the ice thickness where snow-ice is predicted to form (of

5 course, given reasonable values for snow and ice densities).

Ice volume for each sector is simply the product of the average thickness $\bar{h}_i$ and area $A_{sec}$ of each sector,

$$V_{sec} = A_{sec} \bar{h}_i. \tag{13}$$

To examine the potential impact on ice volume due to biases in CS-2 freeboards due to salinity effects, we write,

$$V_{sec}(\delta) = A_{sec}(\bar{h}_i - 5.26\delta) \quad m^3 \tag{14}$$

where $\delta$ is the adjustment factor that accounts for the displacement of the CS-freeboard above the snow-ice interface discussed in Section 4.3.

### 5.2 Monthly ice thickness (April-November)

The monthly thickness composites ($h_i$ and $h_i^0$) and their distributions are shown in Figure 8 and Figure 9, respectively, and the numerical averages are in Table 2. Again, the spatial patterns of the thickness composites are very similar to that of the

15 freeboards and snow depth, and so here we note only the features and differences.

As expected, the thickest ice is found in the W-Wedd sector (mean 2.40±1.00 m) in May) and the CoA-B sectors (2.94±1.56 m in November). These are also sectors where the highest snow depths are found. The thinnest ice is in the Ross (0.87±0.40 m in April) and E-Wedd (<1.4 m for all months) sectors. The tongue of lower ice thickness in the Ross sector (Figure 8) is a clear signature of the outflow of thin/young ice produced in the Ross Sea polynyas. Similarly, for the E-Wedd sector, the

20 large expanse of thinner seasonal ice is also evident. Consistent thinning towards the ice margins is seen almost everywhere and in all months.

The seasonal cycle of ice thickness is surprisingly weak. Seasonal increases in the monthly mean ice thickness are only evident in the A-B and CoA-B sectors. Notably, in the CoA-B sector, the increase in ~1 m (from 1.82±1.11 m in April to 2.94±1.56 m in November) over the eight months, discussed earlier, is connected to coastal ice convergence (the mechanical

redistribution of thin to thicker ice) associated with persistent on-shore wind pattern in 2019. In all other sectors, we find either decreases or relatively unchanging thicknesses (i.e., weak seasonality) between April and November.

There are no seasonally and regionally diverse data set from field records that could be used the assess the large-scale satellite retrievals. Field observations of ice thickness are from two main sources – shipborne observations and mechanical drilling profiles. The most extensive compilation of Antarctic ice thickness is from the ASPeCt database reported in Worby et al.

(2008) – it contains data from 83 voyages and 2 helicopter flights for the period 1980 - 2005. Figure 10 compares our thickness estimates with the ASPeCt data summarized in Worby et al. (2008). The overall ice thickness in the ASPeCt data (circles in Figure 10) in all seasons and locations are less than half the mean thickness in the present data (solid blue line). There are two reasons these data sets are not comparable: 1) the ASPeCt data include very thin and level ice types; and, 2) few of the ASPeCt data have been collected at a similar time and location; indeed, no ASPeCt observations exist for the coastal southern

Bellingshausen and Amundsen Seas in spring. Underway shipboard observations made while traversing the pack ice (in ASPeCt database) favor sampling the thinner end of the thickness distribution due to physical, navigational, and logistical constraints. Hence, the sample population in the ASPeCt database is not likely to represent regional statistics needed for assessment of the





satellite estimates. Drilling data may be more comparable, as they provide a direct measure of thickness distribution and samples only ice thick enough to stand on—which places a similar cutoff imposed here by the exclusion of very thin ice. However, almost all drilling data to date are from thinner floes (Ozsoy-Cicek et al., 2013) and the thickest ice is often avoided. Even though drilling measurements have provided locations on where one should expect thicker ice (e.g., Lange & Eicken, 1991; Massom et al., 2001; Williams et al., 2015), they rarely provide averages at a spatial scales compatible with satellite averages.

Ice thickness estimates from Operation IceBridge provide averages at a larger scale but they are still limited in terms of seasonal coverage. In an examination of three years of OIB ice thickness, Kwok and Kacimi (2018) report October ice thicknesses that ranges from 2.4 to 2.6 m over a transect across the Weddell Sea (from the tip of the Antarctic Peninsula to Cap Norvegia. This is more compatible with the averages in the W-Wedd sector in Figure 10a (solid blue line). In a north-south OIB transect of the Ross Sea in November, Tian et al. (2020) found ice thicknesses between 0.48 and 0.99 m, again more compatible with that seen in Figure 10d (solid blue line). In any case, a more exhaustive evaluation of the present data set remains a challenge.

### 5.3 Are the thickness estimates high?

In sectors where there is predominantly seasonal ice (Ross, Pacific, Indian, E-Wedd) the thickness of ice in the early winter months of April and May, at close to ~ 1 m, seems to be too high. In these sectors, the growth of 1 m of sea ice in the 1-2 months between freeze-up (in February, March) and April/May is unlikely With ice drift that is largely seaward and divergent during these months (Figure 5), the only two processes that contribute significantly to increases in thickness are basal growth and snow-ice formation. In the short 1-2 months from freeze-up, basal thermodynamic growth of 1 m is unlikely given the oceanic conditions (ocean heat flux in a weakly stratified ocean compared to the Arctic). As well, it would require high snowfall rates to create a significant thickness of snow-ice in that amount of time. Thus, this points strongly to biases in the CS-2 freeboards as the estimated thicknesses are highly sensitive to biases in CS-2 freeboards (due to large 3:1 contrast between ice and snow densities in Equation 11).

Clearly, if ice freeboard were zero everywhere, then $h_i^0$ (Equation 12) would be the best estimate of ice thickness given measurements of total freeboard. However, this is unlikely the case especially in the W-Wedd and CoA-B sectors where thicker ice is known to be present (see discussion above). If there were a large-scale bias in the CS-2 freeboards (assuming the processes that contribute to the radar biases are the same everywhere) then areas with the lowest CS-2 freeboards provide a rough guidance on the magnitude of that bias. In the four sectors of largely seasonal ice (Ross, Pacific, Indian, E-Wedd), the sector-averaged CS-2 freeboards have the lowest CS-2 freeboards and low seasonal variability that ranges from 5.41±2.40 cm (minimum) to 9.47±3.87 cm for all months. This suggests a bias ($\delta$) of ~5 cm if we assumed that early-season ice freeboards have to be near zero. This value can be compared to reported biases from different studies, for example:

- The thickness of the high salinity basal layer of 0-3 cm (> 10 psu) reported by Massom et al. (1997).
- Suggested adjustment ($\delta$) of ~7 cm on first-year ice in the Arctic based on a scattering study using profiles of basal salinities (Nandan et al. (2017); Nandan et al. (2020)).
- Observed CS-2 biases of up to 8 cm in the Weddell Sea in an assessment of the IceBridge and CS-2 derived ice thicknesses (Kwok & Kacimi, 2018).
- 2.4 - 3.9 cm estimated based in Section 4.3.

In the following sector, we examine the sensitivity of thickness and volume if these biases were generally representative over the entire ice cover.





### 5.4 Thickness and volume estimates – with and without adjustments

As discussed above, sea ice would be too thick using the CS-2 freeboards directly and too thin if ice freeboard were assumed to be zero everywhere. Guided by the potential range of CS-2 freeboard biases above, we calculate the regional thickness and volume of the Antarctic ice cover with adjustments ($\delta$) of 3 and 6 cm (Equation 13) to assess the variability of sector ice volume between the two extremes of thicknesses (i.e., $h_i$ and $h_i^0$) over the winter of 2019. The monthly $h_i^0$ composites and the sector thicknesses (with $\delta = 0$, 3, 6 cm) can be seen in Figure 10 and Table 3, and the monthly ice volumes are shown in Figure 11.

The adjustments to CS-2 freeboards, as expected, lower the thickness (5 cm per 1 cm of adjustment – based on Equation 10); at $\delta = 6$ cm the sector mean would be reduced by 0.32 m. The impact is higher – in terms of fractional change in total thickness – in sectors with thinner ice (e.g., E-Wedd). The range of thicknesses in Figure 10 gives us at least an indication of a potential range of variability between assuming zero ice freeboard and the rough estimates of $\delta$ (applied as a sector wide bias). Even though current knowledge does not allow us to adjust for individual thickness retrievals, these large scale adjustments likely provide a better estimate than those calculated using $h_i$ or $h_i^0$.

The end-of-season ice volume in each sector is proportional to the area production (Figure 11h) with the largest ice volume in the E-Wedd sector. This, of course, is not the ice volume production in a particular sector. In order to calculate seasonal ice production, one has to account for volume exchanges at the sector boundaries and volume lost to melt at the ice edge. Of interest here is the ice volume and its sensitivity to $\delta$. At the end of the season, the difference in total Antarctic ice volume between assuming $\delta = 0$ and $h_{fs} = h_f$ is ~6000 km³, or one-third of the total volume. Adjustments with $\delta = 3$ and $\delta = 6$ cm reduce the differences by 1200 and 2400 km³, respectively. As with ice thickness, in sectors where the ice is thicker (W-Wedd, Figure 10) the fractional changes are smaller. An adjustment of 6 cm gives a circumpolar ice volume of 12,500 km³ in October, for an average thickness of ~0.93 m.

These volume estimates can be compared to volume estimates from ICESat-1 freeboards. Using AMSR snow depths, Zwally et al. (2008) estimated the average October-November (2004 and 2005) Weddell Sea ice volume to be ~8750 km³, higher than our 2019 estimate of 5731 km³ (without any adjustments). Here, differences are expected as the efficacy of the AMSR snow depths has not been demonstrated.

Assuming snow depth to be the total freeboard (i.e., zero ice freeboard) Kurtz and Markus (2012) estimated an average circumpolar ice volume of 11,111 km³ in the spring (between 2003 and 2008) with an average thickness of 0.83 m; this can be compared to our October estimate of 8260 km³ and 0.61 m with the same assumption. Certainly, our lower volume estimates are partly attributable to the retreat in Antarctic ice coverage (Parkinson, 2019) since the ICESat-1 mission of >10⁶ km². But, with the same assumption of zero ice freeboard, the change of 0.2 m between ICESat-1 and IS-2 in 2019 may be of interest but this is more of an indication of total freeboard changes rather actual change in ice thickness.

## 6    Conclusions

In this paper, we offer a view of the Antarctic sea ice cover from lidar (ICESat-2) and radar (CryoSat-2) altimetry. This is a first joint examination of the IS-2 and CS-2 freeboards, the snow depth derived from their differences, and the calculated sea ice thickness/volume. Our analysis spans an 8-month winter between April, 2019 and November 16, 2019. We characterize the behavior of the circumpolar ice cover in seven geographic sectors. The limitations in our current knowledge in the retrieval of snow depth, thickness, and volume are addressed. Below we highlight some of the results and discuss future opportunities for validation and assessment of this retrieval approach:





- Highest freeboards are seen in the CoA-B and W-Wedd sectors. The remarkable ice convergence due to on-shore wind and ice drift along the coastal Amundsen Sea – associated with the depth, location, and persistence of Amundsen Sea Low pattern – is captured in the correlated changes in IS-2 and CS-2 freeboards with extremes of 48.2±26.2 cm and 17.8±11.8 cm, respectively, and derived thickness of 2.94±1.56 m. The multiyear ice in the W-Wedd sector, as expected, also stands out with high freeboards and thickness (sector mean thickness of 2.40±1.00 m).

- Lowest freeboards, snow depth, and thickness are seen in the proximity of the Ross Sea polynyas and Ronne Polynya. In the Ross Sea sector, the lowest sector-averaged IS-2 and CS-2 freeboards of 13.8±6.45 cm, and 5.55±3.06 cm, respectively, can be contrasted with those in the CoA-B and W-Wedd above.

- With the extremely low variability in CS-2 freeboards in the Antarctic snow depth estimates are highly correlated with IS-2 freeboards, with the IS-2 freeboard explaining >90% of the variance in snow depth. Our results suggest that more than 60-70% of the total freeboard is snow.

- In 2019, the observed seasonality in the sector-averaged freeboards, snow depth, and thickness is surprisingly weak. These sector averages do not follow the expected seasonal increases due to ice growth and snow accumulation seen in the Arctic. We attribute this to the mixture of competing processes (snowfall, snow redistribution, snow-ice formation, ice deformation, basal growth/melt) in different parts of the divergent Antarctic ice cover, and the continuous export of sea ice to the margins, where they subsequently melt.

- Evidence points to biases in CS-2 freeboards that is associated with displacement of the retracking points to a height above the snow-ice interface resulting in snow depths that are too low and ice thicknesses that are too high in the present retrievals. Based on field measurements, a contributing source to the bias is the salinity at the base of the snow layer due to wicking and flooding, the physical basis of expected biases in CS-2 freeboards from basal-layer salinity the is sound. The question is the range of the biases and whether a correction factor could be applied for retrievals at the highest spatial resolution.

- Our calculations show the sector-scale variability of snow depth, thickness and computed ice volume given biases of 3 cm and 6 cm in radar freeboard, and assuming zero ice freeboard. At the sector scale, the adjusted estimates seem to be more credible although better assessment of these parameter awaits better field measurements. An adjustment of 3/6 cm gives a circumpolar ice volume of 14,700/12,500 km$^3$ in October, for an average thickness of ~1.09/0.93 m.

- Validation of Antarctic sea ice parameters remain a challenge. There are no seasonally and regionally diverse data set from field records that could be used to the assess the large-scale satellite retrievals, especially in areas that are inaccessible to ships. The overall ice thickness in the ASPeCt data in all seasons and locations are less than half the mean thickness in the present data and points to the sampling biases from underway shipboard observations

The present analysis, however, is only a first step in the examination of the Antarctic ice cover using both the IS-2 and CS-2 altimeter. There are many aspects of data quality, some of which will only be revealed by assessment with data acquired and processed by dedicated airborne campaigns (e.g., NASA's Operation IceBridge), field programs, and when a longer IS-2/CS-2 time series become available. As mentioned in *K20*, the adjustment of the CS-2 orbits to provide improved coincidence in space-time sampling of the surface is being considered. A Joint NASA-ESA working group is exploring this opportunity, while there is an overlap in the IS-2 and CS-2 missions, to 'tune' the CS-2 orbital parameters slightly to improve the time-separation between near co-incident IS-2 and CS-2 measurements (cross-overs and along-track sampling). If this is approved. the two altimeters will provide a crucial data set for not only understanding the current retrievals but also the design of future instruments tasked to understanding the development of the Antarctic sea ice cover.





**Acknowledgments**

We wish to thank the International Space Science Institute (ISSI) for supporting and hosting the useful workshops on Satellite Remote Sensing of Antarctic sea ice held in Bern, Switzerland over the past decade. The ICESat-2 ATL10 data set used herein are available at https://nsidc.org/data/icesat-2/data-sets. S.K. and R.K. carried out this work at the Jet Propulsion Laboratory, California Institute of Technology, under contract with the National Aeronautics and Space Administration.



**Table 1.** Dependence of number of retrievals on space-time separation. November is not included here because the IS-2 data (Release 002) covered only half a month.

| Space/ Time | 25-km/ 1 day | 25-km/ 10 day | 25-km/ 15 day | 75-km/ 1 day | 75-km/ 10 day | 75-km/ 15 day |
|---|---|---|---|---|---|---|
| Apr | 459 | 2538 | 2971 | 1698 | 4052 | 4180 |
| May | 619 | 4058 | 5063 | 2453 | 6673 | 7128 |
| Jun | 966 | 5397 | 6742 | 3383 | 8982 | 9677 |
| Jul | 898 | 5734 | 7226 | 3205 | 9729 | 10455 |
| Aug | 1340 | 8935 | 11298 | 5065 | 15467 | 16601 |
| Sep | 1112 | 6946 | 8897 | 4014 | 12102 | 13239 |
| Oct | 950 | 6086 | 7734 | 3447 | 10019 | 10962 |





**Table 2**. Monthly mean (standard deviation) of total freeboard $(h_f)$, CS-2 freeboard $(h_{fi}^{CS2})$, and derived snow depth $(h_s^{\Delta f})$.

| (cm) | | Apr | May | Jun | Jul | Aug | Sep | Oct | Nov |
|---|---|---|---|---|---|---|---|---|---|
| E-Wedd | $h_f$ | 25.4±10.0 | 19.0±9.72 | 15.6±8.12 | 15.6±8.21 | 17.5±6.02 | 16.6±5.93 | 15.7±5.82 | 14.7±7.25 |
| | $h_{fi}^{CS2}$ | 7.85±3.91 | 6.12±3.36 | 5.41±2.40 | 5.71±1.60 | 6.42±3.94 | 6.82±2.02 | 6.85±2.22 | 6.85±2.22 |
| | $h_{fs}^{\Delta f}$ | 16.0±9.00 | 13.5±9.82 | 8.40±5.58 | 8.34±5.97 | 9.34±4.25 | 8.70±4.00 | 8.75±3.96 | 8.10±5.20 |
| W-Wedd | $h_f$ | 36.5±20.3 | 41.1±19.2 | 36.2±16.9 | 37.3±20.8 | 38.7±19.7 | 33.8±18.2 | 30.8±15.2 | 30.8±16.8 |
| | $h_{fi}^{CS2}$ | 10.9±4.78 | 11.7±4.50 | 10.5±4.54 | 10.2±4.68 | 10.3±5.14 | 9.72±4.34 | 10.8±4.94 | 11.0±4.54 |
| | $h_{fs}^{\Delta f}$ | 21.4±13.4 | 24.0±12.6 | 21.7±12.9 | 22.6±15.0 | 23.2±14.2 | 21.0±13.5 | 17.7±10.0 | 18.0±12.0 |
| A-B | $h_f$ | 29.5±21.8 | 25.3±17.8 | 26.7±18.8 | 26.5±24.1 | 31.1±23.6 | 32.1±25.0 | 28.2±21.9 | 35.7±25.6 |
| | $h_{fi}^{CS2}$ | 10.0±6.02 | 9.01±5.84 | 9.01±6.84 | 9.53±7.75 | 10.5±8.07 | 11.4±9.37 | 10.6±8.33 | 10.4±7.12 |
| | $h_{fs}^{\Delta f}$ | 18.3±13.8 | 14.6±10.8 | 15.3±11.5 | 15.3±15.6 | 18.0±16.7 | 18.5±15.4 | 18.8±15.4 | 22.3±16.7 |
| CoA-B | $h_f$ | 29.7±19.2 | 29.2±16.6 | 33.6±19.3 | 41.5±26.4 | 46.3±26.2 | 45.5±30.2 | 42.7±24.5 | 48.2±26.2 |
| | $h_{fi}^{CS2}$ | 11.0±6.08 | 11.5±6.32 | 12.6±8.00 | 14.6±9.88 | 16.4±10.4 | 17.8±11.8 | 17.1±10.2 | 16.0±8.48 |
| | $h_{fs}^{\Delta f}$ | 18.0±12.0 | 16.6±10.2 | 18.4±10.5 | 22.7±16.6 | 26.6±18.0 | 24.0±17.7 | 26.1±16.2 | 29.4±16.5 |
| Ross | $h_f$ | 13.8±6.45 | 15.2±6.50 | 17.7±7.93 | 19.3±8.83 | 21.0±10.2 | 19.8±11.7 | 18.8±11.0 | 20.4±13.2 |
| | $h_{fi}^{CS2}$ | 5.55±3.06 | 6.22±2.70 | 6.65±2.52 | 7.54±3.61 | 7.98±3.50 | 8.34±3.90 | 7.98±4.10 | 8.30±5.10 |
| | $h_{fs}^{\Delta f}$ | 7.65±4.54 | 7.70±4.17 | 9.25±4.95 | 9.91±5.00 | 11.5±6.16 | 10.4±7.41 | 10.5±7.62 | 11.6±7.83 |
| Pacific | $h_f$ | 34.8±30.1 | 22.3±16.6 | 27.9±14.4 | 24.5±13.9 | 26.5± 6.5 | 22.9±15.4 | 23.2±20.1 | 21.1±14.5 |
| | $h_{fi}^{CS2}$ | 9.47±3.87 | 8.41±2.90 | 8.33±3.33 | 8.90±3.36 | 7.78±2.78 | 8.42±3.62 | 8.40±3.63 | 8.02±2.97 |
| | $h_{fs}^{\Delta f}$ | 23.7±23.7 | 17.8±15.4 | 18.5±12.0 | 14.6±8.63 | 16.2±9.97 | 14.6±10.5 | 17.0±16.8 | 15.6±9.97 |
| Indian | $h_f$ | 27.5± 22.4 | 19.0±14.1 | 25.3±26.7 | 17.8±7.77 | 17.9±8.55 | 14.6±7.73 | 16.0±8.32 | 16.7±10.0 |
| | $h_{fi}^{CS2}$ | 9.14±3.68 | 7.68±3.12 | 7.05±2.83 | 6.60±2.83 | 6.40±2.33 | 6.20±1.93 | 6.32±2.21 | 6.45±2.44 |
| | $h_{fs}^{\Delta f}$ | 18.4±18.4 | 13.7±12.3 | 16.4±19.7 | 10.0±7.20 | 10.0±5.64 | 8.30±5.66 | 8.88±5.43 | 10.1±6.73 |





**Table 3.** Monthly mean (standard deviation) of estimated ice thickness: 1) $h_i$, with derived snow depth ( $h_i$); 2); $h_i^0$, assuming $h_{fs} = h_f$; 3) $h_i^3$, with $\delta = 3$ cm; and, 4) $h_i^6$, with $\delta = 6$ cm.

| (m) | | Apr | May | Jun | Jul | Aug | Sep | Oct | Nov |
|---|---|---|---|---|---|---|---|---|---|
| E-Wedd | $h_i$ | 1.40±0.54 | 1.21±0.54 | 1.00±0.46 | 1.00±0.40 | 1.11±0.32 | 1.07±0.33 | 1.01±0.32 | 1.00±0.38 |
| | $h_i^0$ | 0.78±0.33 | 0.58±0.30 | 0.48±0.20 | 0.48±0.25 | 0.54±0.18 | 0.51±0.18 | 0.48±0.18 | 0.45±0.22 |
| | $h_i^3$ | 1.24 | 1.05 | 0.84 | 0.84 | 0.95 | 0.91 | 0.85 | 0.84 |
| | $h_i^6$ | 1.08 | 0.89 | 0.68 | 0.68 | 0.79 | 0.75 | 0.69 | 0.68 |
| W-Wedd | $h_i$ | 2.16±1.13 | 2.40±1.00 | 2.11±0.87 | 2.17±1.07 | 2.25±1.05 | 1.95±0.95 | 1.87±0.92 | 1.90±0.85 |
| | $h_i^0$ | 1.13±0.63 | 1.26±0.60 | 1.12±0.52 | 1.15±0.64 | 1.20±0.61 | 1.04±0.56 | 0.95±0.47 | 0.95±0.52 |
| | $h_i^3$ | 2 | 2.24 | 1.95 | 2.01 | 2.09 | 1.79 | 1.71 | 1.74 |
| | $h_i^6$ | 1.84 | 2.08 | 1.79 | 1.85 | 1.93 | 1.63 | 1.55 | 1.58 |
| A-B | $h_i$ | 1.83±1.24 | 1.58±1.00 | 1.70±1.17 | 1.75±1.47 | 1.95±1.36 | 2.06±1.57 | 1.80±1.30 | 2.31±1.50 |
| | $h_i^0$ | 0.91±0.67 | 0.77±0.55 | 0.82±0.58 | 0.81±0.74 | 0.95±0.73 | 0.99±0.77 | 0.87±0.67 | 1.10±0.79 |
| | $h_i^3$ | 1.67 | 1.42 | 1.54 | 1.59 | 1.79 | 1.9 | 1.64 | 2.15 |
| | $h_i^6$ | 1.51 | 1.26 | 1.38 | 1.43 | 1.63 | 1.74 | 1.48 | 1.99 |
| CoA-B | $h_i$ | 1.82±1.11 | 1.78±1.02 | 2.08±1.26 | 2.61±1.64 | 2.78±1.50 | 2.87±1.87 | 2.57±1.43 | 2.94±1.56 |
| | $h_i^0$ | 0.91±0.60 | 0.89±0.51 | 1.03±0.59 | 1.28±0.81 | 1.42±0.80 | 1.40±0.93 | 1.31±0.75 | 1.48±0.81 |
| | $h_i^3$ | 1.66 | 1.62 | 1.92 | 2.45 | 2.62 | 2.71 | 2.41 | 2.78 |
| | $h_i^6$ | 1.5 | 1.46 | 1.76 | 2.29 | 2.46 | 2.55 | 2.25 | 2.62 |
| Ross | $h_i$ | 0.87±0.40 | 0.98±0.40 | 1.13±0.48 | 1.26±0.57 | 1.35±0.61 | 1.30±0.66 | 1.22±0.66 | 1.37±0.82 |
| | $h_i^0$ | 0.42±0.20 | 0.47±0.20 | 0.55±0.24 | 0.59±0.27 | 0.65±0.31 | 0.61±0.36 | 0.58±0.34 | 0.63±0.41 |
| | $h_i^3$ | 0.71 | 0.82 | 0.97 | 1.1 | 1.19 | 1.14 | 1.06 | 1.21 |
| | $h_i^6$ | 0.55 | 0.66 | 0.81 | 0.94 | 1.03 | 0.98 | 0.9 | 1.05 |
| Pacific | $h_i$ | 2.00±1.55 | 1.33±0.88 | 1.62±0.80 | 1.52±0.73 | 1.54±0.90 | 1.33±0.81 | 1.34±1.00 | 1.35±0.70 |
| | $h_i^0$ | 1.07±0.93 | 0.68±0.51 | 0.86±0.44 | 0.75±0.43 | 0.82±0.51 | 0.71±0.50 | 0.71±0.62 | 0.65±0.45 |
| | $h_i^3$ | 1.84 | 1.17 | 1.46 | 1.36 | 1.38 | 1.17 | 1.18 | 1.19 |
| | $h_i^6$ | 1.68 | 1.01 | 1.3 | 1.2 | 1.22 | 1.01 | 1.02 | 1.03 |
| Indian | $h_i$ | 1.50±1.10 | 1.12±0.72 | 1.37±1.18 | 1.07±0.37 | 1.09±0.45 | 0.92±0.39 | 1.00±0.43 | 1.10±0.54 |
| | $h_i^0$ | 0.85±0.68 | 0.58 0.43 | 0.77±0.82 | 0.55±0.24 | 0.55±0.26 | 0.45±0.24 | 0.49±0.25 | 0.51±0.31 |
| | $h_i^3$ | 1.34 | 0.96 | 1.21 | 0.91 | 0.93 | 0.76 | 0.84 | 0.94 |
| | $h_i^6$ | 1.18 | 0.8 | 1.05 | 0.75 | 0.77 | 0.6 | 0.68 | 0.78 |
| Antarctic | $h_i$ | 1.55 | 1.37 | 1.37 | 1.32 | 1.39 | 1.28 | 1.25 | |
| | $h_i^0$ | 0.81 | 0.69 | 0.69 | 0.64 | 0.70 | 0.63 | 0.61 | |
| | $h_i^3$ | 1.39 | 1.21 | 1.21 | 1.16 | 1.23 | 1.12 | 1.09 | |
| | $h_i^6$ | 1.23 | 1.05 | 1.05 | 1.00 | 1.07 | 0.96 | 0.93 | |



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





**Figure Captions**

Figure 1. Coverage of the sectors of the circumpolar Antarctic sea ice cover.

Figure 2. Relationship between the different height quantities.

Figure 3. Monthly composites of IS-2 freeboard ( $h_f$ ), CS-2 freeboard ( $h_{fi}^{CS2}$ ), derived snow depth ( $h_{fs}^{\Delta f}$ ) for the period between

April 2019 and November 2019. (Units: centimeters)

Figure 4. Monthly distributions of (a) IS-2 ( $h_f$ ) and (b) CS-2 ( $h_{fi}^{CS2}$ ) freeboards for the period between April 2019 and

November 2019. Their monthly means are compared in (c). Numerical values in the line plots show the squared

correlation between the two freeboards.

Figure 5. Monthly mean (April through November) ice drift in the Southern Ocean for (a) 2012-2019 and (b) 2019.

Figure 6. Monthly distributions of (a) derived snow depth ( $h_{fs}^{\Delta f}$ ) and (b) ice thickness ( $h_i$ ) for the period between April 2019 and

November 2019.

Figure 7. Monthly relationship between snow depth and freeboard. Parameters from the regression analysis (slope, intercept,

correlation coefficient, and standard error) are shown in the top left corner of each panel.

Figure 8. Monthly composites of calculated ice thicknesses: (a) $h_i$ – using snow depth from freeboard differences ( $h_{fs}^{\Delta f}$ ), and (b)

$h_i^0$ – assuming zero ice freeboard, i.e., $h_{fs} = h_f$ , for the period between April 2019 and November 2019. (Units: meters)

Figure 9. Monthly distributions of calculated ice thicknesses: (a) $h_i$ – using snow depth from freeboard differences ( $h_{fs}^{\Delta f}$ ), and (b)

$h_i^0$ – assuming zero ice freeboard, i.e., $h_{fs} = h_f$ , for the period between April 2019 and November 2019. Their monthly

means are compared in (c).

Figure 10. Comparison of seasonal ice thickness calculated with $\delta$ = 0, 3, and 6 cm, and assuming zero ice freeboard (i.e.,

$h_{fs} = h_f$ ) with shipborne measurements in Worby et al. (2008).

Figure 11. Evolution of the volume and area of the Antarctic sea ice cover between April and October 2019.



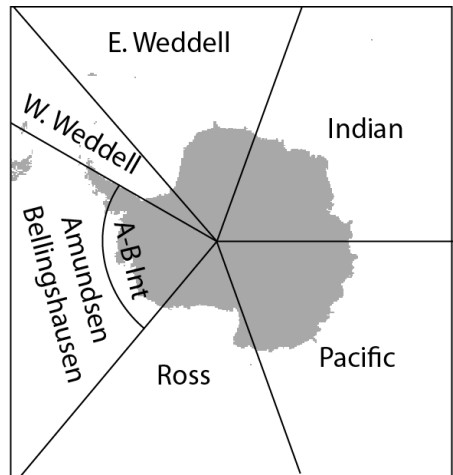

Figure 1. Coverage of the sectors of the circumpolar Antarctic sea ice cover.



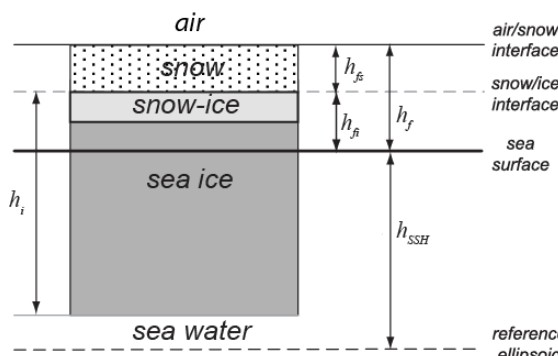

Figure 2. Relationship between the different height quantities in Equation (1).





Figure 3. Monthly composites of IS-2 freeboard ($h_f$), CS-2 freeboard ($h_{fi}^{CS2}$), derived snow depth ($h_{fs}^{\Delta f}$) for the period between April 2019 and November 2019. (Units: centimeters)



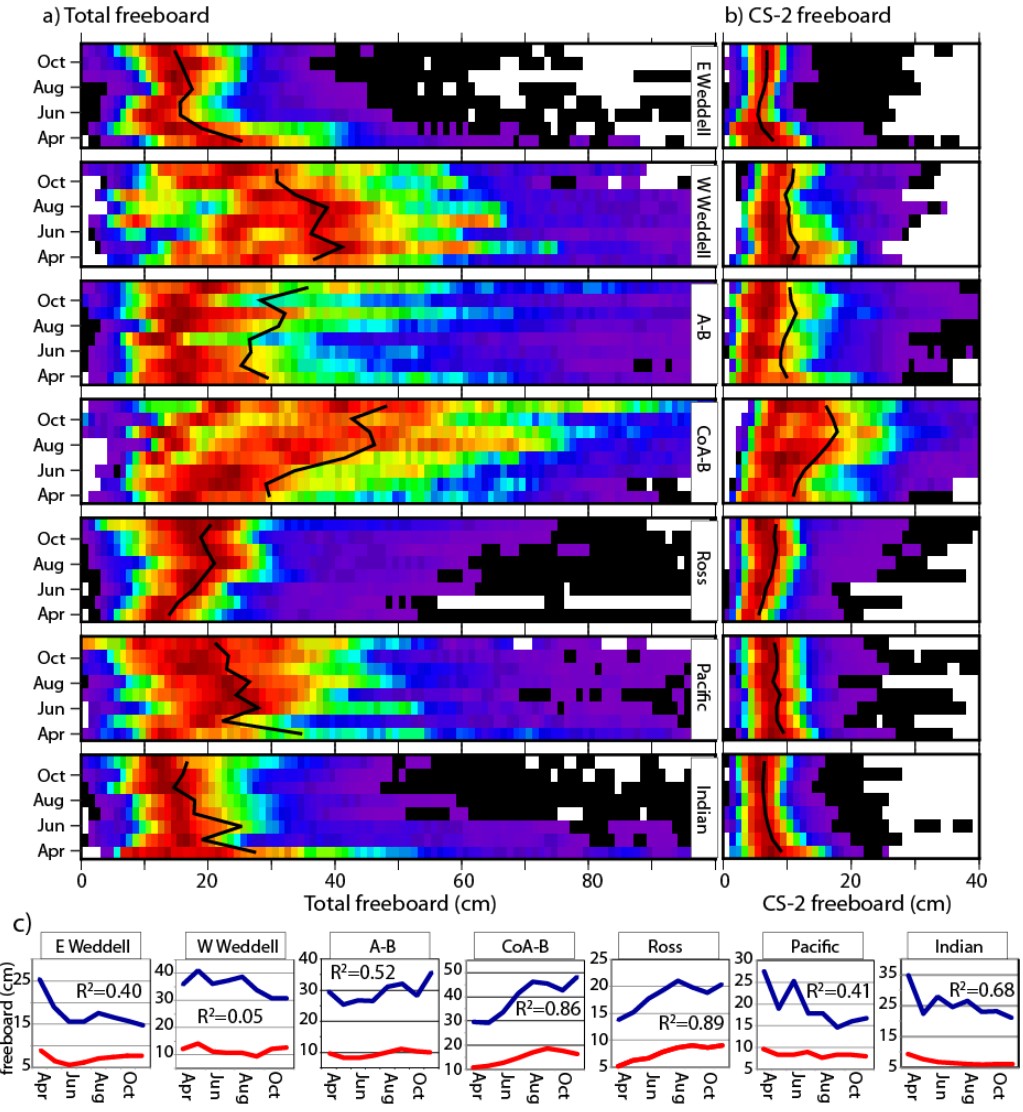

Figure 4. Monthly distributions of (a) IS-2 ( $h_f$ ) and (b) CS-2 ( $h_{fi}^{CS2}$ ) freeboards for the period between April 2019 and November 2019. Their monthly means are compared in (c). Numerical values in the line plots show the squared correlation between the two freeboards.





Figure 5. Monthly mean (April through November) ice drift in the Southern Ocean for (a) 2012-2019 and (b) 2019.





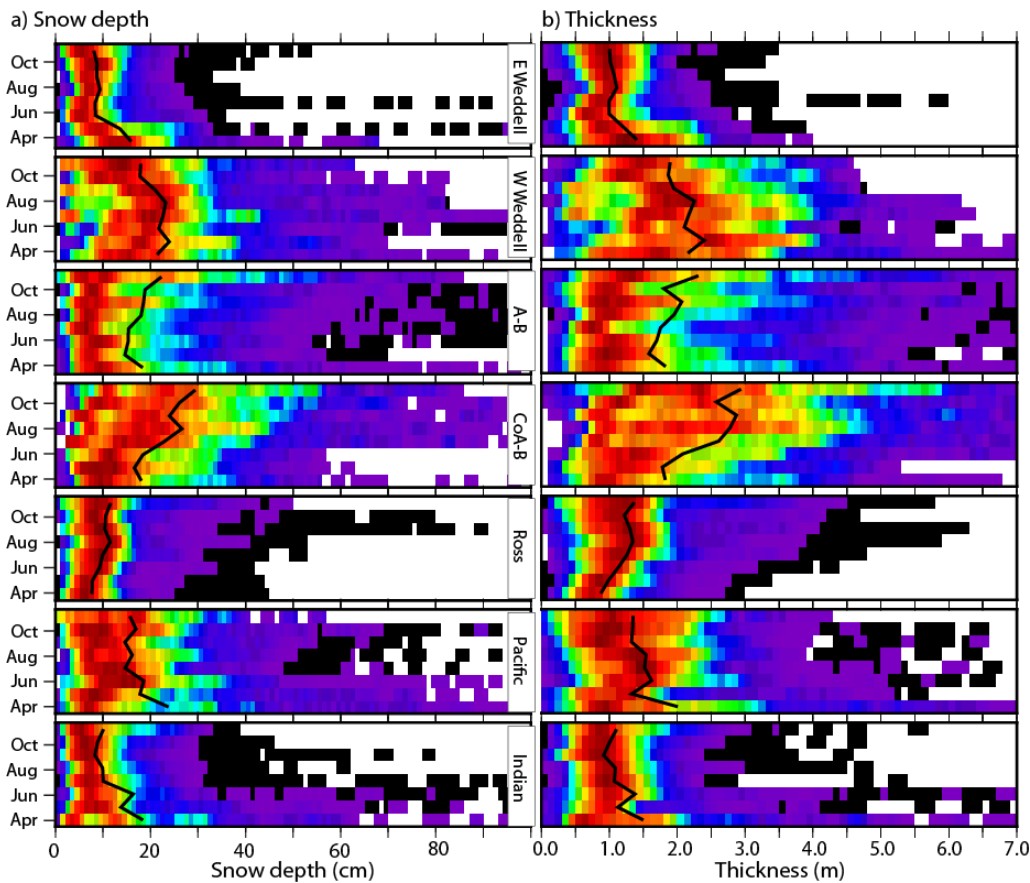

Figure 6. Monthly distributions of (a) derived snow depth ( $h_{fs}^{\Delta f}$ ) and (b) ice thickness ( $h_i$ )  for the period between April 2019 and November 2019.

.





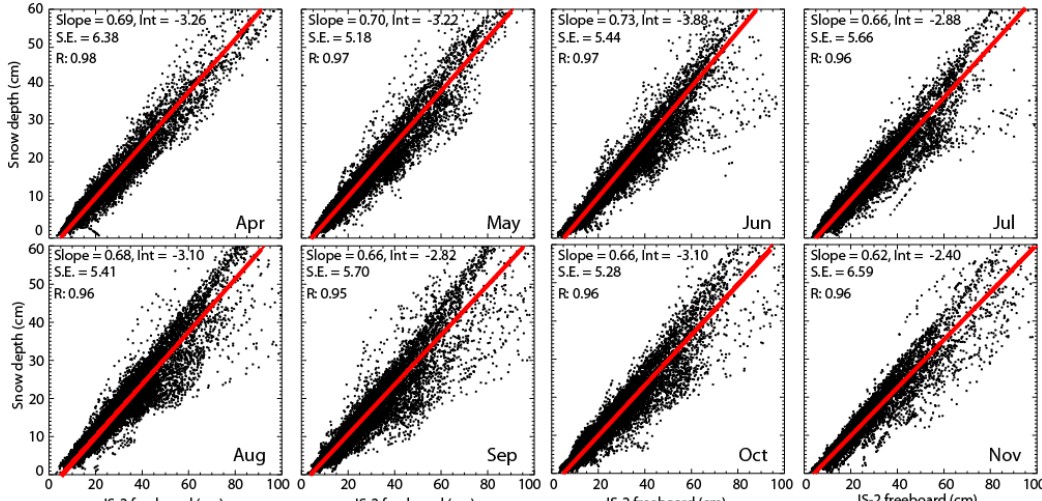

Figure 7. Monthly relationship between snow depth and freeboard. Parameters from the regression analysis (slope, intercept, correlation coefficient, and standard error) are shown in the top left corner of each panel.


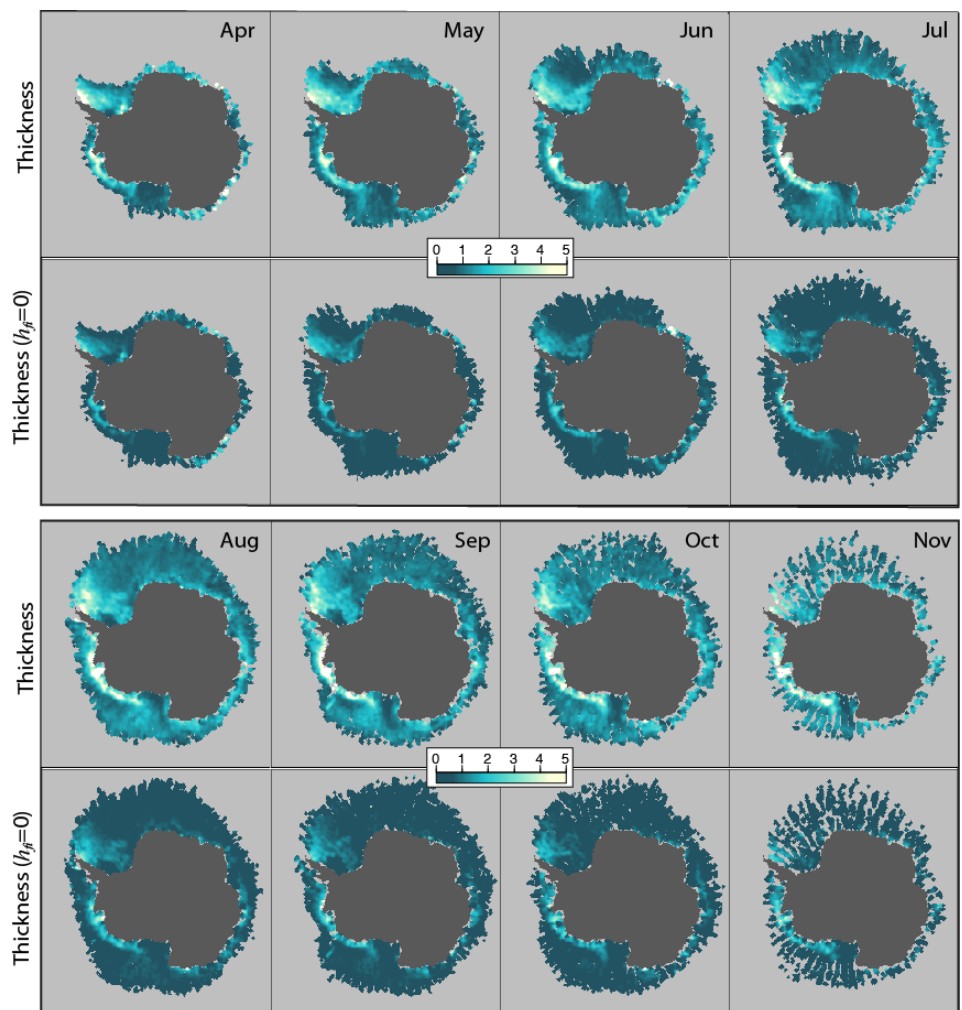

Figure 8. Monthly composites of calculated ice thicknesses: (a) $h_i$ – using snow depth from freeboard differences ( $h_{fs}^{\Delta f}$ ), and (b) $h_i^0$ – assuming zero ice freeboard, i.e., $h_{fs} = h_f$ , for the period between April 2019 and November 2019. (Units: meters)





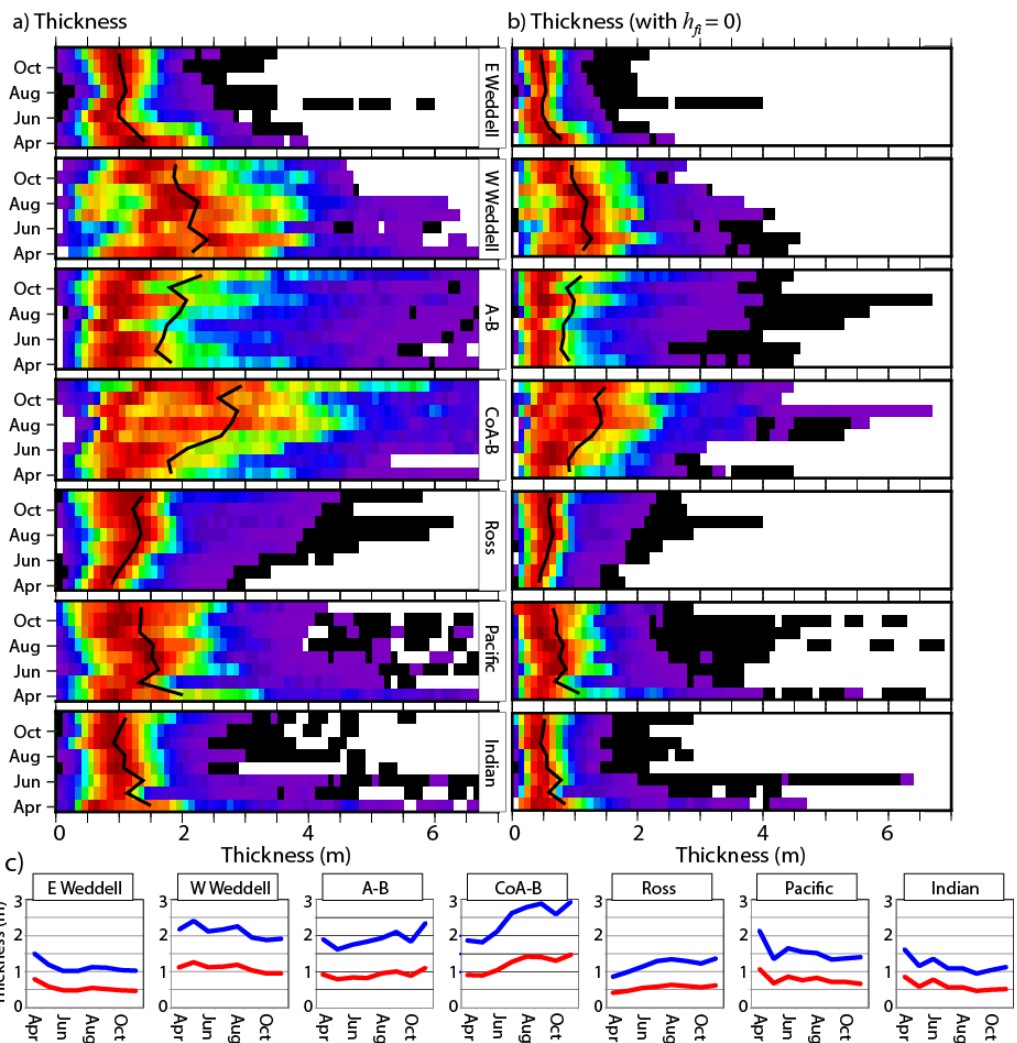

Figure 9. Monthly distributions of calculated ice thicknesses: (a) $h_i$ – using snow depth from freeboard differences ($h_{fs}^{\Delta f}$), and (b) $h_i^0$ – assuming zero ice freeboard, i.e., $h_{fs} = h_f$, for the period between April 2019 and November 2019. Their monthly means are compared in (c).





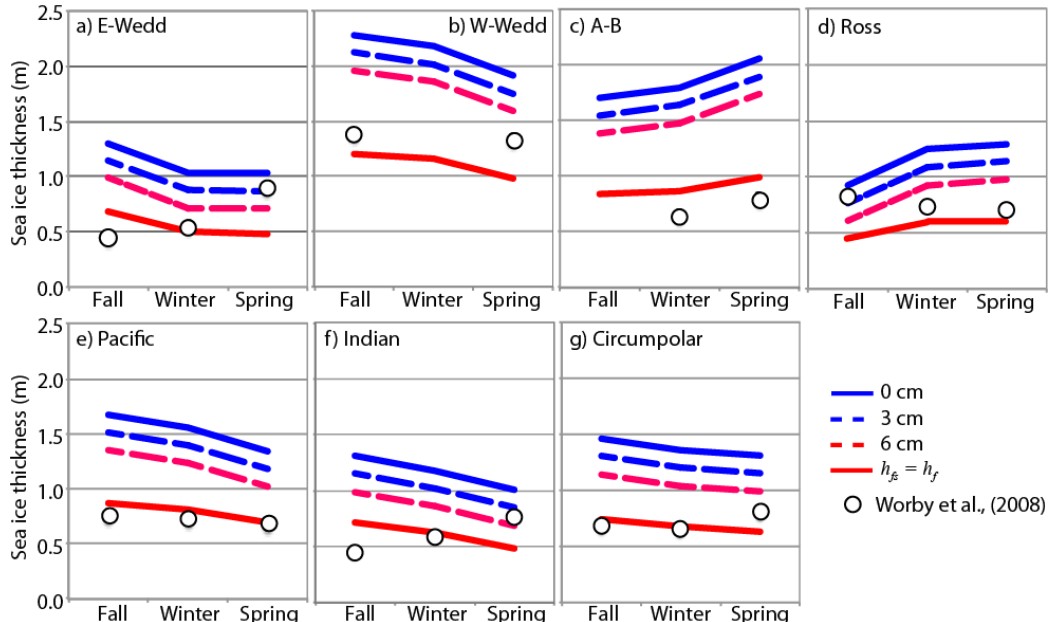

Figure 10. Comparison of ice thicknesses calculated with $\delta$ = 0, 3, and 6 cm, and assuming zero ice freeboard (i.e., $h_{fs} = h_f$) with shipborne measurements in Worby et al. (2008).





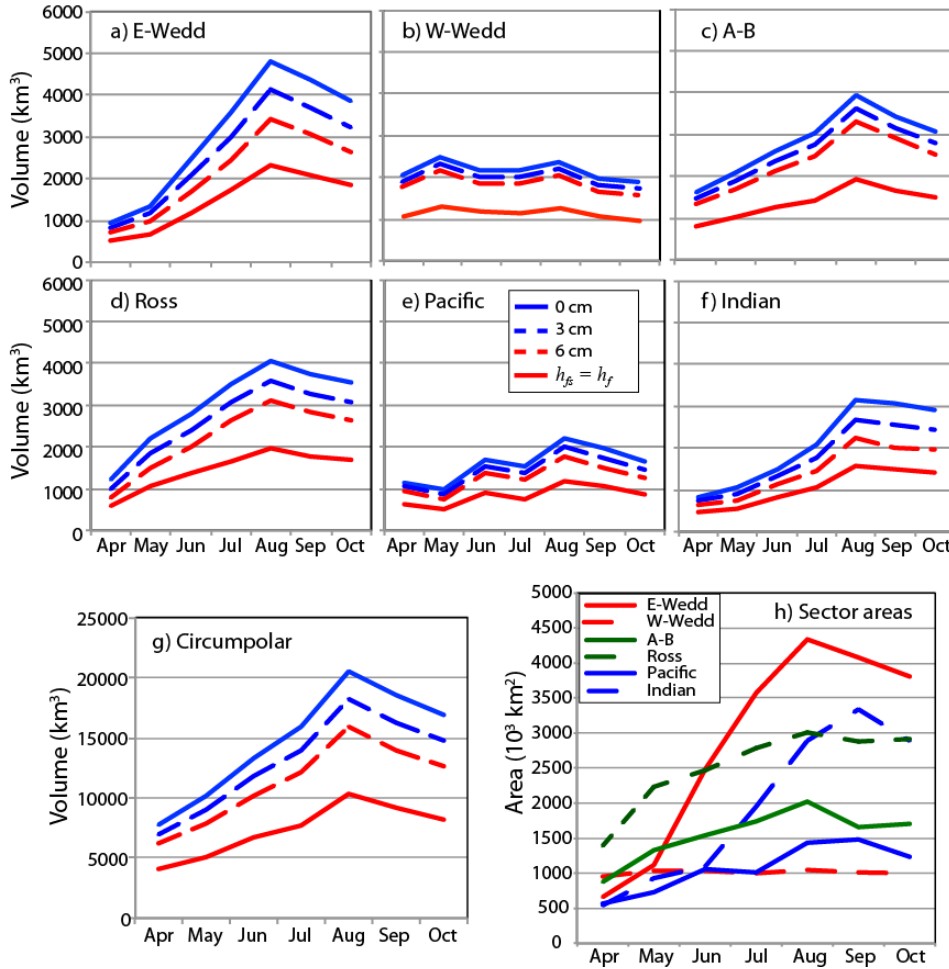

Figure 11. Evolution of the volume and area of the Antarctic sea ice cover between April and October 2019. November is not included here because the IS-2 data covered only half a month.