# Peer review of "The Antarctic sea ice cover from ICESat-2 and CryoSat-2: freeboard, snow depth and ice thickness"

_The Cryosphere, 2020_

## Short Comment (SC1) · 1 Jul 2020

Some additional CS-2 studies exist which support the uncertainty surrounding CS-2 freeboard retrievals which would support the study.

Fons, S. W. and Kurtz, N. T.: Retrieval of snow freeboard of Antarctic sea ice using waveform fitting of CryoSat-2 returns, The Cryosphere, 13, 861–878, https://doi.org/10.5194/tc-13-861-2019, 2019.

Price, D., Soltanzadeh, I., Rack, W., and Dale, E.: Snow-driven uncertainty in CryoSat-2-derived Antarctic sea ice thickness – insights from McMurdo Sound, The Cryosphere,

13, 1409–1422, https://doi.org/10.5194/tc-13-1409-2019, 2019.

Price, D., Beckers, J., Ricker, R., Kurtz, N., Rack, W., Haas, C., . . . Langhorne, P. (2015). Evaluation of CryoSat-2 derived sea-ice freeboard over fast ice in McMurdo Sound, Antarctica. Journal of Glaciology, 61(226), 285-300. doi:10.3189/2015JoG14J157

Schwegmann, S., Rinne, E., Ricker, R., Hendricks, S., and Helm, V.: About the consistency between Envisat and CryoSat-2 radar freeboard retrieval over Antarctic sea ice, The Cryosphere, 10, 1415–1425, https://doi.org/10.5194/tc-10-1415-2016, 2016.

---

## Referee Comment (RC1) · Rachel Tilling (Referee) · 16 Jul 2020

———————————————————- Summary ———————————————————-

This paper presents the first Antarctic-wide combination of ICESat-2 and CryoSat-2 data over sea ice to provide snow depth, freeboard, thickness and volume. Honestly, I was concerned that the first paper of this type to cover Antarctica would feel rushed and leave me with a lot of unanswered questions. But this manuscript was clearly-structured and very thorough. I appreciate that the authors were transparent about the limitations of the method, but have still published what is an interesting, unique study. I'm happy to say that I learned a lot. I commend the authors' efforts and strongly

recommend this paper for publication. However, I do have some comments that should be addressed first. The number of comments is due to the length of the paper, not a reflection on the quality.

———————————————————– General comments ———————————————————-

- The authors need to be very clear and consistent with which "freeboard" they are referring to, i.e. radar(CS2)/lidar(IS2)/ice/snow. "Total" freeboard should really be "IS2 freeboard" for consistency with the fact that they are using "CS2 freeboard" as separate to "ice freeboard". We're still not fully aware of the uncertainties associated with IS2 penetration and retrievals, so to frame it as undisputed total freeboard is misleading.

- I'd like to know why they did not use a whole year of data

- The long sentences are confusing at times (e.g. P1L12-15). I appreciate this is a style preference but it was an issue for me. I suggest the authors re-read the paper and check for clarity throughout.

———————————————————– Specific comments ———————————————————-

P1P7: ". . .freeboards, snow depth, \*\*ice thickness\*\* and ice volume

P1P7: ". . .April \*\*1st\*\*. . ."

P1L8: The phrase "stands out" isn't very explanatory, How about "is the thickest" or similar

P1L15: Don't need the word "broadly" (or "surprisingly" above). These types of phrases distract from the narrative.

P1L15: This relates to my general comment above, about clarity regarding real and observed freeboard. The authors mention "biases in CryoSat-2 freeboards" but a more accurate statement would be ""biases in CryoSat-2 measurements of the ice freeboard".

P2L2: "several decades" -> "four decades"

P2L21: The statement that Kurtz and Markus (2012) "assumed that the snow depth is equal to the ice freeboard" is misleading. The Kurtz and Markus paper assumed that snow depth is equal to snow/lidar/total freeboard ("ice freeboard" should unambiguously be used to refer to the snow-ice interface). Better phrasing might be "assumed that the ice freeboard is zero, and so snow depth is equal to the total freeboard". They should really spell it out here because it's a key concept of the manuscript.

P2L23: "ice **and snow** cover"

P2L23-25: How are the author's familiar with the pros and cons of each method – have they been validated? If so, please provide references.

P22L26-27: Does "these approaches" refer to all approaches, or just empirical?

P3L11-12: Why do the authors use the 10 km product and not the 150-photon aggregate product? It would be useful here to explain that higher spatial resolution IS2 products are available, and why they chose this one.

P3L23: "...freeboard estimates **in the Arctic**."

Section 2.2: Are they using individual waveform thicknesses, and how is the concentration weighting done?

P3L26: "...thickness measurements **in the Arctic**..."

P4L10-13: This is a really nice summary!

P5L28-29: From this I understand that they're using the whole month for CS2 but only 2 weeks for IS2? This wasn't clear in the manuscript until now, or in the abstract. I'd suggest repeating the analysis for just 2 weeks of CS2 so it's a like-for-like comparison even though I know this isn't ideal for coverage. If they really feel strongly that they shouldn't, then the averaging windows and reasoning needs to be very clear in Section 2 and the abstract.

P5L33: Would benefit from a more up-to-date reference than Lange & Eicken, 1991

P7L15-19: Although it's inferred, perhaps really spell it out here that wave propagation is more of an issue in the Pacific and Indian Ocean Sectors because of the small spread in extent. I appreciate that the authors are consistently transparent about the complexities of the signal.

Section 4.1: What's the reference for a bulk density of 0.32 g/cm3 and uncertainty of of ±0.07 g/cm3 for Antarctic sea ice?

Section 4.1.2: I struggled with this description of the method. The way I understand it, they create daily snow depths only in grid cells where data are available. Therefore, the monthly composites should be weighted by the number of measurements in each grid cell. Please describe if/how this weighting was done. How do they account for anomalous cells, or cells that are only present for a few days and may bias averages (especially as they're allowing such large temporal separation)? This is a critical section, and the method should be made clearer.

P9L10-11: Are the IS2 thickness estimates also concentration weighted?

P9L32: "may be" or "is"? There's an important distinction!

Section 4.3: I appreciate that they've included this section, but I think it's unnecessarily complicated. The same point could be made by just this final sentence on Page 10. That sentence does need rewording, for clarity, and I suggest something like "The negative intercepts observed in the scatterplots imply that $h^f$ is an underestimate of true snow depth by +2.4 to +3.9 cm."

P11L4: Reference for the -5C value? I'd really like to read this work.

P12L33: A more accurate statement would be that "the ASPeCt data are biased towards thin and level ice types"

———————————————————— Technical comments ————————————————————

-

P4L7: "export" -> "exports"

P5L26" Remove "generally"

P5 L28: Delete "due"

P6L9: "on" -> "in"

P6L13: "The tails **of** freeboard..."

P7L19: "...**but** some..."

P11L31: Delete "viz"

P12L27: "data set" -> "data"

P13L37: "sector" -> "section"

---

## Referee Comment (RC2) · Anonymous Referee #2 · 10 Aug 2020

General comments: This is a challenging and valuable paper to attempt to map the distribution of snow depth, sea ice thickness, and ice volume for Antarctic sea ice on a hemispheric scale for the first time, by combining satellite lidar (ICESat-2) and radar (CryoSat-2) altimeters. The major motivation is to improve our understanding of the recent decreasing trend of Antarctic sea ice extents. For this purpose, the authors estimated the surface elevation with ICESat-2 and the ice freeboard with CryoSat-2 and obtained the snow depth distribution from the difference between these datasets and the ice thickness and ice volume distribution assuming isostatic balance. They also conducted the error estimates from uncertainties of various factors that contribute to the freeboard measurements. As a result, the geographical and seasonal properties of

freeboard, snow depth, and ice thickness were revealed on a hemispheric for the first time. Besides, by comparting the two datasets, some unique features are suggested, such as more than 60-70% of the total freeboard is snow.

It is well known that the behavior of the Antarctic sea ice extents has different characteristics from that of the Arctic sea ice extents. However, the mechanism has not been well understood due to the lack of the hemispheric scale information of the Antarctic sea ice so far. While Worby et al. (2008) showed the hemispheric ice thickness distribution of Antarctic sea ice by compiling the visual observations conducted according to the ASPeCt protocol, there has been a lot of uncertainties about the seasonality and the biases caused by the observational methods. I think many scientists have been waiting for the estimation of the hemispheric snow depth, ice thickness, and ice volume distribution based on the satellite datasets. This paper can provide a breakthrough about this topic and contain a lot of implications. Therefore, I recommend publication with minor revisions. Having said that, I have several concerns. I would appreciate it if the authors address them before publication. The major points are as follows:

1) The lack of discussion about the different footprints of the two satellite sensors. Since the distributions of snow depth and ice thickness are usually anisotropic especially at deformed ice area, I am wondering if difference in footprint might affect the results. Even though the precise discussion might be difficult, I recommend some discussion about this.

2) Units of parameters In the manuscript, the CGS unit (cm, g, g/cm3) and MKS unit (m, kg, kg/m3) are mixed, which might be confusing. I think it would be better to unify them to SI unit.

3) Discussions with the correlation between IS-2 and CS-2 (Fig. 4) There are several speculations about the dominant growth processes based on the correlation between the IS-2 derived and the CS-2 derived freeboards at each subsection in section 3.2 (for example, P6L39-P7L3, PL14-L19). However, I feel there are some other possible

reasons for good correlations between them and the ground for their speculation is not necessarily strong. So further evidence might be needed. Please first explain in what kind situation the correlation becomes high, and then discuss the possible processes in each sector.

4) The suggestions of future field observation based on the results I would recommend the authors to suggest what kind of field observations will be required in the future to improve the accuracy of their estimations, based on their results, in the conclusion section. In the Antarctic sea ice area, there are complex snow-ice conditions, such as the presence of slush layers caused by flooding, a wide range of snow density caused by snow metamorphosis, the presence of void layers caused by deformation processes. Such suggestions would be very useful for the research community.

Specific comments:

*(P2L23-24) "the first approach... The second.... The third method..." Please add citation.

*(P2, section 2) Please add the footprint of each sensor.

*(P4L18) "signs indication" might be "signs indicate".

*(P4L19) "Snowfall adds to the snow layer" To be exact, "Snowfall precipitation minus evaporation (P-E)".

*(P4L20) "fvalue" might be "value"?

*(P6L19-20) "Both the total and CS-2 freeboards..." What do you mean by "a balance of different processes"?

*(P6L25) "0.75x10^6 km^2" might be "0.75x10^6 km^2 per year"?

*(P7L4) "160oW and 90oE" might be "160oE and 90oE"?

*(P8L7) "one free parameter" Could you explain what this parameter means physically?

*(P8L11) Please add ", respectively" after snow-ice interface".

*(P9L2) What caused the uncertainty in snow density? Spatial variation, or measurement error?

*(P9L3) What do you mean by "one free parameter"?

*(P9L5) I would recommend the authors to change the name of this subsection title to "sensitivity of the sampling frequency to calculations" or something like that. The current title might not be straightforward.

*(P9L30-32) I am wondering if this explanation is sufficient. I think more detailed discussion about the spatial scales of deformation and the sensor's footprint.

*(P10L6) "is likely due to.." You can add "and also smaller amount of P-E compared with other regions" The annual mean P-E distribution around the Antarctica is given by the following paper:

Cullather, R.I., Bromwich, D.H., and Van Woert, M.L. (1998) Spatial and temporal variability of Antarctic precipitation from atmospheric methods. Journal of Climate, 11, 334-367.

Toyota T., Massom R., Lecomte O., Nomura D., Heil P., Tamura T. and Fraser A.D. (2016) On the extraordinary snow on the sea ice off East Antarctica in late winter, 2012. Deep-Sea Res. II, 131, 53-67.

*(P10L7-8) "The spatial patterns show.." It might be possible that this is just because the ice-covered period becomes shorter toward the marginal ice zone. What do you think?

*(P10L14-15) "In all other sectors, we find. . ." The result is quite interesting. This might be a good evidence especially for the loss into leads, as suggested by the above paper.

*(P10L30-37) In the end, what du you think is the major reason for the negative bias?

*(P11L26) "by assuming that the snow depth is equal to the total (or IS-2) freeboard." Is this based on the observational facts? If so, please cite some papers which support this idea. If you can justify this assumption, it would be supportive of your results.

*(P14L31) "an indication of total freeboard changes rather actual change in ice thickness" Then, what caused the change in freeboard?

*(Figure 4) "Total freeboard" might be changed to "Total (IS-2) freeboard" to avoid confusion. Please add the explanation about what the color means in Fig.4a.

*(Figure 5) It is hard to detect what the color means. The color bar should be placed at the bottom of the figure.

*(Figure 6) Please add the explanation about what the thick solid line means.

That is all. Faithfully yours.

---

## Author Comment (AC1) · 21 Aug 2020

Dear Dr. Price,

Thank you for providing the list of publications. We will consider these in the revision of our manuscript.

Sahra Kacimi and Ron Kwok

---

## Author Response (AR1)

Dear Dr. Heil,

Thank you for the detail reading of our manuscript. In additions to the valuable suggestion the corrections to the small details were useful for improving the overall manuscript. In addition to our response to you (attached below) and the reviewers (submitted earlier), we would like to bring to your attention two changes that are of note in the revised manuscript:

1. In the conclusions section of the original manuscript, we stated that: "…the adjustment of the CS-2 orbits to provide improved coincidence in space-time sampling of the surface is being considered. A Joint NASA-ESA working group is exploring this opportunity, while there is an overlap in the IS-2 and CS-2 missions, to 'tune' the CS-2 orbital parameters slightly to improve the time-separation between near co-incident IS-2 and CS-2 measurements (cross-overs and along-track sampling). If this is approved. the two altimeters will provide a crucial data set for not only understanding the current retrievals but also the design of future instruments tasked to understanding the development of the Antarctic sea ice cover…"
   In fact, this has been approved and implemented. In late May (which you noted), after a year of consideration and planning, ESA announced the CRYO2ICE project to adjust the CS-2 orbits to improve coincidence for better utilization of the two altimeter missions to benefit sea ice and other science disciplines. The CS-2 orbit was adjusted in July and is now providing better coincidence in the Arctic. After the Arctic winter, CS-2 will be adjusted to optimize coverage for the lower latitude Southern Ocean ice cover. This is good news and the conclusion has been revised to deliver this message.

2. We reprocessed all of the IS-2 and CS-2 data with our most current analysis software, so all of the results will be slightly different (centimeter level freeboard changes). Also, we have removed July from our results because during July of 2019 there was a spacecraft anomaly and the IS-2 project recommended that we not use this release of July until the altimeter data for that period have been thoroughly checked out. This does not change any of the science and conclusions in the paper although the results are slightly different. In the attached figure (Figure 4 in the text), we show the differences between the mean CS-2 and IS-2 freeboards and their squared correlations from the original manuscript and the revised manuscript (top row: old; bottom row: new). It can see that the mean values are different but the relative month-to-month variabilities and correlations between the two freeboards for the different sectors remain substantially the same (Note also that in the original manuscript the Indian and Pacific sectors were switched – indicated by the arrows, and they have been corrected in the revision.). We feel that leaving out July is the correct approach at this time.

Cordially,

Sahra Kacimi and Ron Kwok

[Figure]

Top panels: submitted manuscript
Bottom panels: revised manuscript

**Responses to Editor's comments** (*in blue*):

*General comments:*
*\* Use of SI units:*
*Similar to reviewer 2, who asked for the uniform use of units, and recommended the use of SI units, I would like to request the same. Including to remove "PSU" as salinity is now given without any units, not even "PSU". (1-16).*

All the density values are now expressed in kg/m^3. The salinity units have been removed. Because of difference in the magnitude of snow depth and ice thickness (a factor of ten) we have consistently and prefer to keep snow depths in centimeters and thicknesses in meters. We believe the community (at least the remote sensing community) generally thinks in those units.

*\* Hyphenation: Pls review your use of hyphens, especially change "snow-radar" to "snow radar" throughout the manuscript (ms).*

Done.

*\* Include information on cloud etc affecting IS-2 data (i.e., missing data) and how this lack of data coverage affects the comparison. Similar, explore effects of different instrument footprints and repeat/overlapping swaths. (I found some info in section 4.12 but would prefer the basic info upfront, when the datasets are introduced.)*

All our calculations are performed with 25-km averages. Similar to our answer to the second reviewer, we will note this in the sampling description (Section 4) to alert the reader to the potential differences due to the resolution in freeboard retrievals from IS2 and CS2. The information about instrument footprints on the IS-2 data has been added in the data description section. Cloud contaminated retrievals are not used in our calculations (we have added this remark to the data description).

*\* In the Southern Ocean, changes in snow thickness/cover are largely driven by a combination of solid precipitation as well as by snow redistribution (incl. loss of snow into open water, such as leads) -- in addition to snow-ice formation. --> Pls include the "snow redistribution" (with appropriate reference) in your ms, i.e., 4-10 to 4-13.*

Done. - (Andreas & Claffey, 1995; Massom et al., 1997; Massom et al., 1998)

*\* Suggest to rename "East Weddell" and "West Weddell" to "eastern Weddell Sea" and "western Weddell Sea", respectively. Abbreviations may remain the same.*

We appreciate the suggestion, however for simplification we prefer the use of East and West Weddell sectors. Also in our definition, the eastern Weddell sector does not cover exclusively the Weddell Sea but also the Lazarev and the Riiser Larsen seas.

*\* Provide essential information for both primary datasets (IS-2 FB and CS-2 radar FB)*

*(Sections 2.1 and 2.2). For example, information on the data granularity for CS-2 radar FB needs to be presented here - despite provision of the KC2016 reference.*

We do not quite understand the granularity requested here. The IS-2 data products are detailed in the project documents available at the National Snow and Ice Data Center. We added more detailed description of both instruments and hope that they are sufficient for the purposes here.

*See below for specific comments.*
*\* Fluxgates: How are they set and what contributes to the lesser ice area in the larger fluxgate? (6-1ff)*

In the paper we mention two flux gates: One along the 1000m isobath that parallels the ice fronts of the Ronne and Filchner ice shelves; and one along the 1000m isobath that parallels the ice front of the Ross Sea Ice Shelf.

*\* Your discussion of the regional freeboard evolutions states that the E-Wedd sector exhibits the "lowest area-averaged freeboards" around Antarctica. Tab. 2 however shows similarly low freeboards for the Indian and not much higher for the Pacific. Would you kindly explore this?*

We have added a remark on their relative freeboards and that the same processes (large divergence) may also be at work here.

*\* The discussion on snow-depth estimates (p9 & 10) should also consider the episodic events during which solid precipitation is delivered, versus the more frequent redistribution of snow (by wind-induced drift). -- This is an area of research by itself, but worthwhile to mention here.*

We agree and added a mention of solid precipitation and wind redistribution.

*\* Update the final lines (15-34ff) as this CS-2 alignment to IS-2 has taken place by now. Suggest to give a small outlook of the anticipated benefits of the Cryo2Ice objectives.*

This had been updated to read: "…An adjustment of the CS-2 orbits (by ESA) – CRYO2ICE – to provide improved coincidence in space-time sampling of the two altimeters has been successfully implemented. We anticipate that the data acquired by CRYO2ICE will provide a crucial and valuable data set for not only understanding current retrievals but also the design of future instruments tasked to understand the development of the Arctic and Antarctic sea ice covers…."

*Specific comments:*
*1-7: Provide start date of your analysis: "April xx, 2019".*
Corrected.

*1-8: Correct "West Weddell sector" to "Western Weddell sector".*
See our comment above.

*1-8: Provide at least a qualitative statement instead of "stands out with a mean sector thickness > 2 m."*
Revised to read: "The multiyear ice observed in the West Weddell sector is the thickest with a mean sector thickness > 2 m…".

*1-9: Spell out "Ronne": "Ronne Ice Shelf".*
Corrected.

*2-2: Correct "sea ice extents" to singular "sea ice extent".*
Corrected.

2-3: Suggest to change "decay rates" to singular as well, i.e., interpreting it as the interannual decay rate.
We believe decay rates are used correctly here.

*2-37: "OIB" needs to be defined. So do this here: "IceBridge [OIB]),".*
Corrected.

*3-15ff: Need to provide some more info on the ATLAS and its build, i.e., 3 pairs of 2 beams ands ome essential info. Short will be fine. Consider a reference too... but need to include essential information here, so reader, for example, can understand meaning of "strong beam".*
For clarification we added a brief description of ATLAS at the beginning of paragraph 2.1: "the Advanced Topographic Laser Altimeter System (ATLAS) onboard ICESat-2 (IS-2) uses three beam pairs to profile the surface. The pairs are separated by about 3.3-km cross track. Each pair consists of a strong and a weak beam with a inter-beam spacing of 90m. The pulse energies of the strong beams are ~4 times that of the weak. Each beam profiles the surface at a pulse repetition rate of 10 kHz."

*3-20: Specify "Vertical" to read "Vertical uncertainty".*
Now reads: "…Uncertainty in IS-2 freeboard retrievals…"

*3-24: Provide information on which "satellite-derived ice concentration(s)" have been used.*
Done.

*3-25: Provide information on why "freeboard is approximately one-ninth of ice thickness", i.e., include density argument and a reference. This might make the sentence long, consider splitting into 2 sentences, as required.*
Revised to read:"… Nothing that freeboard is approximately one-ninth of ice thickness (due to the density contrast between, ice and seawater)…"

*3-26: Move "(Kwok & Cunningham, 2015)" forward to read "differences (Kwok & Cunningham, 2015)".*
Corrected.

*3-30: What changes are meant here? "expected changes in IS-2 and CS-2 freeboards": Temporal. -->Specify here.*
Corrected to read "time-variable …."

*3-34: Suggest to change "east sector and west sector" to "eastern sector and western sector".*
*See our comment above.*

*3-34: Change "added a coastal Amundsen-Bellingshausen region" to "divided the Amundsen-Bellingshausen Sea into a coastal region and remainder". Include info on which latitude was chosen for the separation.*
The Amundsen –Bellingshausen sector includes the coastal sector. The coastal region was added to better capture the mechanical convergence event as will be described in the section.

*3-35: Could be more specific than "remarkable" in "remarkable ice convergence".*
Revised to read: "…remarkable ice convergence seen in 2019 (discussed below)."

*3-35: Change "seen" to "observed" or "recorded".*
Done.

*4-2: Suggest to change "worthwhile reviewing" to "necessary to review".*
We prefer the current phrasing.

*4-5: Add a space " " before "and" near beginning of line.*
Corrected.

*4-5: Remove a space " " before "deltah_i".*
The space is a characteristic of Mathtype.

*4-5: Add a space " " before "(Figure 2)".*
Done.

*4-7: Correct "Arctic Ocean export" to "Arctic Ocean exports".*
Corrected.

*4-7ff: Pls rewrite this longwinded sentence for clarity. Also avoid a "(...)" following another set of brakets "Kwok et al., 2013)".*
Revised to read:" Since Arctic Basin exports only ~10% of its area annually (mainly through the Fram Strait - Kwok et al. (2013)) and there is relatively little melt in winter away from the ice margins. Therefore, it is simpler to observe a coherent seasonal cycle of freeboard growth over a fixed region of the Arctic Basin.(i.e.; the correlated increases in both the IS-2 and CS-2 freeboards seen in Kwok et al. (2020)).."

*4-11: Include a reference for "larger ice divergence" (i.e., for Antarctic sea ice). : In "larger ice divergence", larger than ??? Pls complete sentence.*
Added: "…Compared to the Arctic…"

*4-12: Replace "export" with "transport" in "large-scale export".*
We prefer export because the ice is exported from the interior compact regime.

*4-15: Appears as if there are erronous spaces " " before and after "h_fs" and "h_fi".*
These are formatted with Mathtype.

*4-17: Appears as if there is an erronous space " " before "beta".*
These are formatted with Mathtype.

*4-20: Correct "fvalue" to "value".*
Corrected.

*4-21: "spatially" is redundant. Could be cut.*
Removed.

*4-21: Change "wind forcing" to "wind stress".*
Changed.

*4-21: Change "is sometimes" to "may be".*
Changed.

*4-27: Correct "Mechanical convergence/divergence" to "Mechanical redistribution due to convergence/divergence"*
Changed.

*4-31: Remove "no doubt".*
Removed.

*4-33: Suggest to refocus "is the large export and sea ice melt at the margins." to also consider the advective changes in the regional freeboard estimates, i.e., different 'parcels of sea ice (with snow)' are found in a certain region at a certain time due to the ice being advected.*
Modified to read:"… regional variability of freeboard (below) is the advective changes and sea ice melt at the margins…"

*5-5: Clarify if the "Amundsen-Bellingshausen (A-B)" includes the data for the "coastal Amundsen-Bellingshausen (CoA-B)".*
Corrected.

*5-8: Correct "Amundsen and Bellingshausen Seas" to "Amundsen and Bellingshausen seas".*
Corrected.

*5-10: Naming here "CoA-B sector" but in Fig.1 it is named "A-B Int". --> Unify.*
Corrected.

*5-20: Add "atmospheric" and remove "setup" to "The atmospheric circulation".*

Changed.

*5-23: "i.e., ice freeboard tends to be anti-correlated to snow accumulation": Pls clarify if this is a statement based on previous publications (then pls include reference), or based on the data analyzed here (then add an exact figure reference). -- In general, the thick or deformed ice also exhibits thick snow cover (i.e., Uto et al., 2006; Maksym and Markus, 2008; Sugimoto et al., 2016).*
This is a physical statement: Snow loading tends to depress the sea ice freeboard and therefore anti-correlated.

*5-31: Correct "40^oW and 62^oW" to "40^o and 62^oW".*
Corrected.

*5-32: Need to mention as early as here that the Weddell Sea is a (cyclonic) gyre. Otherwise any reader without prior knowledge would miss out crucial information to interpret the statement "both with boundaries that are open to the north."*
Done.

*5-32: Replace "areas" with "regions".*
Replaced.

*5-33: Add note that not all sea ice in the Weddell Sea is formed in the E-Wedd ... but some also arrives in the Weddell Sea via advection within the westward coastal current.*
We think broadly speaking the largest contribution is ice production.

*5-35: Change "as well," to "as well as".*
We believe the phrasing is correct here.

*5-36: Change "ice areas added" to "ice areas are included".*
We mean area added by divergence.

*5-37: Remove "are entrained in the outflow".*
Removed.

*5-36: Rewrite "and formed in the Ronne and Brunt ice shelves" to correct. Sea ice does not form in ice shelves per se.*
Revised to read:"… formed seaward of the Ronne and Brunt ice shelves…"

*6-2: Pls provide some information on the "flux gate" extent (in km).*
See our comment above.

*6-2: Do you mean "~1100km." or "~1100m.", i.e., refernce to another isobar??*
Here, we are referring the length of the fluxgate.

*6-9: Change "On an area-averaged sense," to "Genearlly,".*
Changed.

*6-16: Clarify the location by replacing "in the sector" with "in the E-Wedd". Suggest to remove "As for the E-Wedd,"*
Changed.

*6-23: Correct "Sound Polynyas." to "Sound polynyas."*
Corrected.

*7-4: Change "160^oW and 90^oE" to "90^oE and 160^oW".*
Changed.

*7-4: Correct "90^oE and 15^oE" to "15^o and 90^oE".*
Changed.

*7-5: Clarify text by chagning "the broader extent of the ice cover in Indian Ocean Sector (around 15^oE and 40^oE)" to "wider latitudinal pack ice extent in the Indian Ocean Sector (from 15^o and 40^oE)".*
Revised to read: "…larger extent…"

*7-6: Change "the ice cover occupies a very narrow band, and" to "the ice covers a narrow band, that"*
Changed.

*7-7: Regarding the 2019 ice drift (being westward): Why is this remarkable? Mention the separation of the westward coastal current with eastward drift in the southern bands of the ACC.*
There is nothing remarkable – we reported that this was consistent with the mean multiyear drift fields in Fig. 5.

*7-10: Rewrite to improve readability "the Indian Ocean sector is seasonal ice from coastal polynyas and from the Pacific Sector." and include that sea ice forms locally in the Indian Ocean sector OUTSIDE "coastal polynyas". Text currently reads as if all sea ice formed in the Ind Oc has done so in a coastral polynya.*
Rewritten as suggested.

*7-19: This sentence is incomplete: "We have filtered most of these anomalous freeboards in the IS-2 and CS-2 processing some are still present."*
Revised.
*At least start with "Although we have".*
*--> Provide some detail and a refernce on the filtering of wave-effected data.*
This was done visually and is now stated in the text.

*8-8: Remove comma ",' before " (Ulaby".*
Removed.

*8-10: Include "snow" to read "bulk snow density".*

Done.

*8-10: Pls use SI units. I.e., avoid "0.32 g/cm 3". - Throughout ms, pls.*
Done.

*8-20: Include brief motivation for chooseing 30cm freeboard differences, for completeness.*
Typical winter value used as an example.

*9-7: Change "(compared to the Arctic)" to "than the Arctic". Or better simplify the complete sentence. I.e.: Antarctic sea ice is found at lower latitudes, hence a lesser coverage by polar-orbiting satellites proves difficult when deriving snow depth from IS-2 or CS-2 freeboards.*
Revised as suggested.

*9-8: "Gridded IS-2 freeboards are averages of the three strong IS-2 beams ..." -- Provide footprint comparison of IS-2 freeboards with CS-2 freeboards.*
Because the freeboards are all matched at a 25km resolution we feel there is no need to discuss their respective footprints.

*9-10ff: Are both, IS-2 and CS-2 derived/treated the same way here? I.e., what about adjusting IS-2 for ice concentration? I am also concerned about the temporal coverage (per location pixel) for each derived data set. --> Would you pls include some background on this here? (I know IS-2 data are only there for ice concentration above 50%.*
The ICESat-2 data is not weighted for ice concentration, as the open water is already included in the freeboard calculations.

*9-15: Correct "75-km" to "75 km".*
We think 75-km is the correct usage here. It refers to a 75-km box, not 75 km in distance.

*9-19: This "advantage", however, reduces the true sample variability and due to IS-2 data gaps this is likely to bias the derived results.*
True, although the simulations suggest the effect to be minor.

*9-30: Would you agree that the spatio-temporal averaging plus the footprint limitations contribute to make this variance in snow depth appear so low? -- Some discussion on this would be useful here. There is also the loss of snow when entering into the open ocean during a deformation event, i.e., it is not accounted for as snow-ice.*
We expect the variance to be representative of the length scale chosen (i.e., 25 km). So yes, this would be lower than what one would expect at the 10s of meters. We have added that the loss of snow during deformation may have a confounding effect.

*9-31: Avoid use of "results" twice in same sentence: "These results suggest that the effect of sea-ice deformation in biasing the derived snow depth may be small."*
Revised to read: "These results suggest that the effects of sea-ice deformation in biasing the snow depth estimates may be small"

*9-31: Ensure to replace "dynamics" with "deformation".*
Corrected.

*10-9: Correct "southern Ross and Weddell Seas" to "southern Ross and Weddell seas".*
Corrected.

*10:31: The zero freeboard/zero snow thickness argument leaves room for further discussion. Is this based on Ozsoy-Cicek et al (2013)? Suggest to include a qualifier here.*
No, this is a physical statement based on definition. Revised to read:"… as one should expect – by definition – zero snow depth at near zero IS-2 freeboard…"

*11-1ff: Provide reference(s) for the retracker and related information such as temperature dependence.*
Revised to read:"… One likely source of these biases is the displacement of retracking point (RP) of the radar altimeter (CS-2) away from the snow-ice interface resulting in higher CS-freeboards (Kwok, 2014). At $K_u$-band frequencies (CS-2), the RP's are displaced from the true ice surface when elevated snow salinities (due to brine-wicking, flooding) are found near the snow-ice interface, or the changes in scattering the presence of moisture in the snow layer when air temperature warms (Winebrenner et al., 1994)…"

*12-3: Consistency is important: Here densities are in kg/m^3. Pls use this SI unit throughout the ms.*
All densities are now in kg/m3.

*12-16: Correct "(mean 2.40±1.00 m) in May)" to "(mean 2.40±1.00 m in May)".*
Corrected.

*12-16: Correct "CoA-B sectors" to "CoA-B sector".*
Corrected.

*12-19: Correct "Ross Sea polynyas" to "Ross Sea Polynya".*
The lower case is correct in this case because we are referring to multiple polynyas in the Ross Sea.

*12-26: Change "between April and November" to "from April to November".*
Changed.

*12-27: Change "field records" to "field observations".*
Corrected.

*12-29: There are updates to the ASPeCt database from those included in Worby et al. (2008). No updated climatology. The cleaned and calibrated data can be accessed at: https://data.aad.gov.au/aspect/ Select "Download cruises". (Access via AADC, so "free login" required.)*

Thank you. We added the data citation to the acknowledgment.

*12-35: Correct "Bellingshausen and Amundsen Seas" to "Bellingshausen and Amundsen seas".*
Corrected.

*12-35: This is correct for "operational underway data". If a vessel however travels along a science transect, then it follows the waypoint line without deviations motivated by the "hunt for thinner ice". -- ASPeCt should flag these.*
Noted.

*13-3: As a field going scientist, I refute this. Sea-ice transects should be laid out to be representative of the ice conditions of the entire ice floe and the general region, so would include ridged areas and thick ice, and a range of snow covers.*
Noted. I assume that during drilling that thinner ice is avoided for safety considerations.

*13-8 and 13-10: Be consistent: Reporting ice thicknesses from the same source should provide them in a uniform presentation: I.e., number of decimals: "2.4 to 2.6 m" versus "0.48 and 0.99 m".*
Revised.

*13-16: Fullstop over "unlikely" missing.*
Corrected.

*14-33: Change "In this paper," to "In this study".*
Corrected.

*15-6: Correct "Ross Sea polynyas" to "Ross Sea Polynya". Alternatively rewrite to "Ross Sea and Ronne polynyas".*
Corrected.

*15-20: Remove erroneous "the" from "from basal-layer salinity the is sound."*
Removed.

*15-26: Add a sentence here to recommend urgent need for sustained and extensive field measurements.*
Added,

*15-30: Add fullstop at end of sentence.*
Corrected.

*15-32: Correct "altimeter" to "altimeters".*
Corrected.

16-4: Include acknowledgement/source of CS-2 data.

Similar for the PM ice-concentration data used, and any other data (i.e., atmospheric reanalysis if the Davis Strait Low pressure pattern was updated here).
Corrected.

20-7f: Check doi in Giles et al. (2008) for correctness. - Remove linebreak.
20-15: Remove "Doi " (including space).
20-39: Remove "Doi".
20-43f: Correct doi for Maksym and Markus (2008).
20-46: Remove "Doi " (including space).
20-48: Remove "Doi " (including space).
21-29: Remove "Doi " (including space).
21-36f: Correct doi for Maksym and Markus (2008).
21-43f: Correct doi for Yi et al. (2008).

All doi's used here are from the endnote and formatting is based on APA recommendations. We assume that the typesetting process will correct these anomalies.

Fig1: Change caption to "Naming convention for the sea-ice sectors around Antarctica." or similar.
Done.

What is "A-B Int"?
Corrected to CoA-B.

Fig3: Include info on pixel size for IS-2 and CS-2 data shown in figure (and hence also for derived snow depth).
Done.

Fig4a: Colourbar missing.
There is no colourbar, they are normalized distributions.

Fig5: Colourbar and scale for ice drift are missing.
The colorbar and scaling is displayed in the middle of the figure.

Fig6a&b: Colourbar missing.
There is no colourbar, they are normalized distributions.

Fig9a&b: Colourbar missing.
There is no colourbar, they are normalized distributions.

Kwok, R., S. Kacimi, M. A. Webster, N. T. Kurtz, & A. A. Petty. (2020). Arctic Snow Depth and Sea Ice Thickness From ICESat‐2 and CryoSat‐2 Freeboards: A First Examination. *Journal of Geophysical Research: Oceans, 125*(3). doi:10.1029/2019jc016008

Massom, R. A., M. R. Drinkwater, & C. Haas. (1997). Winter snow cover on sea ice in the Weddell Sea. *Journal of Geophysical Research-Oceans, 102*(C1), 1101-1117. doi:Doi 10.1029/96jc02992

Massom, R. A., V. I. Lytle, A. P. Worby, & I. Allison. (1998). Winter snow cover variability on East Antarctic sea ice. *Journal of Geophysical Research-Oceans, 103*(C11), 24837-24855. doi:Doi 10.1029/98jc01617

Winebrenner, D. P., E. D. Nelson, R. Colony, & R. D. West. (1994). Observation of Melt Onset on Multiyear Arctic Sea-Ice Using the Ers-1 Synthetic-Aperture-Radar. *Journal of Geophysical Research-Oceans, 99*(C11), 22425-22441. doi:Doi 10.1029/94jc01268

**Responses to Reviewer 1's comments** (*in blue*):

*This paper presents the first Antarctic-wide combination of ICESat-2 and CryoSat-2 data over sea ice to provide snow depth, freeboard, thickness and volume. Honestly, I was concerned that the first paper of this type to cover Antarctica would feel rushed and leave me with a lot of unanswered questions. But this manuscript was clearly structured and very thorough. I appreciate that the authors were transparent about the limitations of the method, but have still published what is an interesting, unique study. I'm happy to say that I learned a lot. I commend the authors' efforts and strongly recommend this paper for publication. However, I do have some comments that should be addressed first. The number of comments is due to the length of the paper, not a reflection on the quality.*

We thank the reviewer for her time in reviewing the manuscript and providing helpful feedback. The suggestions have helped improve the revised manuscript.

—————————————————— *- General comments* ——————————————————

*The authors need to be very clear and consistent with which "freeboard" they are referring to, i.e. radar(CS2)/lidar(IS2)/ice/snow. "Total" freeboard should really be "IS2 freeboard" for consistency with the fact that they are using "CS2 freeboard" as separate to "ice freeboard". We're still not fully aware of the uncertainties associated with IS2 penetration and retrievals, so to frame it as undisputed total freeboard is misleading.*

Agreed. All occurrences of 'total freeboard' in the text have been replaced by 'IS2 freeboard'.

*I'd like to know why they did not use a whole year of data.*

The period of study covers April to November of 2019. The summer and transition months were not included because of the known impact of warmer temperatures on snow wetness and thus radar (CS2) penetration into the snow layer.

*The long sentences are confusing at times (e.g. P1L12-15). I appreciate this is a style preference but it was an issue for me. I suggest the authors re-read the paper and check for clarity throughout.*

The sentences from P1L12-15 were rephrased to read: "The remarkable mechanical convergence in coastal Amundsen Sea, associated with onshore winds, was captured by ICESat-2 and CryoSat-2. We observe a corresponding correlated increase in both freeboards, snow depth and ice thickness. While the spatial patterns in the freeboard, snow depth, and thickness composites are as expected, the observed seasonality in these variables is rather weak. This most likely results from competing processes (snowfall, snow redistribution, snow-ice formation, ice deformation, basal growth/melt) that contribute to uncorrelated changes in the total and radar freeboards."

*P1P7: "...freeboards, snow depth, \*\*ice thickness\*\* and ice volume.*
Corrected.

*P1P7: "...April \*\*1st\*\*..."*
Corrected.

*P1L8: The phrase "stands out" isn't very explanatory, How about "is the thickest" or similar*
Revised to read: "The multiyear ice observed in the West Weddell sector is the thickest with a mean sector thickness > 2 m…"

*P1L15: Don't need the word "broadly" (or "surprisingly" above). These types of phrases distract from the narrative.*
We deleted the word "Broadly".

*P1L15: This relates to my general comment above, about clarity regarding real and observed freeboard. The authors mention "biases in CryoSat-2 freeboards" but a more accurate statement would be ""biases in CryoSat-2 measurements of the ice freeboard".*
We replaced "biases in CryoSat-2 freeboards" by "biases in CryoSat-2 estimates of ice freeboard".

*P2L2: "several decades" -> "four decades"*
Done.

*P2L21: The statement that Kurtz and Markus (2012) "assumed that the snow depth is equal to the ice freeboard" is misleading. The Kurtz and Markus paper assumed that snow depth is equal to snow/lidar/total freeboard ("ice freeboard" should unambiguously be used to refer to the snow-ice interface). Better phrasing might be "assumed that the ice freeboard is zero, and so snow depth is equal to the total freeboard". They should really spell it out here because it's a key concept of the manuscript.*
We replaced "assumed that the snow depth is equal to the ice freeboard" by "assumed that the snow depth is equal to the total freeboard "

*P2L23: "ice \*\*and snow\*\* cover"*
Corrected.

*P2L23-25: How are the author's familiar with the pros and cons of each method – have they been validated? If so, please provide references.*
The approaches and shortcomings are discussed in the cited references - P2L20-22.

*P22L26-27: Does "these approaches" refer to all approaches, or just empirical?*
By "these approaches" we are referring to all of the approaches. To improve clarity, we changed to: "all these approaches".

*P3L11-12: Why do the authors use the 10 km product and not the 150-photon aggregate product? It would be useful here to explain that higher spatial resolution IS2 products are available, and why they chose this one.*

The freeboards that we use (from ATL10) have identical resolutions as those of the derived heights. Individual freeboards (i.e., 150-photon aggregate freeboards) are derived only if there was a local sea surface reference available within a 10-km along-track segment. The 10-km does not refer to the freeboard resolution, only the constraint placed on proximity to the sea surface reference.

*P3L23: "...freeboard estimates \*\*in the Arctic\*\*."*

We added "in the Arctic" at the end of the sentence P3L23.

*Section 2.2: Are they using individual waveform thicknesses, and how is the concentration weighting done?*

The weighting was done by multiplying the gridded CS2-freeboard (25-km averages) by the corresponding ice concentration at a given grid cell.

*P3L26: "...thickness measurements \*\*in the Arctic\*\*..."*

Added.

*P4L10-13: This is a really nice summary!*

Thank you.

*P5L28-29: From this I understand that they're using the whole month for CS2 but only 2 weeks for IS2? This wasn't clear in the manuscript until now, or in the abstract. I'd suggest repeating the analysis for just 2 weeks of CS2 so it's a like-for-like comparison even though I know this isn't ideal for coverage. If they really feel strongly that they shouldn't, then the averaging windows and reasoning needs to be very clear in Section 2 and the abstract.*

The November CS-2 composite and distribution (in Figures 3 and 4) have been updated using only two weeks of CS-2 data – to be consistent with the availability of IS2.

*P5L33: Would benefit from a more up-to-date reference than Lange & Eicken, 1991*

Added Vernet et al. (2019): Vernet, M., Geibert, W., Hoppema, M., Brown, P. J., Haas, C., Hellmer, H. H., . . . Verdy, A. (2019). The Weddell Gyre, Southern Ocean: Present Knowledge and Future Challenges. Reviews of Geophysics, 57(3), 623-708. doi:10.1029/2018rg000604

*P7L15-19: Although it's inferred, perhaps really spell it out here that wave propagation is more of an issue in the Pacific and Indian Ocean Sectors because of the small spread in extent. I appreciate that the authors are consistently transparent about the complexities of the signal.*

We agree and added the following text: 'This effect is predominant in the Pacific and Indian Ocean sectors because of smaller sea ice extent'.

*Section 4.1: What's the reference for a bulk density of 0.32 g/cm3 and uncertainty of*

_0.07 g/cm3 for Antarctic sea ice?

There is no generally accepted average value of the bulk density of snow over Antarctic sea ice. Massom et al. (2001) suggest 200-300 kg.m$^{-3}$ under cold/dry condition and higher density (350-500 kg.m$^{-3}$) for warm windy conditions – not unlike the Arctic. We elected to use the average winter bulk density of 320 kg.m$^{-3}$ (like that of the Arctic) but increased the spread to 70 kg.m$^{-3}$ to cover the range of average conditions. We have added this discussion to the text.

*Section 4.1.2: I struggled with this description of the method. The way I understand it, they create daily snow depths only in grid cells where data are available. Therefore, the monthly composites should be weighted by the number of measurements in each grid cell. Please describe if/how this weighting was done. How do they account for anomalous cells, or cells that are only present for a few days and may bias averages (especially as they're allowing such large temporal separation)? This is a critical section, and the method should be made clearer.*

For clarification purposes, we substituted the text P9L11-12 by: '…First, the daily along-track IS2 and CS2 freeboards are binned on a 25-km resolution grid. Gridded IS-2 freeboards are averages of the three strong IS-2 beams and thus provide a better sampling of the spatial mean (compared to CS-2 freeboards). In grid cells with available data, an average is computed while the other bins are assigned with a missing value. Freeboard differences are then computed at each valid IS2 grid cell using CS2 binned freeboards (weighted by ice concentration) with time separations |DT|<10 days and within a 75-km box. We find that this sampling strategy provides the best spatial coverage without sacrificing precision'.

*P9L10-11: Are the IS2 thickness estimates also concentration weighted?*

The IS2 estimates are not weighted by passive microwave ice concentration because the open water samples (i.e., H=0) are included in the population for computing thickness.

*P9L32: "may be" or "is"? There's an important distinction!*

We used "may be" because the results of the sensitivity analysis may not have sampled the range of conditions expected.

*Section 4.3: I appreciate that they've included this section, but I think it's unnecessarily complicated. The same point could be made by just this final sentence on Page 10. That sentence does need rewording, for clarity, and I suggest something like "The negative intercepts observed in the scatterplots imply that h$_f$ is an underestimate of true snow depth by +2.4 to +3.9 cm."*

We think this section is useful, as it constitutes the basis of the discussion about the possible sources of biases in the snow depth estimates. The final sentence on page 10 has been rephrased as suggested.

*P11L4: Reference for the -5C value? I'd really like to read this work.*

Figure 5 of Winebrenner et al. (1994) - Instead of T=0C, the appearance of moisture in the snow layer from around -5 C would cause changes in bulk dielectric constant affecting penetration and backscatter at radar frequencies. This can be seen in the

fluctuations in radar backscatter even before the air temperature reaches the melting temperature.

Winebrenner, D. P., Nelson, E. D., Colony, R., & West, R. D. (1994). Observation of Melt Onset on Multiyear Arctic Sea-Ice Using the Ers-1 Synthetic-Aperture-Radar. *Journal of Geophysical Research-Oceans, 99*(C11), 22425-22441. doi:Doi 10.1029/94jc01268

*P12L33: A more accurate statement would be that "the ASPeCt data are biased towards thin and level ice types"*
Revised as suggested.

──────────────── *- Technical comments* ────────────────

*P4L7: "export" -> "exports"*
*P5L26" Remove "generally"*
*P5 L28: Delete "due"*
*P6L9: "on" -> "in"*
*P6L13: "The tails **of** freeboard…"*
*P7L19: "…**but** some…"*
*P11L31: Delete "viz"*
*P12L27: "data set" -> "data"*
*P13L37: "sector" -> "section"*

The above have been corrected as suggested.

**Responses to Reviewer 2's comments** (*in blue):*

*General comments: This is a challenging and valuable paper to attempt to map the distribution of snow depth, sea ice thickness, and ice volume for Antarctic sea ice on a hemispheric scale for the first time, by combining satellite lidar (ICESat-2) and radar (CryoSat-2) altimeters. The major motivation is to improve our understanding of the recent decreasing trend of Antarctic sea ice extents. For this purpose, the authors estimated the surface elevation with ICESat-2 and the ice freeboard with CryoSat-2 and obtained the snow depth distribution from the difference between these datasets and the ice thickness and ice volume distribution assuming isostatic balance. They also conducted the error estimates from uncertainties of various factors that contribute to the freeboard measurements. As a result, the geographical and seasonal properties of freeboard, snow depth, and ice thickness were revealed on a hemispheric for the first time. Besides, by comparting the two datasets, some unique features are suggested; such as more than 60-70% of the total freeboard is snow. It is well known that the behavior of the Antarctic sea ice extents has different characteristics from that of the Arctic sea ice extents. However, the mechanism has not been well understood due to the lack of the hemispheric scale information of the Antarctic sea ice so far. While Worby et al. (2008) showed the hemispheric ice thickness distribution of Antarctic sea ice by compiling the visual observations conducted according to the ASPeCt protocol, there has been a lot of uncertainties about the seasonality and the biases caused by the observational methods. I think many scientists have been waiting for the estimation of the hemispheric snow depth, ice thickness, and ice volume distribution based on the satellite datasets. This paper can provide a breakthrough about this topic and contain a lot of implications. Therefore, I recommend publication with minor revisions. Having said that, I have several concerns. I would appreciate it if the authors address them before publication.*

We thank the reviewer for his/her time in reviewing the manuscript and providing helpful comments for improving the revised manuscript.

*The major points are as follows:*

*1) The lack of discussion about the different footprints of the two satellite sensors. Since the distributions of snow depth and ice thickness are usually anisotropic especially at deformed ice area, I am wondering if difference in footprint might affect the results. Even though the precise discussion might be difficult, I recommend some discussion about this.*

All our calculations are performed with 25-km averages to avoid some of the pitfalls of sampling disparities between the two altimeters. We will note this in the sampling description (Section 4) to alert the reader to the potential differences due to the resolution in freeboard retrievals from IS2 and CS2.

*2) Units of parameters In the manuscript, the CGS unit (cm, g, g/cm3) and MKS unit (m, kg, kg/m3) are mixed, which might be confusing. I think it would be better to unify them to SI unit.*

Because of difference in the magnitude of snow depth and ice thickness (a factor of ten) we have consistently kept snow depths in centimeters and thicknesses in meters. We believe the community (at least the remote sensing community) generally thinks in those units.

*3) Discussions with the correlation between IS-2 and CS-2 (Fig. 4) There are several speculations about the dominant growth processes based on the correlation between the IS-2 derived and the CS-2 derived freeboards at each subsection in section 3.2 (for example, P6L39-P7L3, PL14-L19). However, I feel there are some other possible reasons for good correlations between them and the ground for their speculation is not necessarily strong. So further evidence might be needed. Please first explain in what kind situation the correlation becomes high, and then discuss the possible processes in each sector.*

The key processes that affect the variability and co-variability of the total and ice freeboards are addressed in the Section 3.1. The discussion, where we examined the contribution of the processes that affect freeboards, provided the basis for our interpretation of the time-varying IS-2 and CS-2 freeboard estimates in subsequent section.

*4) The suggestions of future field observation based on the results I would recommend the authors to suggest what kind of field observations will be required in the future to improve the accuracy of their estimations, based on their results, in the conclusion section. In the Antarctic sea ice area, there are complex snow-ice conditions, such as the presence of slush layers caused by flooding, a wide range of snow density caused by snow metamorphosis, the presence of void layers caused by deformation processes. Such suggestions would be very useful for the research community.*

Our last bullet in the conclusion highlighted the need for field and other observations for validation of these satellite data sets. The suggestions of specific observations and spacetime sampling are quite beyond the scope of the current manuscript (which already is quite lengthy). However, we added that coordinated field/remote sensing observations are needed if large-scale satellite retrievals are to be validated and made more useful to the broader community.

*Specific comments:*

*\*(P2L23-24) "the first approach. . . The second. . .. The third method. . ." Please add citation.*

These different approaches are cited beforehand, P2L20-22.

*\*(P2, section 2) Please add the footprint of each sensor. \*(P4L18) "signs indication" might be "signs indicate".*

Section 2 (page 2) describes the products used in this study. We added the size of the footprints of the sea ice retrievals in the ICESat-2 and CS-2 height estimates.

*\*(P4L19) "Snowfall adds to the snow layer" To be exact, "Snowfall precipitation minus evaporation (P-E)".*

Yes, we have clarified this in the text.

*\*(P4L20) "fvalue" might be "value"?*

Corrected.

*(P6L19-20) "Both the total and CS-2 freeboards···" What do you mean by "a balance of different processes"?*

We explain the rather low-variability of the total and radar freeboards by a balance of competing processes that affect them (thermodynamics and dynamics).

*(P6L25) "0.75x10^6 km^2" might be "0.75x10^6 km^2 per year"?*
The value of the average annual export (a 34-year average) – this has been clarified in the text.

*(P7L4) "160°W and 90°E" might be "160°E and 90°E"?*

Corrected.

*(P8L7) "one free parameter" Could you explain what this parameter means physically?*

The free parameter here is the refractive index of the medium, which is described P8L8-9. Physically, it describes the speed of light in the snow layer and, to first order, is dependent on the bulk density of snow.

*(P8L11) Please add ", respectively" after snow-ice interface".*

Corrected.

*(P9L2) What caused the uncertainty in snow density? Spatial variation, or measurement error?*

This is largely due to the spatial variability of the snow layer, which is dependent on age of the snow, the prevailing and weather conditions.

*(P9L3) What do you mean by "one free parameter"?*

The one free parameter in our approach is the snow density (that is used to compute the refractive index – see above). By free parameter we mean that all the other parameters in the equation are determined by observations and the only parameter left is the refractive index.

*(P9L5) I would recommend the authors to change the name of this subsection title to "sensitivity of the sampling frequency to calculations" or something like that. The current title might not be straightforward.*

We appreciate the reviewer's suggestion; we have added sensitivity to the title.

*(P9L30-32) I am wondering if this explanation is sufficient. I think more detailed discussion about the spatial scales of deformation and the sensor's footprint.*

See our comment above.

*(P10L6) "is likely due to.." You can add "and also smaller amount of P-E compared with other regions" The annual mean P-E distribution around the Antarctica is given by the following paper:*

*Cullather, R.I., Bromwich, D.H., and Van Woert, M.L. (1998) Spatial and temporal variability of Antarctic precipitation from atmospheric methods. Journal of Climate, 11, 334-367.*

*Toyota T., Massom R., Lecomte O., Nomura D., Heil P., Tamura T. and Fraser A.D. (2016) On the extraordinary snow on the sea ice off East Antarctica in late winter, 2012. Deep-Sea Res. II, 131, 53-67.*

We think that the largest contribution to the thin snow cover is the age of the ice produced in the polynya. Perhaps a higher order process is the smaller P-E in the region. We have cited the above as a potential explanation of the observed retrieval.

*(P10L7-8) "The spatial patterns show⋯" It might be possible that this is just because the ice-covered period becomes shorter toward the marginal ice zone. What do you think?*

Yes, this is what we meant to imply – the ice-covered period becomes progressively shorter on average towards the MIZ.

*(P10L14-15) "In all other sectors, we find. . ." The result is quite interesting. This might be a good evidence especially for the loss into leads, as suggested by the above paper.*

Yes, we agree although this is a different process compared to P-E.

*(P10L30-37) In the end, what du you think is the major reason for the negative bias?*

The discussion about the different reason for the observed biases are described P11L1-16.

*(P11L26) "by assuming that the snow depth is equal to the total (or IS-2) freeboard." Is this based on the observational facts? If so, please cite some papers which support this idea. If you can justify this assumption, it would be supportive of your results.*

There is no physical evidence that the total freeboard everywhere in the Antarctic is composed of snow. The calculation (in Equation 12) only allows us to obtain a lower bound of ice thickness by assuming that total freeboard to be equal to snow depth (i.e., lower density than ice).

*(P14L31) "an indication of total freeboard changes rather actual change in ice thickness" Then, what caused the change in freeboard?*

Due to a change in snow depth - we clarified this in the text.

*(Figure 4) "Total freeboard" might be changed to "Total (IS-2) freeboard" to avoid confusion. Please add the explanation about what the color means in Fig.4a.*

Changed and added the explanation of colors.

*(Figure 5) It is hard to detect what the color means. The color bar should be placed at the bottom of the figure.*

Modified as suggested.

*(Figure 6) Please add the explanation about what the thick solid line means.*

Added.

**The Antarctic sea ice cover from ICESat-2 and CryoSat-2: freeboard, snow depth and ice thickness**

Sahra Kacimi[1], Ron Kwok[1*]

[1]Jet Propulsion Laboratory, California Institute of Technology, Pasadena, California, USA

Correspondence to: Sahra Kacimi (sahra.kacimi@jpl.nasa.gov)

**Abstract** We offer a view of the Antarctic sea ice cover from lidar (ICESat-2) and radar (CryoSat-2) altimetry, with retrievals of freeboard, snow depth, and ice thickness that span an 8-month winter between April 1, 2019 and November 16, 2019. Snow depths are from freeboard differences. The multiyear ice observed in the West Weddell sector is the thickest with a mean sector thickness > 2 m. Thinnest ice is found near polynyas (Ross Sea and Ronne Ice Shelf) where new ice areas are exported seaward and entrained in the surrounding ice cover. For all months, the results suggest that ~65-70% of the total freeboard is comprised of snow. The remarkable mechanical convergence in coastal Amundsen Sea, associated with onshore winds, was captured by ICESat-2 and CryoSat-2. We observe a corresponding correlated increase in both freeboards, snow depth and ice thickness. While the spatial patterns in the freeboard, snow depth, and thickness composites are as expected, the observed seasonality in these variables is rather weak. This most likely results from competing processes (snowfall, snow redistribution, snow-ice formation, ice deformation, basal growth/melt) that contribute to uncorrelated changes in the total and radar freeboards. Evidence points to biases in CryoSat-2 estimates of ice freeboard of at least a few centimeters from high salinity snow (>10) in the basal layer resulting in lower/higher snow depth/ice thickness retrievals although the extent of these areas cannot be established in the current data set. Adjusting CryoSat-2 freeboards by 3/6 cm gives a circumpolar ice volume of 17,900/15,600 km³ in October, for an average thickness of ~1.29/1.13 m. Validation of Antarctic sea ice parameters remains a challenge, there are no seasonally and regionally diverse data sets that could be used to assess these large-scale satellite retrievals.

*Now at the Applied Physics Laboratory, University of Washington, Seattle, Washington, USA.

© 2020 California Institute of Technology. Government sponsorship acknowledged.

**1 Introduction**

The gradual increase in Antarctic sea ice extent in satellite records over the last four decades reversed in 2014, with subsequent rates of decrease in 2014–2019 exceeding the decay rates in the Arctic. For these past years, the Antarctic sea ice extents were reduced to their lowest levels in the 40-year satellite record (Parkinson, 2019). Our current understanding of the behavior of the Antarctic ice cover is largely informed by these ice coverage measurements from satellite passive microwave sensors. Ice extent, however, provides an incomplete picture of sea ice response to climate change and variability. But, even with the large observed changes, available measurements are still too few to be able to determine the long-term trend of ice production and volume of the of Antarctic sea ice cover. (Vaughan et al., 2013)

Prior to the 2014 decline in Antarctic ice extent, coupled ice-ocean models have suggested that significant changes in ice volume and thickness are correlated to changes in ice extent s (Massonnet et al., 2013; Holland et al., 2014), and increases in ice thickness may have been driven by the intensification of the wind field (Zhang, 2014) noted by Holland and Kwok (2012). As well, fully-coupled climate models generally fail to capture the observed trends and variability in ice coverage during the last few decades (e.g., Mahlstein et al., 2013; Polvani & Smith, 2013; Zunz et al., 2013; Hobbs et al., 2015; Turner et al., 2015). However, large-scale estimates of ice thickness and ice production necessary to improve attribution of change, model evaluation and improvements, and for projection of future behavior have been challenging to obtain. Retrievals of Antarctic ice thickness remain a research topic, largely due to uncertainties in snow depth and freeboard (Giles et al., 2008) required for computing snow loading in the conversion of freeboard to thickness.

Wide discrepancy between ice thickness estimates from recent approaches to determine sea ice thickness persists (Yi et al., 2011; Kurtz & Markus, 2012; Xie et al., 2013). Current algorithms to derive ice thickness from data collected by ICESat-1 (Ice, Cloud, and land Elevation Satellite) have to relied on the following simplifying assumptions: 1) an independent measure of snow depth (Yi et al., 2011); 2) the snow depth is equal to the total freeboard (Kurtz & Markus, 2012); or, 3) empirical relationships between total freeboard and ice thickness determined from field data (Xie et al., 2013). All these approaches have limitations. The first approach tends to underestimate of snow depth in areas of deformed ice. The second seems more appropriate for the thinner ice in the outer pack with low ice thickness. The third method may be most suitable for thicker ice, where knowledge of densities is subsumed into the regression coefficients. Such empirical relationships vary seasonally and regionally (Ozsoy-Cicek et al., 2013), and so the confidence in the derivations is reduced. Even so, these approaches have provided a large-scale depiction of the spatial variability of the ice and snow cover based on limited knowledge of the Antarctic ice cover.

With the launch of NASA's ICESat-2 (IS-2) in late 2018 and the extension of ESA's CryoSat-2 (CS-2) mission, we are now able to combine lidar and radar altimetry of the Arctic and Antarctic ice covers from IS-2 and CS-2 for understanding ice behavior. A recent paper by Kwok et al. (2020) demonstrated the retrieval of basin-scale estimates of both Arctic snow depth and sea ice thickness from differences in IS-2 and CS-2 freeboards. Here, we follow the same approaches to examine the large-scale seasonal cycle of Antarctic freeboards, retrieved snow depth and ice thickness from a joint analysis of IS-2 and CS-2 data (between April and November of 2019). At the outset, we note that the results from this study remains exploratory because of current understanding of the snow cover of Antarctic sea ice. There are many aspects of data quality, some of which will only be revealed by assessment with snow data acquired and processed by dedicated airborne campaigns (e.g., NASA's Operation IceBridge), field programs and when a longer IS-2/CS-2 time series becomes available.

The paper is organized as follows. The next section describes the IS-2 and CS-2 freeboard data sets used in our analysis. In Section 3, we first discuss the key processes that contribute to the time evolution of Antarctic freeboards, and then describe the observed evolution of the two freeboards during the eight winter months. Section 4 outlines the principle behind the derivation of snow depth from freeboard differences, the sampling of the satellite freeboards for calculation of snow depth, and the derived

monthly estimates. Section 5 compares the thickness and volume of the Antarctic ice cover computed using the derived snow depth, and assuming that snow depth is equal to the IS-2 freeboard. Potential biases in the data are discussed. Section 6 concludes the paper by the highlighting these first observations and discuss challenges in having the appropriate data sets for assessment of the retrievals from the two altimeters.

**2 Data description**

The primary data sets are freeboards from IS-2 and CS-2. Their attributes are described below.

**2.1 ICESat-2 (IS-2) freeboards**

The Advanced Topographic Laser Altimeter System (ATLAS) onboard ICESat-2 uses three beam pairs to profile the surface. The pairs are separated by about 3.3 km cross track. Each pair consists of a strong and a weak beam with an inter-beam spacing of 90m. The pulse energies of the strong beams are ~4 times that of the weak. Each beam profiles the surface at a pulse repetition rate of 10 kHz and footprints of ~14 m (Neumann et al., 2019). Along-track freeboards are from the ICESat-2 ATL10 products (Release 002) from the National Snow and Ice Data Center (Kwok et al., 2019b). The ATL10 product provides sea ice freeboard estimates – with a variable along-track resolutions (~27 to 200 m) – in 10-km segments that contain a sea surface reference. Local sea surface references ($h_{ref}$) (i.e., the estimated local sea level) are from available sea ice leads within a 10-km segment. Freeboard heights ($h_f$) are the differences between surface heights ($h_s$) and the local sea surface reference (i.e., $h_f = h_s - h_{ref}$). For individual beams, freeboard profiles are calculated with sea surface references from that beam with no dependence on estimates from other beams. In ATL10, freeboards are calculated only where the ice concentration is >50% and where the height samples are at least 25 km away from the coast (to avoid uncertainties in coastal tide corrections). Details of the sea ice algorithms can be found in Kwok et al. (2019c) and an early assessment of surface heights are in Kwok et al. (2019a). Only the freeboards from the strong beams are used in the following analyses and also cloud contaminated retrievals are not used. We note that, in the IS-2 data set used here, there is a one-month gap in coverage (July), indicated in the figures due to a spacecraft anomaly and that data are only available for the first two weeks of November 2019 in this release of the IS-2 data set. Uncertainty in IS-2 freeboard retrievals is ~2-4 cm based on assessment in Kwok et al. (2019a).

**2.2 CS-2 radar freeboards**

Along-track CS-2 freeboards are derived using the procedure in Kwok and Cunningham (2015), which contains a detailed description of the retrievals and an assessment of these freeboard estimates in the Arctic. The pulse-limited footprint of the CryoSat-2 synthetic aperture radar altimeter is approximately 0.31 km by 1.67 km along- and across-track. Freeboards are retrieved for individual returns but the derived CS-2 freeboards used here have been averaged to 25-km resolution and weighted by AMSR-derived ice concentration. As there are no large-scale assessments of these freeboard estimates, only comparisons with available ice thickness measurements from variety of sensors (e.g., upward looking sonars, airborne lidars, and airborne electromagnetic profilers, etc.) provide an indirect measure of quality. Nothing that freeboard is approximately one-ninth of ice thickness (due to the density contrast between ice and seawater) differences between CS-2 and various thickness measurements in the Arctic in Kwok and Cunningham (2015) are: 0.06±0.29 m (ice draft from moorings), 0.07±0.44 m (submarine ice draft), 0.12±0.82 m (airborne electromagnetic profiles), and -0.16±0.87 m (Operation IceBridge).

**3 IS-2 and CS-2 freeboards**

[revised manuscript text omitted]

$$h_{fs} = h_f^{IS2} - h_{fi}. \tag{2}$$

The snow depth ( $h_{fs}^{\Delta f}$ ) is then given by,

$$h_{fs}^{\Delta f} = \frac{(h_f^{IS2} - h_{fi}^{CS2})}{\eta_s}. \tag{3}$$

assuming that the scattering from the snow-ice interface dominates the returns at $K_u$-band wavelengths (CS-2 altimeter). With one free parameter, $\eta_s$, this equation relates snow depth to the IS-2 and CS-2 freeboard differences (i.e., the two observables here) – $\eta_s$ is the refractive index at $K_u$-band, $\eta_s = c/c_s(\rho_s)$ (Ulaby et al., 1986), $c$ is the speed of light in free space, and $\rho_s$ is the bulk snow density. Equation (3) accounts for the reduced propagation speed of the radar wave ($c_s$) in a snow layer with bulk density $\rho_s$. At temperatures below freezing, the lidar and radar returns can be assumed to be from the air-snow and the snow-ice interfaces respectively and thus provide observations of total and ice freeboards. The validity and shortcomings of this assumption and its implications are discussed in Section 6. A bulk snow density of 320 kg/m³ is used in all our calculations. There is no generally accepted value for the bulk density of snow in the Antarctic. Massom et al. (2001) suggest 200-300 kg/m³ under cold/dry condition and higher density (320-500 kg/m³) for warm windy conditions, which is not unlike the Arctic. Below, we elected to use an average winter bulk density of 320 kg/m³ (like that of the Arctic) but with a higher variability of 70 kg/m³ to cover the range of conditions.

$$h_{fs}^{\Delta f} = \frac{(h_f^{IS2} - h_{fi}^{CS2})}{\eta_s}. \tag{3}$$

This

**4.1.1 Sensitivity of snow depth and ice thickness to snow density**

Similarly, following *K20*, we write the sensitivity of $h_{fs}^{\Delta f}$ to bulk density (for the parameterization of $\eta_s$ given above) as:

$$\frac{\partial h_{fs}^{\Delta f}}{\partial \rho_s} = -0.77(1+0.51\times10^{-3}\rho_s)^{-2.5}(h_f^{IS2} - h_{fi}^{CS2}). \tag{4}$$

which gives the fractional change in snow depth associated with a change in density as,

$$\frac{\Delta h_{fs}^{\Delta f}}{(h_f^{IS2} - h_{fi}^{CS2})} = -0.53\times10^{-3}\Delta\rho_s \quad for \quad \rho_s = 320\,kg/m^3. \tag{5}$$

Relative to a nominal density of 320 kg/m$^3$ and an uncertainty in density of ±70 kg/m$^3$, the uncertainty in the snow depth is ~4% of the difference in freeboard. In effect, this represents ~1 cm uncertainty in snow depth for freeboard differences of 30 cm, suggesting that snow depth is relatively insensitive to uncertainties in the bulk density. The sign indicates that snow depth will be underestimated if the density is overestimated.

As well, the sensitivity of thickness estimates to uncertainties in snow density in *K20* (for a fixed total freeboard) is written as,

$$\left.\frac{\partial h_i}{\partial \rho_s}\right|_{h_f} = (h_f^{IS2} - h_{fi}^{CS2})\frac{1-0.77\eta_s^{-5/3}(\rho_s - \rho_w)}{\eta_s(\rho_w - \rho_i)}. \tag{6}$$

The fractional change in ice thickness associated with a change in density is,

$$\left.\frac{\Delta h_i}{(h_f^{IS2} - h_{fi}^{CS2})}\right|_{h_f} \sim 10.5\times10^{-3}\Delta\rho_s. \quad for \quad \rho_s = 320\,kg/m^3. \tag{7}$$

Again, relative to a nominal density of 320±70 kg/m$^3$, the calculated thickness uncertainty is ~70% of the difference in freeboards. For a 30-cm freeboard difference (typically winter value used as an example), this translates into ~0.2 m uncertainty in thickness. If the density is overestimated, the snow depth is underestimated (see above) and the ice thickness is overestimated – a larger fraction of the total freeboard is now assigned to the higher density sea ice. The above values serve as bounds on the expected density-induced errors in the retrieval estimates if a $\Delta\rho_s$ of ±70 kg/m$^3$ is indeed representative of the density variability Antarctic snow cover. In our simple model to convert freeboard differences to snow depth, the above analysis quantifies the expected sensitivity of the calculations to snow density.

**4.1.2 Sensitivity of freeboard sampling for snow depth calculations**

The sampling of the IS-2 and CS-2 freeboards for snow depth calculations follows the procedure in *K20*. Since Antarctic sea ice is found at lower latitudes, coverage is challenging due to the lower density of ground tracks from polar orbiting satellites. First, daily along-track IS-2 and CS-2 freeboards are averaged separately onto their own 25-km grid. Gridded IS-2 freeboards are averages of the three strong IS-2 beams and thus provide a better sampling of the spatial mean (compared to single-track profiles of CS-2 freeboards). Freeboard differences are then computed at each IS-2 grid cell using CS-2 freeboards (weighted by ice concentration) with time separations |ΔT|<10 days and within a 75-km box. We find that this sampling strategy provides the best spatial coverage without sacrificing precision.

We examined the sensitivity to space-time sampling (as in *K20*), by assessing differences in calculated snow depths with time separations of |ΔT| <1 day, <10 days and <15 days, using CS-2 freeboards at collocated grid cells only and then freeboards within a 75-km box (i.e., including the eight neighboring grids cells); this provides six space-time combinations. The standard

deviation of the differences in calculated snow depths (for the six combinations) were all less than 1 cm. This suggests that the spatial variability of the CS-2 freeboards is lower than IS-2 freeboards. As seen in the Section 4.2, the range of the area-averaged IS-2 freeboard between April and November (18.9 to 50.4 cm) is more than double the range of the CS-2 freeboards (6.6 to 15.6 cm). The added advantage of longer time separations and looking over longer distances for CS-2 freeboards is the improved coverage for constructing full composites. In fact, a time-separation of 10 days (i.e., |ΔT|<10 days) provides the best coverage (see Table 1).

**4.1.3 Ice deformation**

The episodic and localized nature of ice deformation and the impact of this process on differencing freeboards separated in time are discussed in *K20*. Here, we provide a brief summary. The time order of freeboard sampling has an asymmetric effect, i.e., the impact of a convergence or divergence event separating the freeboard samples would be different. If the selected CS-2 freeboard precedes an IS-2 freeboard in time, the snow depth would be overestimated (underestimated) if a convergence (divergence) event occurred in the interim. If the selected CS-2 freeboard is from a later time and a convergence (divergence) event occurred in between, the snow depths would be underestimated (overestimated). Also note is that the loss of snow during a convergence event may have a confounding effect. Here, the selected CS-2 freeboards are centered on the time of the IS-2 samples; hence, random events around that center time would increase the snow depth variance but would have a small impact on the average monthly snow depth. These results, discussed in the previous section, suggest that the effect of sea-ice deformation in biasing the snow depth estimates may be small. For the six combinations of space-time sampling of the two freeboards, the variability in retrieved snow depths were less than a centimeter.

**4.2 Snow depth estimates in 2019**

The monthly snow depth composites and their distributions are shown in Figure 3 and Figure 6a, respectively. Table 2 shows the numerical values. Due to the low variability of the CS-2 freeboards, the spatial pattern of the snow depth estimates and the IS-2 freeboards are highly correlated in all the sectors ( $\rho > 0.95$ - see Figure 7). Here, we summarize the spatial features of note. A more in-depth discussion of the relationship between snow depth and freeboard can be found in the next section and an assessment of the quality of the snow depth estimates (whether they are biased) are given the following section and Section 5, where these estimates were used to calculate ice thickness.

The thickest snow is seen in the W-Wedd sector (sector mean = $22.8 \pm 12.4$ cm in May) and the CoA-B sectors ($31.4 \pm 23.1$ cm in September). With the multiyear sea ice cover in the W-Wedd sector, thicker snow is expected. The thinnest snow is found in the Ross ($7.35 \pm 4.30$ cm in April) and E-Wedd ($8.21 \pm 5.81$ cm in June) sectors. The thinner snow depth in the Ross sector is likely due to the extensive coverage by thin/young ice exported from the active Ross Sea polynyas, and in the E-Wedd sector due to the large seasonal ice cover. Lower snowfall rates may also contribute to these results (Cullather et al., 1998; Toyota et al., 2016). The spatial patterns show consistent thinning of the snow cover towards the ice margins almost everywhere and in all months; we see no spatial anomalies in snow depth near the ice edge expected of higher precipitation. Except for coastal zones with active polynyas (e.g., southern Ross and Weddell seas), snow depth is generally higher in coastal zones.

Seasonal increases in the monthly mean snow depth are seen only in the A-B and CoA-B sectors. In the CoA-B sector, the increase is ~13 cm (approximately half that of the IS-2 freeboard increase) over the eight months. This is likely due to precipitation delivered by the on-shore wind pattern linked to the location and depth of the Amundsen Sea Low (ASL) discussed earlier. In all other sectors, we find slowly varying snow covers between April and November, similar to the observed behavior of IS-2 and CS-2 freeboards. This is quite remarkable and suggests the processes that remove snow from the surface (e.g., snow-ice transformation, loss into leads, divergence, etc.) must be significant and overwhelm all precipitation signals in all months.

**Formatted** … [11]

Consequently, an in-depth study of these processes will be important for understanding of the behavior of the Antarctic snow cover.

**4.3 Relationship between freeboard and snow depth**

$K20$ examined the relationship between freeboard and retrieved snow depth for the Arctic ice cover. This of geophysical interest as the connection could be potentially utilized to provide rough estimates of snow depths where there are gaps in CS-2 observation. Figure 7 shows the monthly scatterplots of $h_{fs}^{\Delta f}$ and Antarctic IS-2 freeboard for the eight months between April and November. At the length scale of 25 km, the regression analysis (slope, intercept, and standard error in each plot) of the monthly fields shows that the two values are highly correlated (with the freeboard explaining >95% of the variance in snow depth); this is not entirely surprising as snow depth is derived from IS-2 freeboard. The regression slopes vary between 0.66 and 0.70 between April and November. For this Antarctic winter at least, the results suggest that between 66 and 70% of the IS-2 freeboard is snow. This can be contrasted with the 2019 Arctic winter ($K20$) where snow occupies a lower fraction or ~50-55% of the IS-2 freeboard.

The negative intercepts of between -3.4 and -4.5 cm are worth noting, as one should expect –by definition – zero snow depth at near zero IS-2 freeboard. The consistent values of the monthly intercepts suggest that one of the estimates may be biased. Here, we write:

$$\hat{h}_{fs} = \alpha h_f + \beta = f(h_f) \qquad (8)$$

where $\hat{h}_{fs}$ is the snow depth estimate, and $\alpha$ and $\beta$ are the regression slope and intercept. If zero snow depth is expected at zero total freeboard, then an unbiased estimate of snow depth ($h_{fs}$) can be written as,

$$h_{fs} = \hat{h}_{fs} + \delta = f(h_f) + \delta \quad and \quad \delta = -\beta \quad if \quad h_{fs} = f(0) = 0 \qquad (9)$$

where $\delta$ is the bias. To obtain the true unbiased estimate of snow depth ($h_{fs}$), an adjustment of $\hat{h}_{fs}$ by $\delta$ (or $-\beta$) is needed. The negative intercepts observed in the scatterplots imply that $\hat{h}_{fs}$ is overestimated by +3.4 and +4.5 cm.

One likely source of these biases is the displacement of retracking point (RP) of the radar altimeter (CS-2) away from the snow-ice interface resulting in higher CS-freeboards (Kwok, 2014). At $K_u$-band frequencies (CS-2), the RP's are displaced from the true ice surface when elevated snow salinities (due to brine-wicking, flooding) are found near the snow-ice interface, or the changes in scattering the presence of moisture in the snow layer when air temperature warms (Winebrenner et al., 1994). 
[revised manuscript text omitted]
 two altimeters has been successfully implemented. We anticipate that the data acquired by CRYO2ICE will provide a crucial and valuable data set for not only understanding current retrievals but also the design of future instruments tasked to understand the development of the Arctic and Antarctic sea ice covers.

**Acknowledgments**

We thank the reviewers (Rachel Tilling and one other) for their careful reading of the submission and useful comments/suggestions, which helped improve the manuscript. We wish to thank the International Space Science Institute (ISSI) for supporting and hosting the useful workshops on Satellite Remote Sensing of Antarctic sea ice held in Bern, Switzerland over the past decade. The ASPeCt database can be accessed at https://data.aad.gov.au/aspect. AMSR ice concentration are available at https://nsidc.org/data/icesat-2/data-sets. CS-2 data are from the ESA data portal (https://earth.esa.int). The ICESat-2 ATL10 data set used herein are available at https://nsidc.org/data/icesat-2/data-sets. S.K. and R.K. carried out this work at the Jet Propulsion Laboratory, California Institute of Technology, under contract with the National Aeronautics and Space Administration.

**Table 1.** Dependence of number of retrievals on space-time separation. November is not included here because the IS-2 data (Release 002) covered only half a month.

| Space/Time | 25-km/1 day | 25-km/10 day | 25-km/15 day | 75-km/1 day | 75-km/10 day | 75-km/15 day |
|---|---|---|---|---|---|---|
| Apr | 774 | 2967 | 3476 | 2107 | 4246 | 4433 |
| May | 1023 | 4461 | 5543 | 1980 | 6898 | 7405 |
| Jun | 1413 | 5895 | 7293 | 3968 | 9243 | 9941 |
| Jul | | | | | | |
| Aug | 2108 | 9516 | 11783 | 6253 | 15782 | 16673 |
| Sep | 2073 | 8556 | 10716 | 5751 | 14581 | 15536 |
| Oct | 1818 | 8270 | 10127 | 5291 | 13125 | 13928 |

Table 2. Monthly mean (standard deviation) of IS-2 freeboard $(h_f)$, CS-2 freeboard $(h_{fi}^{CS2})$, and derived snow depth $(h_{fs}^{\Delta f})$.

| (cm) | | Apr | May | Jun | Jul | Aug | Sep | Oct | Nov |
|---|---|---|---|---|---|---|---|---|---|
| E-Wedd | $h_f$ | 25.4±10.9 | 19.0±9.72 | 15.6±8.12 | | 17.5±6.02 | 18.7±6.10 | 17.8±5.82 | 16.4±6.50 |
| | $h_{fi}^{CS2}$ | 8.37±3.14 | 7.14±2.74 | 7.00±2.16 | 6.60±2.00 | 7.07±1.86 | 7.63±2.20 | 7.36±2.16 | 5.86±2.50 |
| | $h_{fs}^{\Delta f}$ | 14.7±8.90 | 13.1±10.6 | 8.21±5.81 | | 8.90±4.30 | 9.45±4.05 | 9.24±3.96 | 8.76±4.63 |
| W-Wedd | $h_f$ | 36.5±20.3 | 41.1±19.2 | 36.2±16.9 | | 38.7±19.7 | 38.0±19.2 | 38.2±20.5 | 39.5±18.7 |
| | $h_{fi}^{CS2}$ | 11.8±4.56 | 13.5±5.73 | 12.5±5.00 | 11.5±5.43 | 11.4±5.68 | 10.8±5.05 | 11.3±5.36 | 12.4±5.03 |
| | $h_{fs}^{\Delta f}$ | 20.7±13.8 | 22.8±12.4 | 20.3±12.3 | | 22.5±14.3 | 22.5±14.1 | 22.7±16.2 | 22.1±13.0 |
| A-B | $h_f$ | 29.5±21.8 | 25.3±17.8 | 26.7±18.8 | | 31.1±23.6 | 36.3±28.5 | 32.1±23.7 | 38.2±25.0 |
| | $h_{fi}^{CS2}$ | 11.0±5.65 | 9.53±5.78 | 10.0±6.21 | 9.56±6.23 | 10.2±6.04 | 11.1±6.26 | 10.6±6.54 | 10.5±6.54 |
| | $h_{fs}^{\Delta f}$ | 17.3±14.0 | 14.8±11.0 | 16.4±14.7 | | 18.6±17.5 | 21.7±19.7 | 19.7±17.3 | 23.6±16.5 |
| CoA-B | $h_f$ | 29.7±19.2 | 29.2±16.6 | 33.6±19.3 | | 46.3±26.2 | 54.0±32.5 | 49.1±24.7 | 50.4±25.5 |
| | $h_{fi}^{CS2}$ | 11.5±5.88 | 11.2±6.03 | 13.1±7.00 | 13.3±7.46 | 14.9±6.36 | 15.3±6.68 | 15.6±6.83 | 14.6±6.73 |
| | $h_{fs}^{\Delta f}$ | 17.1±11.6 | 16.6±10.3 | 18.0±11.1 | | 27.5±19.4 | 31.4±23.1 | 30.0±18.5 | 30.8±17.0 |
| Ross | $h_f$ | 13.8±6.45 | 15.2±6.50 | 17.7±7.93 | | 21.0±10.2 | 22.7±13.1 | 22.2±12.7 | 23.0±13.6 |
| | $h_{fi}^{CS2}$ | 6.95±3.32 | 6.78±2.81 | 7.62±2.48 | 8.25±3.78 | 8.81±3.72 | 9.35±4.00 | 8.83±4.30 | 8.45±4.87 |
| | $h_{fs}^{\Delta f}$ | 7.35±4.30 | 7.50±4.21 | 9.10±5.37 | | 11.1±6.21 | 11.8±8.35 | 12.0±9.44 | 12.4±8.48 |
| Pacific | $h_f$ | 34.8±30.1 | 22.3±16.6 | 27.9±14.4 | | 26.5±16.5 | 25.7±15.7 | 27.8±18.5 | 27.0±20.0 |
| | $h_{fi}^{CS2}$ | 10.3±3.67 | 8.13±2.11 | 8.35±2.80 | 8.18±2.88 | 8.06±2.97 | 7.36±2.91 | 7.43±2.96 | 7.28±3.16 |
| | $h_{fs}^{\Delta f}$ | 24.5±23.0 | 18.4±15.3 | 19.3±13.7 | | 19.8±14.7 | 19.3±13.8 | 21.3±17.5 | 19.0±12.4 |
| Indian | $h_f$ | 27.5±22.4 | 19.0±14.1 | 25.3±26.7 | | 17.9±8.55 | 16.8±8.45 | 18.3±9.55 | 18.0±9.86 |
| | $h_{fi}^{CS2}$ | 10.1±4.00 | 7.71±2.48 | 7.46±2.55 | 7.40±2.55 | 6.74±2.06 | 7.00±2.17 | 6.85±2.55 | 5.77±2.88 |
| | $h_{fs}^{\Delta f}$ | 19.8±20.4 | 16.6±17.2 | 17.3±19.8 | | 12.0±9.51 | 9.27±6.61 | 11.0±7.60 | 10.3±6.71 |

Formatted [59]
Formatted [61]
Formatted [57]
Formatted [58]
Formatted [60]
Formatted [62]
Formatted [63]
Formatted [64]
Formatted [65]
Formatted [66]
Formatted [69]
Formatted [67]
Formatted [68]
Formatted [70]
Formatted [71]
Formatted [72]
Formatted [73]
Formatted [74]
Formatted [75]
Formatted [76]
Formatted [77]
Formatted

**Table 3.** Monthly mean (standard deviation) of estimated ice thickness: 1) $h_i$, with derived snow depth ($h_i$); 2); $h_i^0$, assuming $h_{fs} = h_f$; 3) $h_i^3$, with $\delta = 3$ cm; and, 4) $h_i^6$, with $\delta = 6$ cm.

[revised manuscript text omitted]

Font color: Auto

| Page 10: [11] Formatted | Ronald Kwok | 9/18/20 8:02:00 AM |
|---|---|---|

Font color: Auto

| Page 10: [12] Deleted | Ronald Kwok | 9/18/20 8:02:00 AM |
|---|---|---|

| Page 10: [12] Deleted | Ronald Kwok | 9/18/20 8:02:00 AM |
|---|---|---|

| Page 10: [12] Deleted | Ronald Kwok | 9/18/20 8:02:00 AM |
|---|---|---|

| Page 10: [13] Deleted | Ronald Kwok | 9/18/20 8:02:00 AM |
|---|---|---|

| Page 10: [13] Deleted | Ronald Kwok | 9/18/20 8:02:00 AM |
|---|---|---|

| Page 11: [14] Deleted | Ronald Kwok | 9/18/20 8:02:00 AM |
|---|---|---|

| Page 11: [14] Deleted | Ronald Kwok | 9/18/20 8:02:00 AM |
|---|---|---|

| Page 11: [14] Deleted | Ronald Kwok | 9/18/20 8:02:00 AM |
|---|---|---|

| Page 11: [14] Deleted | Ronald Kwok | 9/18/20 8:02:00 AM |
|---|---|---|

| Page 11: [14] Deleted | Ronald Kwok | 9/18/20 8:02:00 AM |
|---|---|---|

| Page 11: [14] Deleted | Ronald Kwok | 9/18/20 8:02:00 AM |
|---|---|---|

| Page 11: [15] Deleted | Ronald Kwok | 9/18/20 8:02:00 AM |
|---|---|---|

| Page 11: [15] Deleted | Ronald Kwok | 9/18/20 8:02:00 AM |
|---|---|---|

| Page 11: [15] Deleted | Ronald Kwok | 9/18/20 8:02:00 AM |
|---|---|---|

| Page 11: [15] Deleted | Ronald Kwok | 9/18/20 8:02:00 AM |
|---|---|---|

| Page 11: [15] Deleted | Ronald Kwok | 9/18/20 8:02:00 AM |
|---|---|---|

| Page 11: [15] Deleted | Ronald Kwok | 9/18/20 8:02:00 AM |
|---|---|---|

| Page 11: [15] Deleted | Ronald Kwok | 9/18/20 8:02:00 AM |
|---|---|---|

| Page 11: [15] Deleted | Ronald Kwok | 9/18/20 8:02:00 AM |
|---|---|---|

| Page 11: [15] Deleted | Ronald Kwok | 9/18/20 8:02:00 AM |
|---|---|---|

| Page 11: [15] Deleted | Ronald Kwok | 9/18/20 8:02:00 AM |
|---|---|---|

| Page 11: [15] Deleted | Ronald Kwok | 9/18/20 8:02:00 AM |
|---|---|---|

| Page 11: [15] Deleted | Ronald Kwok | 9/18/20 8:02:00 AM |
|---|---|---|

| Page 11: [16] Deleted | Ronald Kwok | 9/18/20 8:02:00 AM |
|---|---|---|

| Page 11: [16] Deleted | Ronald Kwok | 9/18/20 8:02:00 AM |
|---|---|---|

| Page 11: [16] Deleted | Ronald Kwok | 9/18/20 8:02:00 AM |
|---|---|---|

| Page 11: [17] Deleted | Ronald Kwok | 9/18/20 8:02:00 AM |
|---|---|---|

| Page 11: [17] Deleted | Ronald Kwok | 9/18/20 8:02:00 AM |
|---|---|---|

| Page 11: [18] Deleted | Ronald Kwok | 9/18/20 8:02:00 AM |
|---|---|---|

| Page 11: [18] Deleted | Ronald Kwok | 9/18/20 8:02:00 AM |
|---|---|---|

| Page 11: [19] Deleted | Ronald Kwok | 9/18/20 8:02:00 AM |
|---|---|---|

| Page 11: [19] Deleted | Ronald Kwok | 9/18/20 8:02:00 AM |
|---|---|---|

| Page 13: [20] Formatted | Ronald Kwok | 9/18/20 8:02:00 AM |
|---|---|---|

Font color: Auto

| Page 13: [20] Formatted | Ronald Kwok | 9/18/20 8:02:00 AM |
|---|---|---|

Font color: Auto

| Page 13: [21] Formatted | Ronald Kwok | 9/18/20 8:02:00 AM |
|---|---|---|

Font color: Auto

| Page 13: [21] Formatted | Ronald Kwok | 9/18/20 8:02:00 AM |
|---|---|---|

Font color: Auto

| Page 13: [22] Deleted | Ronald Kwok | 9/18/20 8:02:00 AM |
|---|---|---|

| Page 13: [22] Deleted | Ronald Kwok | 9/18/20 8:02:00 AM |
|---|---|---|

| Page 13: [22] Deleted | Ronald Kwok | 9/18/20 8:02:00 AM |
|---|---|---|

| Page 13: [23] Deleted | Ronald Kwok | 9/18/20 8:02:00 AM |
|---|---|---|

| Page 13: [23] Deleted | Ronald Kwok | 9/18/20 8:02:00 AM |
|---|---|---|

| Page 13: [23] Deleted | Ronald Kwok | 9/18/20 8:02:00 AM |
|---|---|---|

| Page 13: [23] Deleted | Ronald Kwok | 9/18/20 8:02:00 AM |
|---|---|---|

| Page 13: [23] Deleted | Ronald Kwok | 9/18/20 8:02:00 AM |
|---|---|---|

| Page 13: [23] Deleted | Ronald Kwok | 9/18/20 8:02:00 AM |
|---|---|---|

| Page 13: [23] Deleted | Ronald Kwok | 9/18/20 8:02:00 AM |
|---|---|---|

| Page 13: [23] Deleted | Ronald Kwok | 9/18/20 8:02:00 AM |
|---|---|---|

| Page 13: [23] Deleted | Ronald Kwok | 9/18/20 8:02:00 AM |
|---|---|---|

| Page 13: [23] Deleted | Ronald Kwok | 9/18/20 8:02:00 AM |
|---|---|---|

| Page 13: [23] Deleted | Ronald Kwok | 9/18/20 8:02:00 AM |
|---|---|---|

| Page 13: [23] Deleted | Ronald Kwok | 9/18/20 8:02:00 AM |
|---|---|---|

| Page 13: [23] Deleted | Ronald Kwok | 9/18/20 8:02:00 AM |
|---|---|---|

| Page 13: [23] Deleted | Ronald Kwok | 9/18/20 8:02:00 AM |
|---|---|---|

| Page 13: [23] Deleted | Ronald Kwok | 9/18/20 8:02:00 AM |
|---|---|---|

| Page 13: [24] Deleted | Ronald Kwok | 9/18/20 8:02:00 AM |
|---|---|---|

| Page 13: [24] Deleted | Ronald Kwok | 9/18/20 8:02:00 AM |
|---|---|---|

| Page 13: [24] Deleted | Ronald Kwok | 9/18/20 8:02:00 AM |
|---|---|---|

| Page 15: [25] Deleted | Ronald Kwok | 9/18/20 8:02:00 AM |
|---|---|---|

| Page 15: [25] Deleted | Ronald Kwok | 9/18/20 8:02:00 AM |

| Page 15: [25] Deleted | Ronald Kwok | 9/18/20 8:02:00 AM |

| Page 15: [26] Deleted | Ronald Kwok | 9/18/20 8:02:00 AM |

| Page 15: [26] Deleted | Ronald Kwok | 9/18/20 8:02:00 AM |

| Page 15: [27] Deleted | Ronald Kwok | 9/18/20 8:02:00 AM |

| Page 15: [27] Deleted | Ronald Kwok | 9/18/20 8:02:00 AM |

| Page 15: [27] Deleted | Ronald Kwok | 9/18/20 8:02:00 AM |

| Page 15: [27] Deleted | Ronald Kwok | 9/18/20 8:02:00 AM |

| Page 15: [28] Deleted | Ronald Kwok | 9/18/20 8:02:00 AM |

| Page 15: [28] Deleted | Ronald Kwok | 9/18/20 8:02:00 AM |

| Page 15: [28] Deleted | Ronald Kwok | 9/18/20 8:02:00 AM |

| Page 15: [29] Deleted | Ronald Kwok | 9/18/20 8:02:00 AM |

| Page 15: [29] Deleted | Ronald Kwok | 9/18/20 8:02:00 AM |

| Page 15: [29] Deleted | Ronald Kwok | 9/18/20 8:02:00 AM |

| Page 15: [29] Deleted | Ronald Kwok | 9/18/20 8:02:00 AM |

| Page 15: [29] Deleted | Ronald Kwok | 9/18/20 8:02:00 AM |

| Page 15: [29] Deleted | Ronald Kwok | 9/18/20 8:02:00 AM |
|---|---|---|

▼

| Page 15: [29] Deleted | Ronald Kwok | 9/18/20 8:02:00 AM |
|---|---|---|

▼

| Page 15: [29] Deleted | Ronald Kwok | 9/18/20 8:02:00 AM |
|---|---|---|

▼

| Page 15: [29] Deleted | Ronald Kwok | 9/18/20 8:02:00 AM |
|---|---|---|

▼

| Page 15: [29] Deleted | Ronald Kwok | 9/18/20 8:02:00 AM |
|---|---|---|

▼

| Page 15: [29] Deleted | Ronald Kwok | 9/18/20 8:02:00 AM |
|---|---|---|

▼

| Page 15: [30] Deleted | Ronald Kwok | 9/18/20 8:02:00 AM |
|---|---|---|

▼

.

| Page 15: [30] Deleted | Ronald Kwok | 9/18/20 8:02:00 AM |
|---|---|---|

▼

.

| Page 15: [31] Deleted | Ronald Kwok | 9/18/20 8:02:00 AM |
|---|---|---|

▼

.

| Page 15: [31] Deleted | Ronald Kwok | 9/18/20 8:02:00 AM |
|---|---|---|

▼

.

| Page 15: [32] Deleted | Ronald Kwok | 9/18/20 8:02:00 AM |
|---|---|---|

▼

.

| Page 15: [32] Deleted | Ronald Kwok | 9/18/20 8:02:00 AM |
|---|---|---|

▼

.

| Page 15: [32] Deleted | Ronald Kwok | 9/18/20 8:02:00 AM |
|---|---|---|

▼

.

| Page 18: [33] Formatted | Ronald Kwok | 9/18/20 8:02:00 AM |
|---|---|---|

Centered

| Page 18: [34] Formatted Table | Ronald Kwok | 9/18/20 8:02:00 AM |
|---|---|---|

Formatted Table

| Page 18: [35] Formatted | Ronald Kwok | 9/18/20 8:02:00 AM |

Centered

| Page 18: [36] Formatted | Ronald Kwok | 9/18/20 8:02:00 AM |

Centered

| Page 18: [37] Formatted | Ronald Kwok | 9/18/20 8:02:00 AM |

Centered

| Page 18: [38] Formatted | Ronald Kwok | 9/18/20 8:02:00 AM |

Centered

| Page 18: [39] Formatted | Ronald Kwok | 9/18/20 8:02:00 AM |

Centered

| Page 18: [40] Formatted | Ronald Kwok | 9/18/20 8:02:00 AM |

Font color: Text 1

| Page 18: [41] Formatted | Ronald Kwok | 9/18/20 8:02:00 AM |

Centered

| Page 18: [42] Formatted | Ronald Kwok | 9/18/20 8:02:00 AM |

Font color: Text 1

| Page 18: [43] Formatted | Ronald Kwok | 9/18/20 8:02:00 AM |

Font color: Text 1

| Page 18: [44] Formatted | Ronald Kwok | 9/18/20 8:02:00 AM |

Font color: Text 1

| Page 18: [45] Formatted | Ronald Kwok | 9/18/20 8:02:00 AM |

Font color: Text 1

| Page 18: [46] Formatted | Ronald Kwok | 9/18/20 8:02:00 AM |

Font color: Text 1

| Page 18: [47] Formatted | Ronald Kwok | 9/18/20 8:02:00 AM |

Font color: Text 1

| Page 18: [48] Formatted | Ronald Kwok | 9/18/20 8:02:00 AM |

Font color: Text 1

| Page 18: [49] Formatted | Ronald Kwok | 9/18/20 8:02:00 AM |

Centered

| Page 18: [50] Formatted | Ronald Kwok | 9/18/20 8:02:00 AM |
|---|---|---|

Font color: Text 1

| Page 18: [51] Formatted | Ronald Kwok | 9/18/20 8:02:00 AM |
|---|---|---|

Font color: Text 1

| Page 18: [52] Formatted | Ronald Kwok | 9/18/20 8:02:00 AM |
|---|---|---|

Font color: Text 1

| Page 18: [53] Formatted | Ronald Kwok | 9/18/20 8:02:00 AM |
|---|---|---|

Font color: Text 1

| Page 18: [54] Formatted | Ronald Kwok | 9/18/20 8:02:00 AM |
|---|---|---|

Font color: Text 1

| Page 18: [55] Formatted | Ronald Kwok | 9/18/20 8:02:00 AM |
|---|---|---|

Font color: Text 1

| Page 19: [56] Deleted | Ronald Kwok | 9/18/20 8:02:00 AM |
|---|---|---|

| Page 19: [56] Deleted | Ronald Kwok | 9/18/20 8:02:00 AM |
|---|---|---|

| Page 19: [57] Formatted | Ronald Kwok | 9/18/20 8:02:00 AM |
|---|---|---|

Font color: Black

| Page 19: [57] Formatted | Ronald Kwok | 9/18/20 8:02:00 AM |
|---|---|---|

Font color: Black

| Page 19: [58] Formatted | Ronald Kwok | 9/18/20 8:02:00 AM |
|---|---|---|

Font color: Black

| Page 19: [58] Formatted | Ronald Kwok | 9/18/20 8:02:00 AM |
|---|---|---|

Font color: Black

| Page 19: [59] Formatted | Ronald Kwok | 9/18/20 8:02:00 AM |
|---|---|---|

Centered

| Page 19: [60] Formatted | Ronald Kwok | 9/18/20 8:02:00 AM |
|---|---|---|

Centered

| Page 19: [61] Formatted | Ronald Kwok | 9/18/20 8:02:00 AM |
|---|---|---|

Font color: Black

| Page 19: [62] Formatted | Ronald Kwok | 9/18/20 8:02:00 AM |
|---|---|---|

Font color: Black

| Page 19: [63] Formatted | Ronald Kwok | 9/18/20 8:02:00 AM |
|---|---|---|

Font color: Black

| Page 19: [64] Formatted | Ronald Kwok | 9/18/20 8:02:00 AM |
|---|---|---|

Font color: Black

| Page 19: [65] Formatted | Ronald Kwok | 9/18/20 8:02:00 AM |
|---|---|---|

Centered

| Page 19: [66] Formatted | Ronald Kwok | 9/18/20 8:02:00 AM |
|---|---|---|

Font color: Black

| Page 19: [67] Formatted | Ronald Kwok | 9/18/20 8:02:00 AM |
|---|---|---|

Centered

| Page 19: [68] Formatted | Ronald Kwok | 9/18/20 8:02:00 AM |
|---|---|---|

Font color: Black

| Page 19: [69] Formatted | Ronald Kwok | 9/18/20 8:02:00 AM |
|---|---|---|

Font color: Black

| Page 19: [70] Formatted | Ronald Kwok | 9/18/20 8:02:00 AM |
|---|---|---|

Font color: Black

| Page 19: [71] Formatted | Ronald Kwok | 9/18/20 8:02:00 AM |
|---|---|---|

Font color: Black

| Page 19: [72] Formatted | Ronald Kwok | 9/18/20 8:02:00 AM |
|---|---|---|

Font color: Black

| Page 19: [73] Formatted | Ronald Kwok | 9/18/20 8:02:00 AM |
|---|---|---|

Font color: Black

| Page 19: [74] Formatted | Ronald Kwok | 9/18/20 8:02:00 AM |
|---|---|---|

Centered

| Page 19: [75] Formatted | Ronald Kwok | 9/18/20 8:02:00 AM |
|---|---|---|

Font color: Black

| Page 19: [76] Formatted | Ronald Kwok | 9/18/20 8:02:00 AM |
|---|---|---|

Centered

| Page 19: [77] Formatted | Ronald Kwok | 9/18/20 8:02:00 AM |
|---|---|---|

Font color: Black

| Page 19: [78] Formatted | Ronald Kwok | 9/18/20 8:02:00 AM |
|---|---|---|

Font color: Black

| Page 19: [79] Formatted | Ronald Kwok | 9/18/20 8:02:00 AM |
|---|---|---|

Font color: Black

| Page 19: [80] Formatted | Ronald Kwok | 9/18/20 8:02:00 AM |
|---|---|---|

Centered

| Page 19: [81] Formatted | Ronald Kwok | 9/18/20 8:02:00 AM |
|---|---|---|

Centered

| Page 19: [82] Formatted | Ronald Kwok | 9/18/20 8:02:00 AM |
|---|---|---|

Font color: Black

| Page 19: [83] Formatted | Ronald Kwok | 9/18/20 8:02:00 AM |
|---|---|---|

Font color: Black

| Page 19: [84] Formatted | Ronald Kwok | 9/18/20 8:02:00 AM |
|---|---|---|

Font color: Black

| Page 19: [85] Formatted | Ronald Kwok | 9/18/20 8:02:00 AM |
|---|---|---|

Font color: Black

| Page 19: [86] Formatted | Ronald Kwok | 9/18/20 8:02:00 AM |
|---|---|---|

Centered

| Page 19: [87] Formatted | Ronald Kwok | 9/18/20 8:02:00 AM |
|---|---|---|

Font color: Black

| Page 19: [88] Formatted | Ronald Kwok | 9/18/20 8:02:00 AM |
|---|---|---|

Centered

| Page 19: [89] Formatted | Ronald Kwok | 9/18/20 8:02:00 AM |
|---|---|---|

Centered

| Page 19: [90] Formatted | Ronald Kwok | 9/18/20 8:02:00 AM |
|---|---|---|

Pattern: Clear

| Page 19: [91] Formatted | Ronald Kwok | 9/18/20 8:02:00 AM |

Centered

| Page 19: [92] Formatted | Ronald Kwok | 9/18/20 8:02:00 AM |

Font color: Black

| Page 19: [93] Formatted | Ronald Kwok | 9/18/20 8:02:00 AM |

Font color: Black

| Page 19: [94] Formatted | Ronald Kwok | 9/18/20 8:02:00 AM |

Font color: Black

| Page 19: [95] Formatted | Ronald Kwok | 9/18/20 8:02:00 AM |

Centered

| Page 19: [96] Formatted | Ronald Kwok | 9/18/20 8:02:00 AM |

Font color: Black

| Page 19: [97] Formatted | Ronald Kwok | 9/18/20 8:02:00 AM |

Centered

| Page 19: [98] Formatted | Ronald Kwok | 9/18/20 8:02:00 AM |

Font color: Black

| Page 19: [99] Formatted | Ronald Kwok | 9/18/20 8:02:00 AM |

Font color: Black

| Page 19: [100] Formatted | Ronald Kwok | 9/18/20 8:02:00 AM |

Font color: Black

| Page 19: [101] Formatted | Ronald Kwok | 9/18/20 8:02:00 AM |

Centered

| Page 19: [102] Formatted | Ronald Kwok | 9/18/20 8:02:00 AM |

Centered

| Page 19: [103] Formatted | Ronald Kwok | 9/18/20 8:02:00 AM |

Font color: Black

| Page 19: [104] Formatted | Ronald Kwok | 9/18/20 8:02:00 AM |

Font color: Black

| Page 19: [105] Formatted | Ronald Kwok | 9/18/20 8:02:00 AM |

Font color: Black

| Page 19: [106] Formatted | Ronald Kwok | 9/18/20 8:02:00 AM |
|---|---|---|

Font color: Black

| Page 19: [107] Formatted | Ronald Kwok | 9/18/20 8:02:00 AM |
|---|---|---|

Font color: Black

| Page 19: [108] Formatted | Ronald Kwok | 9/18/20 8:02:00 AM |
|---|---|---|

Font color: Black

| Page 19: [109] Formatted | Ronald Kwok | 9/18/20 8:02:00 AM |
|---|---|---|

Centered

| Page 19: [110] Formatted | Ronald Kwok | 9/18/20 8:02:00 AM |
|---|---|---|

Font color: Black

| Page 19: [111] Formatted | Ronald Kwok | 9/18/20 8:02:00 AM |
|---|---|---|

Font color: Black

| Page 19: [112] Formatted | Ronald Kwok | 9/18/20 8:02:00 AM |
|---|---|---|

Centered

| Page 19: [113] Formatted | Ronald Kwok | 9/18/20 8:02:00 AM |
|---|---|---|

Font color: Black

| Page 19: [114] Formatted | Ronald Kwok | 9/18/20 8:02:00 AM |
|---|---|---|

Font color: Black

| Page 19: [115] Formatted | Ronald Kwok | 9/18/20 8:02:00 AM |
|---|---|---|

Font color: Black

| Page 19: [116] Formatted | Ronald Kwok | 9/18/20 8:02:00 AM |
|---|---|---|

Font color: Black

| Page 19: [117] Formatted | Ronald Kwok | 9/18/20 8:02:00 AM |
|---|---|---|

Font color: Black

| Page 19: [118] Formatted | Ronald Kwok | 9/18/20 8:02:00 AM |
|---|---|---|

Centered

| Page 19: [119] Formatted | Ronald Kwok | 9/18/20 8:02:00 AM |
|---|---|---|

Font color: Black

| Page 19: [120] Formatted | Ronald Kwok | 9/18/20 8:02:00 AM |
|---|---|---|

Centered

| Page 19: [121] Formatted | Ronald Kwok | 9/18/20 8:02:00 AM |
|---|---|---|

Font color: Black

| Page 19: [122] Formatted | Ronald Kwok | 9/18/20 8:02:00 AM |
|---|---|---|

Centered

| Page 19: [123] Formatted | Ronald Kwok | 9/18/20 8:02:00 AM |
|---|---|---|

Centered

| Page 19: [124] Formatted | Ronald Kwok | 9/18/20 8:02:00 AM |
|---|---|---|

Font color: Black

| Page 19: [125] Formatted | Ronald Kwok | 9/18/20 8:02:00 AM |
|---|---|---|

Font color: Black

| Page 19: [126] Formatted | Ronald Kwok | 9/18/20 8:02:00 AM |
|---|---|---|

Font color: Black

| Page 19: [127] Formatted | Ronald Kwok | 9/18/20 8:02:00 AM |
|---|---|---|

Font color: Black

| Page 19: [128] Formatted | Ronald Kwok | 9/18/20 8:02:00 AM |
|---|---|---|

Font color: Black

| Page 19: [129] Formatted | Ronald Kwok | 9/18/20 8:02:00 AM |
|---|---|---|

Font color: Black

| Page 19: [130] Formatted | Ronald Kwok | 9/18/20 8:02:00 AM |
|---|---|---|

Centered

| Page 19: [131] Formatted | Ronald Kwok | 9/18/20 8:02:00 AM |
|---|---|---|

Centered

| Page 19: [132] Formatted | Ronald Kwok | 9/18/20 8:02:00 AM |
|---|---|---|

Font color: Black

| Page 19: [132] Formatted | Ronald Kwok | 9/18/20 8:02:00 AM |
|---|---|---|

Font color: Black

| Page 19: [133] Formatted | Ronald Kwok | 9/18/20 8:02:00 AM |
|---|---|---|

Font color: Black

| Page 19: [134] Formatted | Ronald Kwok | 9/18/20 8:02:00 AM |

Centered

| Page 19: [135] Formatted | Ronald Kwok | 9/18/20 8:02:00 AM |

Centered

| Page 19: [136] Formatted | Ronald Kwok | 9/18/20 8:02:00 AM |

Centered

| Page 19: [137] Formatted | Ronald Kwok | 9/18/20 8:02:00 AM |

Centered

| Page 19: [138] Formatted | Ronald Kwok | 9/18/20 8:02:00 AM |

Font color: Black

| Page 19: [139] Formatted | Ronald Kwok | 9/18/20 8:02:00 AM |

Font color: Black

| Page 19: [140] Formatted | Ronald Kwok | 9/18/20 8:02:00 AM |

Font color: Black

| Page 19: [141] Formatted | Ronald Kwok | 9/18/20 8:02:00 AM |

Font color: Black

| Page 19: [142] Formatted | Ronald Kwok | 9/18/20 8:02:00 AM |

Font color: Black

| Page 19: [143] Formatted | Ronald Kwok | 9/18/20 8:02:00 AM |

Font color: Black

| Page 19: [144] Formatted | Ronald Kwok | 9/18/20 8:02:00 AM |

Centered

| Page 19: [145] Formatted | Ronald Kwok | 9/18/20 8:02:00 AM |

Font color: Black

| Page 19: [146] Formatted | Ronald Kwok | 9/18/20 8:02:00 AM |

Font color: Black

| Page 19: [147] Formatted | Ronald Kwok | 9/18/20 8:02:00 AM |

Centered

| Page 19: [148] Formatted | Ronald Kwok | 9/18/20 8:02:00 AM |

Font color: Black

| Page 19: [149] Formatted | Ronald Kwok | 9/18/20 8:02:00 AM |

Font color: Black

| Page 19: [150] Formatted | Ronald Kwok | 9/18/20 8:02:00 AM |

Font color: Black

| Page 19: [151] Formatted | Ronald Kwok | 9/18/20 8:02:00 AM |

Font color: Black

| Page 19: [152] Formatted | Ronald Kwok | 9/18/20 8:02:00 AM |

Font color: Black

| Page 19: [153] Formatted | Ronald Kwok | 9/18/20 8:02:00 AM |

Font color: Black

| Page 19: [154] Formatted | Ronald Kwok | 9/18/20 8:02:00 AM |

Font color: Black

| Page 19: [155] Formatted | Ronald Kwok | 9/18/20 8:02:00 AM |

Centered

| Page 19: [156] Formatted | Ronald Kwok | 9/18/20 8:02:00 AM |

Font color: Black

| Page 19: [157] Formatted | Ronald Kwok | 9/18/20 8:02:00 AM |

Font color: Black

| Page 19: [158] Formatted | Ronald Kwok | 9/18/20 8:02:00 AM |

Centered

| Page 19: [159] Formatted | Ronald Kwok | 9/18/20 8:02:00 AM |

Centered

| Page 19: [160] Formatted | Ronald Kwok | 9/18/20 8:02:00 AM |

Centered

| Page 19: [161] Formatted | Ronald Kwok | 9/18/20 8:02:00 AM |

Font color: Black

| Page 19: [162] Formatted | Ronald Kwok | 9/18/20 8:02:00 AM |

Font color: Black

| Page 19: [163] Formatted | Ronald Kwok | 9/18/20 8:02:00 AM |
|---|---|---|

Font color: Black

| Page 19: [164] Formatted | Ronald Kwok | 9/18/20 8:02:00 AM |
|---|---|---|

Font color: Black

| Page 19: [165] Formatted | Ronald Kwok | 9/18/20 8:02:00 AM |
|---|---|---|

Font color: Black

| Page 19: [166] Formatted | Ronald Kwok | 9/18/20 8:02:00 AM |
|---|---|---|

Font color: Black

| Page 19: [167] Formatted | Ronald Kwok | 9/18/20 8:02:00 AM |
|---|---|---|

Centered

| Page 19: [168] Formatted | Ronald Kwok | 9/18/20 8:02:00 AM |
|---|---|---|

Font color: Black

| Page 19: [169] Formatted | Ronald Kwok | 9/18/20 8:02:00 AM |
|---|---|---|

Centered

| Page 19: [170] Formatted | Ronald Kwok | 9/18/20 8:02:00 AM |
|---|---|---|

Font color: Black

| Page 19: [171] Formatted | Ronald Kwok | 9/18/20 8:02:00 AM |
|---|---|---|

Font color: Black

| Page 19: [172] Formatted | Ronald Kwok | 9/18/20 8:02:00 AM |
|---|---|---|

Centered

| Page 19: [173] Formatted | Ronald Kwok | 9/18/20 8:02:00 AM |
|---|---|---|

Centered

| Page 19: [174] Formatted | Ronald Kwok | 9/18/20 8:02:00 AM |
|---|---|---|

Font color: Black

| Page 19: [175] Formatted | Ronald Kwok | 9/18/20 8:02:00 AM |
|---|---|---|

Centered

| Page 19: [176] Formatted | Ronald Kwok | 9/18/20 8:02:00 AM |
|---|---|---|

Centered

| Page 19: [177] Formatted | Ronald Kwok | 9/18/20 8:02:00 AM |
|---|---|---|

Font color: Black

| Page 19: [178] Formatted | Ronald Kwok | 9/18/20 8:02:00 AM |

Font color: Black

| Page 19: [179] Formatted | Ronald Kwok | 9/18/20 8:02:00 AM |

Font color: Black

| Page 19: [180] Formatted | Ronald Kwok | 9/18/20 8:02:00 AM |

Font color: Black

| Page 19: [181] Formatted | Ronald Kwok | 9/18/20 8:02:00 AM |

Centered

| Page 19: [182] Formatted | Ronald Kwok | 9/18/20 8:02:00 AM |

Font color: Black

| Page 19: [183] Formatted | Ronald Kwok | 9/18/20 8:02:00 AM |

Font color: Black

| Page 19: [184] Formatted | Ronald Kwok | 9/18/20 8:02:00 AM |

Font color: Black

| Page 19: [185] Formatted | Ronald Kwok | 9/18/20 8:02:00 AM |

Centered

| Page 19: [186] Formatted | Ronald Kwok | 9/18/20 8:02:00 AM |

Font color: Black

| Page 19: [187] Formatted | Ronald Kwok | 9/18/20 8:02:00 AM |

Font color: Black

| Page 19: [188] Formatted | Ronald Kwok | 9/18/20 8:02:00 AM |

Font color: Black

| Page 19: [189] Formatted | Ronald Kwok | 9/18/20 8:02:00 AM |

Font color: Black

| Page 19: [190] Formatted | Ronald Kwok | 9/18/20 8:02:00 AM |

Font color: Black

| Page 19: [191] Formatted | Ronald Kwok | 9/18/20 8:02:00 AM |

Centered

| Page 19: [192] Formatted | Ronald Kwok | 9/18/20 8:02:00 AM |

Font color: Black

| Page 19: [193] Formatted | Ronald Kwok | 9/18/20 8:02:00 AM |
|---|---|---|

Centered

| Page 19: [194] Formatted | Ronald Kwok | 9/18/20 8:02:00 AM |
|---|---|---|

Font color: Black

| Page 19: [195] Formatted | Ronald Kwok | 9/18/20 8:02:00 AM |
|---|---|---|

Font color: Black

| Page 20: [196] Formatted | Ronald Kwok | 9/18/20 8:02:00 AM |
|---|---|---|

Font color: Black

| Page 20: [197] Formatted | Ronald Kwok | 9/18/20 8:02:00 AM |
|---|---|---|

Font color: Black

| Page 20: [198] Formatted | Ronald Kwok | 9/18/20 8:02:00 AM |
|---|---|---|

Font color: Black

| Page 20: [198] Formatted | Ronald Kwok | 9/18/20 8:02:00 AM |
|---|---|---|

Font color: Black

| Page 20: [199] Formatted | Ronald Kwok | 9/18/20 8:02:00 AM |
|---|---|---|

Line spacing:  single

| Page 20: [200] Deleted | Ronald Kwok | 9/18/20 8:02:00 AM |
|---|---|---|

| Page 20: [200] Deleted | Ronald Kwok | 9/18/20 8:02:00 AM |
|---|---|---|

| Page 20: [201] Deleted | Ronald Kwok | 9/18/20 8:02:00 AM |
|---|---|---|

| Page 20: [201] Deleted | Ronald Kwok | 9/18/20 8:02:00 AM |
|---|---|---|

| Page 20: [202] Formatted | Ronald Kwok | 9/18/20 8:02:00 AM |
|---|---|---|

Line spacing:  single

| Page 20: [203] Deleted | Ronald Kwok | 9/18/20 8:02:00 AM |
|---|---|---|

| Page 20: [203] Deleted | Ronald Kwok | 9/18/20 8:02:00 AM |
|---|---|---|

| Page 20: [204] Formatted | Ronald Kwok | 9/18/20 8:02:00 AM |
|---|---|---|

Font color: Black

| Page 20: [205] Formatted | Ronald Kwok | 9/18/20 8:02:00 AM |
|---|---|---|

Font color: Black

| Page 20: [206] Formatted | Ronald Kwok | 9/18/20 8:02:00 AM |
|---|---|---|

Font color: Black

| Page 20: [207] Formatted | Ronald Kwok | 9/18/20 8:02:00 AM |
|---|---|---|

Font color: Black

| Page 20: [208] Formatted | Ronald Kwok | 9/18/20 8:02:00 AM |
|---|---|---|

Font color: Black

| Page 20: [209] Formatted | Ronald Kwok | 9/18/20 8:02:00 AM |
|---|---|---|

Font color: Black

| Page 20: [210] Formatted | Ronald Kwok | 9/18/20 8:02:00 AM |
|---|---|---|

Font color: Black

| Page 20: [211] Formatted | Ronald Kwok | 9/18/20 8:02:00 AM |
|---|---|---|

Font color: Black

| Page 20: [212] Formatted | Ronald Kwok | 9/18/20 8:02:00 AM |
|---|---|---|

Font color: Black

| Page 20: [213] Formatted | Ronald Kwok | 9/18/20 8:02:00 AM |
|---|---|---|

Font color: Black

| Page 20: [214] Formatted | Ronald Kwok | 9/18/20 8:02:00 AM |
|---|---|---|

Font color: Black

| Page 20: [215] Formatted | Ronald Kwok | 9/18/20 8:02:00 AM |
|---|---|---|

Font color: Black

| Page 20: [216] Formatted | Ronald Kwok | 9/18/20 8:02:00 AM |
|---|---|---|

Font color: Black

| Page 20: [217] Formatted | Ronald Kwok | 9/18/20 8:02:00 AM |
|---|---|---|

Font color: Black

| Page 20: [218] Formatted | Ronald Kwok | 9/18/20 8:02:00 AM |
|---|---|---|

Font color: Black

| Page 20: [219] Formatted | Ronald Kwok | 9/18/20 8:02:00 AM |
|---|---|---|

Font color: Black

| Page 20: [220] Formatted | Ronald Kwok | 9/18/20 8:02:00 AM |
|---|---|---|

Font color: Black

| Page 20: [221] Formatted | Ronald Kwok | 9/18/20 8:02:00 AM |
|---|---|---|

Font color: Black

| Page 20: [222] Formatted | Ronald Kwok | 9/18/20 8:02:00 AM |
|---|---|---|

Font color: Black

| Page 20: [223] Formatted | Ronald Kwok | 9/18/20 8:02:00 AM |
|---|---|---|

Font color: Black

| Page 20: [224] Formatted | Ronald Kwok | 9/18/20 8:02:00 AM |
|---|---|---|

Font color: Black

| Page 20: [225] Formatted | Ronald Kwok | 9/18/20 8:02:00 AM |
|---|---|---|

Font color: Black

| Page 20: [226] Formatted | Ronald Kwok | 9/18/20 8:02:00 AM |
|---|---|---|

Font color: Black

| Page 20: [227] Formatted | Ronald Kwok | 9/18/20 8:02:00 AM |
|---|---|---|

Font color: Black

| Page 20: [228] Formatted | Ronald Kwok | 9/18/20 8:02:00 AM |
|---|---|---|

Font color: Black

| Page 20: [229] Formatted | Ronald Kwok | 9/18/20 8:02:00 AM |
|---|---|---|

Font color: Black

| Page 20: [230] Formatted | Ronald Kwok | 9/18/20 8:02:00 AM |
|---|---|---|

Font color: Black

| Page 20: [231] Formatted | Ronald Kwok | 9/18/20 8:02:00 AM |
|---|---|---|

Font color: Black

| Page 20: [232] Formatted | Ronald Kwok | 9/18/20 8:02:00 AM |
|---|---|---|

Font color: Black

| Page 20: [233] Formatted | Ronald Kwok | 9/18/20 8:02:00 AM |
|---|---|---|

Font color: Black

| Page 20: [234] Formatted | Ronald Kwok | 9/18/20 8:02:00 AM |
|---|---|---|

Font color: Black

| Page 20: [235] Formatted | Ronald Kwok | 9/18/20 8:02:00 AM |
|---|---|---|

Font color: Black

| Page 20: [236] Formatted | Ronald Kwok | 9/18/20 8:02:00 AM |
|---|---|---|

Font color: Black

| Page 20: [237] Formatted | Ronald Kwok | 9/18/20 8:02:00 AM |
|---|---|---|

Font color: Black

| Page 20: [238] Formatted | Ronald Kwok | 9/18/20 8:02:00 AM |
|---|---|---|

Font color: Black

| Page 20: [239] Formatted | Ronald Kwok | 9/18/20 8:02:00 AM |
|---|---|---|

Font color: Black

| Page 20: [240] Formatted | Ronald Kwok | 9/18/20 8:02:00 AM |
|---|---|---|

Font color: Black

| Page 20: [241] Formatted | Ronald Kwok | 9/18/20 8:02:00 AM |
|---|---|---|

Font color: Black

| Page 20: [242] Formatted | Ronald Kwok | 9/18/20 8:02:00 AM |
|---|---|---|

Font color: Black

| Page 20: [242] Formatted | Ronald Kwok | 9/18/20 8:02:00 AM |
|---|---|---|

Font color: Black

| Page 20: [243] Formatted | Ronald Kwok | 9/18/20 8:02:00 AM |
|---|---|---|

Font color: Black

| Page 20: [243] Formatted | Ronald Kwok | 9/18/20 8:02:00 AM |
|---|---|---|

Font color: Black

| Page 20: [244] Formatted | Ronald Kwok | 9/18/20 8:02:00 AM |
|---|---|---|

Font color: Black

| Page 20: [245] Formatted | Ronald Kwok | 9/18/20 8:02:00 AM |
|---|---|---|

Font color: Black

| Page 20: [246] Formatted | Ronald Kwok | 9/18/20 8:02:00 AM |
|---|---|---|

Font color: Black

| Page 20: [247] Formatted | Ronald Kwok | 9/18/20 8:02:00 AM |
|---|---|---|

Font color: Black

| Page 20: [248] Formatted | Ronald Kwok | 9/18/20 8:02:00 AM |
|---|---|---|

Font color: Black

| Page 20: [249] Formatted | Ronald Kwok | 9/18/20 8:02:00 AM |
|---|---|---|

Font color: Black

| Page 20: [250] Formatted | Ronald Kwok | 9/18/20 8:02:00 AM |
|---|---|---|

Font color: Black

| Page 20: [251] Formatted | Ronald Kwok | 9/18/20 8:02:00 AM |
|---|---|---|

Font color: Black

| Page 20: [252] Formatted | Ronald Kwok | 9/18/20 8:02:00 AM |
|---|---|---|

Font color: Black

| Page 20: [253] Formatted | Ronald Kwok | 9/18/20 8:02:00 AM |
|---|---|---|

Font color: Black

| Page 20: [254] Formatted | Ronald Kwok | 9/18/20 8:02:00 AM |
|---|---|---|

Font color: Black

| Page 20: [255] Formatted | Ronald Kwok | 9/18/20 8:02:00 AM |
|---|---|---|

Font color: Black

| Page 20: [256] Formatted | Ronald Kwok | 9/18/20 8:02:00 AM |
|---|---|---|

Font color: Black

| Page 20: [257] Formatted | Ronald Kwok | 9/18/20 8:02:00 AM |
|---|---|---|

Font color: Black

| Page 20: [258] Formatted | Ronald Kwok | 9/18/20 8:02:00 AM |
|---|---|---|

Font color: Black

| Page 20: [259] Formatted | Ronald Kwok | 9/18/20 8:02:00 AM |
|---|---|---|

Font color: Black

| Page 20: [260] Formatted | Ronald Kwok | 9/18/20 8:02:00 AM |
|---|---|---|

Font color: Black

| Page 20: [261] Formatted | Ronald Kwok | 9/18/20 8:02:00 AM |
|---|---|---|

Font color: Black

| Page 20: [262] Formatted | Ronald Kwok | 9/18/20 8:02:00 AM |
|---|---|---|

Font color: Black

| Page 20: [263] Formatted | Ronald Kwok | 9/18/20 8:02:00 AM |
|---|---|---|

Font color: Black

| Page 20: [264] Formatted | Ronald Kwok | 9/18/20 8:02:00 AM |
|---|---|---|

Font color: Black

| Page 20: [265] Formatted | Ronald Kwok | 9/18/20 8:02:00 AM |
|---|---|---|

Font color: Black

| Page 20: [266] Formatted | Ronald Kwok | 9/18/20 8:02:00 AM |
|---|---|---|

Font color: Black

| Page 20: [267] Formatted | Ronald Kwok | 9/18/20 8:02:00 AM |
|---|---|---|

Font color: Black

| Page 20: [268] Formatted | Ronald Kwok | 9/18/20 8:02:00 AM |
|---|---|---|

Font color: Black

| Page 20: [269] Formatted | Ronald Kwok | 9/18/20 8:02:00 AM |
|---|---|---|

Font color: Black

| Page 20: [270] Formatted | Ronald Kwok | 9/18/20 8:02:00 AM |
|---|---|---|

Font color: Black

| Page 20: [271] Formatted | Ronald Kwok | 9/18/20 8:02:00 AM |
|---|---|---|

Font color: Black

| Page 20: [272] Formatted | Ronald Kwok | 9/18/20 8:02:00 AM |
|---|---|---|

Font color: Black

| Page 20: [273] Formatted | Ronald Kwok | 9/18/20 8:02:00 AM |
|---|---|---|

Font color: Black

| Page 20: [274] Formatted | Ronald Kwok | 9/18/20 8:02:00 AM |
|---|---|---|

Font color: Black

| Page 20: [275] Formatted | Ronald Kwok | 9/18/20 8:02:00 AM |
|---|---|---|

Font color: Black

| Page 20: [276] Formatted | Ronald Kwok | 9/18/20 8:02:00 AM |
|---|---|---|

Font color: Black

| Page 20: [277] Formatted | Ronald Kwok | 9/18/20 8:02:00 AM |
|---|---|---|

Font color: Black

| Page 20: [278] Formatted | Ronald Kwok | 9/18/20 8:02:00 AM |
|---|---|---|

Font color: Black

| Page 20: [279] Formatted | Ronald Kwok | 9/18/20 8:02:00 AM |
|---|---|---|

Font color: Black

| Page 20: [280] Formatted | Ronald Kwok | 9/18/20 8:02:00 AM |
|---|---|---|

Font color: Black

| Page 20: [280] Formatted | Ronald Kwok | 9/18/20 8:02:00 AM |
|---|---|---|

Font color: Black

| Page 20: [281] Formatted | Ronald Kwok | 9/18/20 8:02:00 AM |
|---|---|---|

Font color: Black

| Page 20: [282] Formatted | Ronald Kwok | 9/18/20 8:02:00 AM |
|---|---|---|

Font color: Black

| Page 20: [283] Formatted | Ronald Kwok | 9/18/20 8:02:00 AM |
|---|---|---|

Font color: Black

| Page 20: [284] Formatted | Ronald Kwok | 9/18/20 8:02:00 AM |
|---|---|---|

Font color: Black

| Page 20: [285] Formatted | Ronald Kwok | 9/18/20 8:02:00 AM |
|---|---|---|

Font color: Black

| Page 20: [286] Formatted | Ronald Kwok | 9/18/20 8:02:00 AM |
|---|---|---|

Font color: Black

| Page 20: [287] Formatted | Ronald Kwok | 9/18/20 8:02:00 AM |
|---|---|---|

Font color: Black

| Page 20: [288] Formatted | Ronald Kwok | 9/18/20 8:02:00 AM |
|---|---|---|

Font color: Black

| Page 20: [289] Formatted | Ronald Kwok | 9/18/20 8:02:00 AM |
|---|---|---|

Font color: Black

| Page 20: [290] Formatted | Ronald Kwok | 9/18/20 8:02:00 AM |
|---|---|---|

Font color: Black

| Page 20: [291] Formatted | Ronald Kwok | 9/18/20 8:02:00 AM |
|---|---|---|

Font color: Black

| Page 20: [292] Formatted | Ronald Kwok | 9/18/20 8:02:00 AM |
|---|---|---|

Font color: Black

| Page 20: [293] Formatted | Ronald Kwok | 9/18/20 8:02:00 AM |
|---|---|---|

Font color: Black

| Page 20: [294] Formatted | Ronald Kwok | 9/18/20 8:02:00 AM |
|---|---|---|

Font color: Black

| Page 20: [295] Formatted | Ronald Kwok | 9/18/20 8:02:00 AM |
|---|---|---|

Font color: Black

| Page 20: [296] Formatted | Ronald Kwok | 9/18/20 8:02:00 AM |
|---|---|---|

Font color: Black

| Page 20: [297] Formatted | Ronald Kwok | 9/18/20 8:02:00 AM |
|---|---|---|

Font color: Black

| Page 20: [298] Formatted | Ronald Kwok | 9/18/20 8:02:00 AM |
|---|---|---|

Font color: Black

| Page 20: [299] Formatted | Ronald Kwok | 9/18/20 8:02:00 AM |
|---|---|---|

Font color: Black

| Page 20: [300] Formatted | Ronald Kwok | 9/18/20 8:02:00 AM |
|---|---|---|

Font color: Black

| Page 20: [301] Formatted | Ronald Kwok | 9/18/20 8:02:00 AM |
|---|---|---|

Font color: Black

| Page 20: [302] Formatted | Ronald Kwok | 9/18/20 8:02:00 AM |
|---|---|---|

Font color: Black

| Page 20: [303] Formatted | Ronald Kwok | 9/18/20 8:02:00 AM |
|---|---|---|

Font color: Black

| Page 20: [304] Formatted | Ronald Kwok | 9/18/20 8:02:00 AM |
| --- | --- | --- |

Font color: Black

| Page 20: [305] Formatted | Ronald Kwok | 9/18/20 8:02:00 AM |
| --- | --- | --- |

Font color: Black

| Page 20: [306] Formatted | Ronald Kwok | 9/18/20 8:02:00 AM |
| --- | --- | --- |

Font color: Black

| Page 20: [307] Formatted | Ronald Kwok | 9/18/20 8:02:00 AM |
| --- | --- | --- |

Font color: Black

| Page 20: [308] Formatted | Ronald Kwok | 9/18/20 8:02:00 AM |
| --- | --- | --- |

Font color: Black

| Page 20: [309] Formatted | Ronald Kwok | 9/18/20 8:02:00 AM |
| --- | --- | --- |

Font color: Black

| Page 20: [310] Formatted | Ronald Kwok | 9/18/20 8:02:00 AM |
| --- | --- | --- |

Font color: Black

| Page 20: [311] Formatted | Ronald Kwok | 9/18/20 8:02:00 AM |
| --- | --- | --- |

Font color: Black

| Page 20: [312] Formatted | Ronald Kwok | 9/18/20 8:02:00 AM |
| --- | --- | --- |

Font color: Black

| Page 20: [313] Formatted | Ronald Kwok | 9/18/20 8:02:00 AM |
| --- | --- | --- |

Font color: Black

| Page 20: [314] Formatted | Ronald Kwok | 9/18/20 8:02:00 AM |
| --- | --- | --- |

Font color: Black

| Page 20: [315] Formatted | Ronald Kwok | 9/18/20 8:02:00 AM |
| --- | --- | --- |

Font color: Black

| Page 20: [316] Formatted | Ronald Kwok | 9/18/20 8:02:00 AM |
| --- | --- | --- |

Font color: Black

| Page 20: [317] Formatted | Ronald Kwok | 9/18/20 8:02:00 AM |
| --- | --- | --- |

Font color: Black

| Page 20: [318] Formatted | Ronald Kwok | 9/18/20 8:02:00 AM |
| --- | --- | --- |

Font color: Black

| Page 20: [319] Formatted | Ronald Kwok | 9/18/20 8:02:00 AM |
|---|---|---|

Font color: Black

| Page 20: [320] Formatted | Ronald Kwok | 9/18/20 8:02:00 AM |
|---|---|---|

Font color: Black

| Page 20: [321] Formatted | Ronald Kwok | 9/18/20 8:02:00 AM |
|---|---|---|

Font color: Black

| Page 20: [322] Formatted | Ronald Kwok | 9/18/20 8:02:00 AM |
|---|---|---|

Font color: Black

| Page 20: [323] Formatted | Ronald Kwok | 9/18/20 8:02:00 AM |
|---|---|---|

Font color: Black

| Page 20: [324] Formatted | Ronald Kwok | 9/18/20 8:02:00 AM |
|---|---|---|

Font color: Black

| Page 20: [325] Formatted | Ronald Kwok | 9/18/20 8:02:00 AM |
|---|---|---|

Font color: Black

| Page 20: [326] Formatted | Ronald Kwok | 9/18/20 8:02:00 AM |
|---|---|---|

Font color: Black

| Page 20: [327] Formatted | Ronald Kwok | 9/18/20 8:02:00 AM |
|---|---|---|

Font color: Black

| Page 20: [328] Formatted | Ronald Kwok | 9/18/20 8:02:00 AM |
|---|---|---|

Font color: Black

| Page 20: [329] Formatted | Ronald Kwok | 9/18/20 8:02:00 AM |
|---|---|---|

Font color: Black

| Page 20: [330] Formatted | Ronald Kwok | 9/18/20 8:02:00 AM |
|---|---|---|

Font color: Black

| Page 20: [331] Formatted | Ronald Kwok | 9/18/20 8:02:00 AM |
|---|---|---|

Font color: Black

| Page 20: [331] Formatted | Ronald Kwok | 9/18/20 8:02:00 AM |
|---|---|---|

Font color: Black

| Page 20: [332] Formatted | Ronald Kwok | 9/18/20 8:02:00 AM |
|---|---|---|

Font color: Black

| Page 20: [333] Formatted | Ronald Kwok | 9/18/20 8:02:00 AM |
|---|---|---|

Font color: Black

| Page 20: [334] Formatted | Ronald Kwok | 9/18/20 8:02:00 AM |
|---|---|---|

Font color: Black

| Page 20: [334] Formatted | Ronald Kwok | 9/18/20 8:02:00 AM |
|---|---|---|

Font color: Black

| Page 20: [335] Formatted | Ronald Kwok | 9/18/20 8:02:00 AM |
|---|---|---|

Font color: Black

| Page 20: [336] Formatted | Ronald Kwok | 9/18/20 8:02:00 AM |
|---|---|---|

Font color: Black

| Page 20: [337] Formatted | Ronald Kwok | 9/18/20 8:02:00 AM |
|---|---|---|

Font color: Black

| Page 20: [338] Formatted | Ronald Kwok | 9/18/20 8:02:00 AM |
|---|---|---|

Font color: Black

| Page 20: [339] Formatted | Ronald Kwok | 9/18/20 8:02:00 AM |
|---|---|---|

Font color: Black

| Page 20: [340] Formatted | Ronald Kwok | 9/18/20 8:02:00 AM |
|---|---|---|

Font color: Black

| Page 20: [341] Formatted | Ronald Kwok | 9/18/20 8:02:00 AM |
|---|---|---|

Font color: Black

| Page 20: [342] Formatted | Ronald Kwok | 9/18/20 8:02:00 AM |
|---|---|---|

Font color: Black

| Page 20: [343] Formatted | Ronald Kwok | 9/18/20 8:02:00 AM |
|---|---|---|

Font color: Black

| Page 20: [344] Formatted | Ronald Kwok | 9/18/20 8:02:00 AM |
|---|---|---|

Font color: Black

| Page 20: [345] Formatted | Ronald Kwok | 9/18/20 8:02:00 AM |
|---|---|---|

Font color: Black

| Page 20: [346] Formatted | Ronald Kwok | 9/18/20 8:02:00 AM |

Font color: Black

| Page 20: [347] Formatted | Ronald Kwok | 9/18/20 8:02:00 AM |

Font color: Black

| Page 20: [348] Formatted | Ronald Kwok | 9/18/20 8:02:00 AM |

Font color: Black

| Page 20: [349] Formatted | Ronald Kwok | 9/18/20 8:02:00 AM |

Font color: Black

| Page 20: [350] Formatted | Ronald Kwok | 9/18/20 8:02:00 AM |

Font color: Black

| Page 20: [351] Formatted | Ronald Kwok | 9/18/20 8:02:00 AM |

Font color: Black

| Page 20: [352] Formatted | Ronald Kwok | 9/18/20 8:02:00 AM |

Font color: Black

| Page 20: [353] Formatted | Ronald Kwok | 9/18/20 8:02:00 AM |

Font color: Black

| Page 20: [354] Formatted | Ronald Kwok | 9/18/20 8:02:00 AM |

Font color: Black

| Page 20: [355] Formatted | Ronald Kwok | 9/18/20 8:02:00 AM |

Font color: Black

| Page 20: [356] Formatted | Ronald Kwok | 9/18/20 8:02:00 AM |

Font color: Black

| Page 20: [357] Formatted | Ronald Kwok | 9/18/20 8:02:00 AM |

Font color: Black

| Page 20: [358] Formatted | Ronald Kwok | 9/18/20 8:02:00 AM |

Font color: Black

| Page 20: [359] Formatted | Ronald Kwok | 9/18/20 8:02:00 AM |

Font color: Black

| Page 20: [360] Formatted | Ronald Kwok | 9/18/20 8:02:00 AM |

Font color: Black

| **Page 20: [361] Formatted** | **Ronald Kwok** | **9/18/20 8:02:00 AM** |

Font color: Black

| **Page 20: [362] Formatted** | **Ronald Kwok** | **9/18/20 8:02:00 AM** |

Font color: Black

| **Page 20: [363] Formatted** | **Ronald Kwok** | **9/18/20 8:02:00 AM** |

Font color: Black

| **Page 20: [364] Formatted** | **Ronald Kwok** | **9/18/20 8:02:00 AM** |

Font color: Black

| **Page 20: [365] Formatted** | **Ronald Kwok** | **9/18/20 8:02:00 AM** |

Font color: Black

| **Page 20: [366] Formatted** | **Ronald Kwok** | **9/18/20 8:02:00 AM** |

Font color: Black

| **Page 20: [367] Formatted** | **Ronald Kwok** | **9/18/20 8:02:00 AM** |

Font color: Black

| **Page 20: [368] Formatted** | **Ronald Kwok** | **9/18/20 8:02:00 AM** |

Font color: Black

| **Page 20: [369] Formatted** | **Ronald Kwok** | **9/18/20 8:02:00 AM** |

Font color: Black

| **Page 20: [370] Formatted** | **Ronald Kwok** | **9/18/20 8:02:00 AM** |

Font color: Black

| **Page 20: [371] Formatted** | **Ronald Kwok** | **9/18/20 8:02:00 AM** |

Font color: Black

| **Page 20: [372] Formatted** | **Ronald Kwok** | **9/18/20 8:02:00 AM** |

Font color: Black

| **Page 20: [373] Formatted** | **Ronald Kwok** | **9/18/20 8:02:00 AM** |

Font color: Black

| **Page 20: [374] Formatted** | **Ronald Kwok** | **9/18/20 8:02:00 AM** |

Font color: Black

| **Page 20: [375] Formatted** | **Ronald Kwok** | **9/18/20 8:02:00 AM** |

Font color: Black

| Page 20: [376] Formatted | Ronald Kwok | 9/18/20 8:02:00 AM |
|---|---|---|

Font color: Black

| Page 20: [377] Formatted | Ronald Kwok | 9/18/20 8:02:00 AM |
|---|---|---|

Font color: Black

| Page 20: [378] Formatted | Ronald Kwok | 9/18/20 8:02:00 AM |
|---|---|---|

Font color: Black

| Page 20: [379] Formatted | Ronald Kwok | 9/18/20 8:02:00 AM |
|---|---|---|

Font color: Black

| Page 20: [380] Formatted | Ronald Kwok | 9/18/20 8:02:00 AM |
|---|---|---|

Font color: Black

| Page 20: [381] Formatted | Ronald Kwok | 9/18/20 8:02:00 AM |
|---|---|---|

Font color: Black

| Page 20: [382] Formatted | Ronald Kwok | 9/18/20 8:02:00 AM |
|---|---|---|

Font color: Black

| Page 20: [383] Formatted | Ronald Kwok | 9/18/20 8:02:00 AM |
|---|---|---|

Font color: Black

| Page 20: [384] Formatted | Ronald Kwok | 9/18/20 8:02:00 AM |
|---|---|---|

Font color: Black

| Page 20: [385] Formatted | Ronald Kwok | 9/18/20 8:02:00 AM |
|---|---|---|

Font color: Black

| Page 20: [386] Formatted | Ronald Kwok | 9/18/20 8:02:00 AM |
|---|---|---|

Font color: Black

| Page 20: [387] Formatted | Ronald Kwok | 9/18/20 8:02:00 AM |
|---|---|---|

Font color: Black

| Page 20: [388] Formatted | Ronald Kwok | 9/18/20 8:02:00 AM |
|---|---|---|

Font color: Black

| Page 20: [389] Formatted | Ronald Kwok | 9/18/20 8:02:00 AM |
|---|---|---|

Font color: Black

| Page 20: [390] Formatted | Ronald Kwok | 9/18/20 8:02:00 AM |
|---|---|---|

Font color: Black

| Page 20: [391] Formatted | Ronald Kwok | 9/18/20 8:02:00 AM |
|---|---|---|

Font color: Black

| Page 20: [392] Formatted | Ronald Kwok | 9/18/20 8:02:00 AM |
|---|---|---|

Font color: Black

| Page 20: [393] Formatted | Ronald Kwok | 9/18/20 8:02:00 AM |
|---|---|---|

Font color: Black

| Page 20: [394] Formatted | Ronald Kwok | 9/18/20 8:02:00 AM |
|---|---|---|

Font color: Black

| Page 20: [395] Formatted | Ronald Kwok | 9/18/20 8:02:00 AM |
|---|---|---|

Font color: Black

| Page 20: [396] Formatted | Ronald Kwok | 9/18/20 8:02:00 AM |
|---|---|---|

Font color: Black

| Page 20: [397] Formatted | Ronald Kwok | 9/18/20 8:02:00 AM |
|---|---|---|

Font color: Black

| Page 20: [398] Formatted | Ronald Kwok | 9/18/20 8:02:00 AM |
|---|---|---|

Font color: Black

| Page 20: [399] Formatted | Ronald Kwok | 9/18/20 8:02:00 AM |
|---|---|---|

Font color: Black

| Page 20: [400] Formatted | Ronald Kwok | 9/18/20 8:02:00 AM |
|---|---|---|

Font color: Black

| Page 20: [401] Formatted | Ronald Kwok | 9/18/20 8:02:00 AM |
|---|---|---|

Font color: Black

| Page 20: [402] Formatted | Ronald Kwok | 9/18/20 8:02:00 AM |
|---|---|---|

Font color: Black

| Page 20: [403] Formatted | Ronald Kwok | 9/18/20 8:02:00 AM |
|---|---|---|

Font color: Black

| Page 26: [404] Deleted | Ronald Kwok | 9/18/20 8:02:00 AM |
|---|---|---|